# Assessing downscaling methods to simulate hydrologically relevant weather scenarios from a global atmospheric reanalysis: case study of the Upper Rhône River (1902-2009)

Caroline Legrand[1], Benoît Hingray[1], Bruno Wilhelm[1,†], and Martin Ménégoz[1]

[1]Univ. Grenoble Alpes, CNRS, INRAE, IRD, Grenoble INP*, IGE, 38000 Grenoble, France
[1]*Institute of Engineering and Management Univ. Grenoble Alpes
[†]deceased, 5 April 2022

**Correspondence:** Caroline Legrand (caroline.legrand@univ-grenoble-alpes.fr)

**Abstract.** We assess the ability of two modeling chains to reproduce, over the last century (1902-2009) and from large-scale atmospheric information only, the temporal variations in river discharges, low flow sequences and flood events, observed at different locations of the Upper Rhône River catchment, an alpine river straddling France and Switzerland $(10,900\,\text{km}^2)$. The two modeling chains are made up of a downscaling model, either statistical (SCAMP) or dynamical (MAR), and the glacio-hydrological model GSM-SOCONT. Both downscaling models, forced by atmospheric information from the global atmospheric reanalysis ERA-20C, provide time series of daily scenarios of precipitation and temperature used as inputs to the hydrological model. With hydrological regimes ranging from highly glaciated ones in its upper part to mixed ones dominated by snow and rain downstream, the Upper Rhône River catchment is ideal to evaluate the different downscaling models in contrasting and demanding hydro-meteorological configurations where the interplay between weather variables, both in space and time, is determinant. Whatever the river sub-basin considered, the simulated discharges are in good agreement with the reference ones, provided that the weather scenarios are bias-corrected. The observed multi-scale variations in discharges (daily, seasonal and interannual) are well reproduced. The low frequency hydrological situations, such as annual monthly discharge minima (used as low flows proxy indicators) and annual daily discharge maxima (used as flood proxy indicators) are reasonably well reproduced. The observed increase in flood activity over the last century is also rather well reproduced. The observed low flow activity is conversely overestimated, and its variations from one sub-period to another are only partially reproduced. Bias correction is crucial for both precipitation and temperature and both downscaling models. For the dynamical one, a bias correction is also essential to get realistic daily temperature lapse rates. Uncorrected scenarios lead to irrelevant hydrological simulations, especially for the sub-basins at high elevation, mainly due to irrelevant snowpack dynamic simulations. The simulations also highlight the difficulty to simulate precipitation dependency to elevation over mountainous areas.

## 1 Introduction

Climate change is expected to exacerbate flood hazard through an intensification of the hydrological cycle, which will likely alter the magnitude, frequency, and/or seasonality of floods (Blöschl et al., 2017). Another concern is that of future low flows

and drought situations, which are also expected to be more frequent, longer and more intense (Ruosteenoja et al., 2018; Masson-Delmotte et al., 2021). However, projecting the possible evolution of hydrological extremes at the catchment-scale is still challenging, and although a large number of works has been developed for a number of rivers worldwide, considerable uncertainty about possible future changes remains, both for changes in the intensity and frequency of extreme events (e.g. Kundzewicz et al., 2016; Roudier et al., 2016; Vidal et al., 2016; Di Sante et al., 2021; Evin et al., 2021; Lemaitre-Basset et al., 2021).

Hydrological scenarios required for climate change impact studies are commonly obtained by simulation with hydrological models from ensembles of projected meteorological scenarios. To allow for a relevant impact assessment, meteorological scenarios have to fulfill some constraints imposed by the strong non-linearity and the high spatial and temporal variability of hydrological processes (e.g. strong dependency of temperature, radiative fluxes, precipitation, etc, on elevation and aspect in mountainous environments). For instance, the meteorological scenarios have to be bias-corrected (e.g. with respect to space and seasonality) and to have rather high spatial and temporal resolution (Lafaysse et al., 2014). Because such requirements are not fulfilled by General Circulation Models (GCMs), meteorological scenarios are classically obtained with downscaling models, either dynamical or statistical.

Dynamical Downscaling Models (DDMs) are Regional Climate Models (RCMs) nested within a GCM to generate fine resolution climate information (Giorgi and Mearns, 1991). They solve the full equations of mass, energy, and momentum conservation laws in the atmosphere to account for the physical interactions of land-atmosphere processes with consideration of the heterogeneity of topography, soil, vegetation, and climate variables in a region or a catchment. Over the past three decades, RCMs have been widely used in a number of studies for hydrological purposes (e.g. Arnell et al., 2003; Leander et al., 2008; Leung and Qian, 2009; see Tapiador et al., 2020 for a complete review).

On the other hand, Statistical Downscaling Models (SDMs) are based on empirical relationships identified from observations between large-scale atmospheric variables (or predictors) and local weather variables (or predictands) (Von Storch et al., 1993). Weather scenarios derived from SDMs are produced from time series of predictors extracted from large-scale atmospheric outputs of climate models. SDMs have been widely used to i) generate local weather scenarios for past or future climates from GCM experiments (e.g. Wilby et al., 1999; Hanssen-Bauer et al., 2005; Boé et al., 2007; Lafaysse et al., 2014; Dayon et al., 2015), ii) produce local weather forecasts from large-scale weather forecasts of numerical weather prediction models (e.g. Obled et al., 2002; Gangopadhyay et al., 2005; Marty et al., 2012), and iii) produce reconstructions of past weather conditions from observations and global atmospheric reanalysis data (e.g. Wilby and Quinn, 2013; Kuentz et al., 2015; Bonnet et al., 2020; Devers et al., 2021).

A general discussion of the pros-and-cons of selecting one downscaling approach over the other has been the subject of multiple manuscripts (see Fowler et al., 2007 or Maraun et al., 2010 for a review). DDMs have the advantage of being formulated on physical principles, but their simulations are computationally intensive and time-consuming, with a resolution that is generally still to coarse to catch local processes. They are obviously not free of limitations and their outputs often need to be corrected for impact studies. In contrast, SDMs are very popular because of their computational efficiency and ease of use.

However, they may sometimes miss some important interactions and/or correlations between meteorological variables, both in space and time.

A large number of studies have set up modeling chains to simulate river discharges in response to large-scale atmospheric trajectories over specific periods. As shown by Wood et al. (2004) and Quintana Seguí et al. (2010), for instance, the choice of the downscaling approach can strongly influence the simulation results. Not all downscaling approaches are necessarily relevant for the targeted simulations, but impact-oriented assessments can guide the model selection. For climate impact analyses focusing on hydrology, for instance, the modeling chains have to be able to reproduce in a relevant way the multi-scale hydrological variations that result from the large-scale atmospheric trajectories of the considered period (e.g. Lafaysse et al., 2014).

In this work, we assess and compare the ability of two modeling chains to reproduce, from large-scale atmospheric information only, the observed temporal variations in discharges in the Upper Rhône River (URR) catchment. The modeling chains are made up of a downscaling model, either statistical (SCAMP; Raynaud et al., 2020) or dynamical (MAR; Gallée and Schayes, 1994), and the glacio-hydrological model GSM-SOCONT (Schaefli et al., 2005). Both downscaling models are forced by the global atmospheric reanalysis ERA-20C (Poli et al., 2016).

We combine three innovative features. i) We evaluate the modeling chains in contrasting and demanding hydro-meteorological configurations where the interplay between weather variables, both in space and time, is determinant. The URR catchment, a mesoscale alpine catchment straddling France and Switzerland, indeed presents a number of different hydrological regimes, ranging from highly glaciated ones in its upper part to mixed ones dominated by snow and rain downstream. ii) We evaluate the modeling chains over the entire 20th century, a period long enough to assess the ability of the modeling chains to reproduce daily variations in observed discharges, low frequency events (annual monthly discharge minima and annual daily discharge maxima), and variations in low flow and flood activities (rate of occurrence of flood/ low flow discharges above/below a given threshold). iii) For both downscaling models, we evaluate the need for additional bias correction prior to application of the hydrological model for precipitation, temperature and temperature lapse rates.

The paper is structured as follows. Section 2 describes the study area and Sect. 3 the data. Section 4 presents the components of the different modeling chains. A meteorological and hydrological assessment is carried out in Sect. 5. Results are discussed in Sect. 6. Finally, Sect. 7 sums up the main results of this study and outlines future lines of research.

## 2 Study area

The Upper Rhône River (URR) catchment $(10,900\,\mathrm{km}^2)$ covers the southwestern part of the Swiss Alps and a part of the northern French Alps (Fig. 1). The altitude of this catchment ranges from $300\,\mathrm{m}$ to above $4,800\,\mathrm{m}$ at the top of the Mont Blanc. The presence of steep slopes makes this area particularly prone to natural hazards such as landslides, floods and avalanches, which are strongly connected to meteorological conditions (Beniston, 2006; Raymond et al., 2019).

In this region, the climate is continental and the temporal variability of precipitation and temperature is high. Mean annual precipitation ranges from $600\,\mathrm{mm}$ (in some parts of Wallis canton, Switzerland) to $1100\,\mathrm{mm}$ (Chamonix, France). It rains from

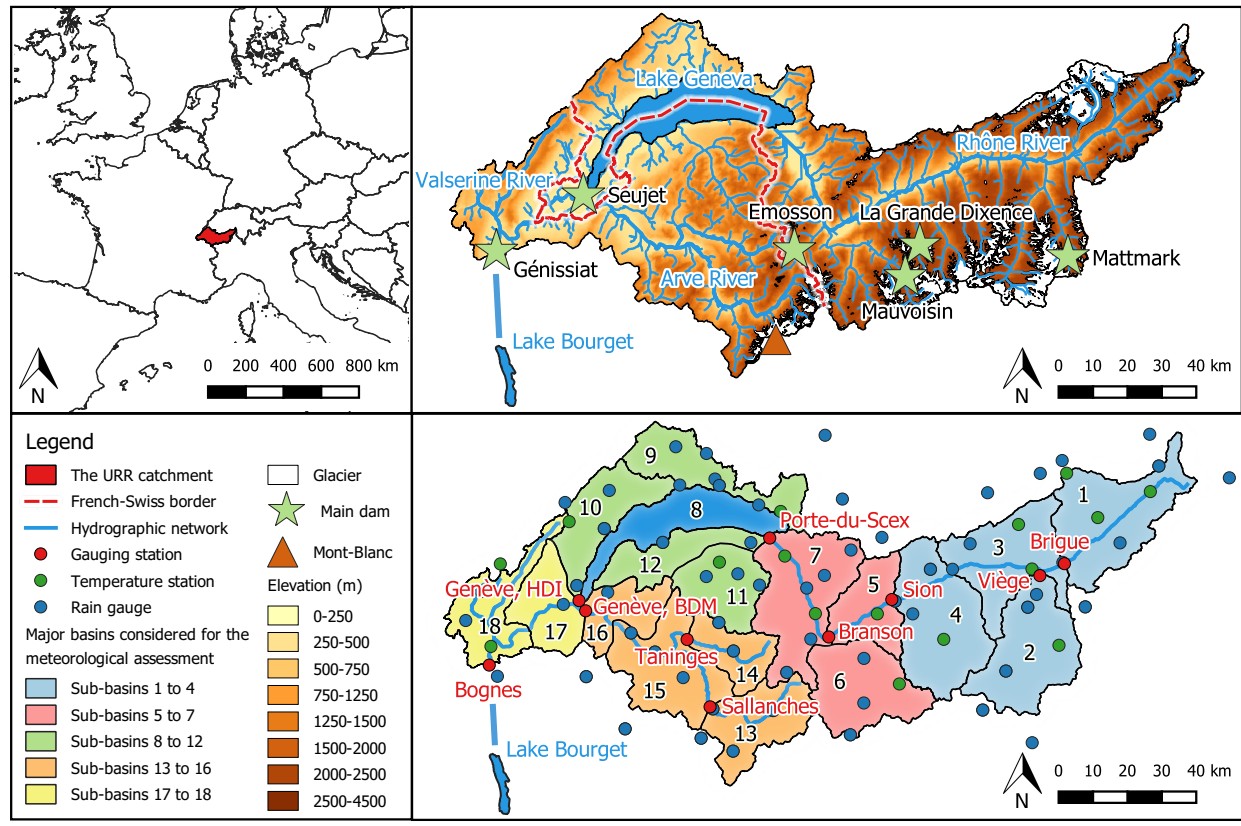

**Figure 1.** Study area. (**top**) Localisation and hydrological characteristics of the Upper Rhône River (URR) catchment. (**bottom**) Locations of the different weather and gauging stations. Genève, HDI: Genève, Halle-de-l'Ile. Genève, BDM: Genève, Bout-du-Monde.

30 % to 45 % of the days, with an annual maximum daily precipitation reaching locally 45 mm day$^{-1}$ to 105 mm day$^{-1}$ on average (Isotta et al., 2014). Glaciated areas covered 17 % of the catchment upstream to Lake Geneva and 10 % of the whole URR catchment in 2015 (GLIMS, 2015).

The gauging station of Bognes (Rhône@Bognes hereafter) records daily mean discharges at the outlet of the URR catchment. It is located at Injoux-Génissiat (France), 46 km downstream of the confluence of the Rhône and the Arve Rivers. Before the confluence of the two rivers, the hydrological regime of the URR catchment is glacio-nival: the important seasonal snowpack dynamics, combined with the late summer contribution of glacier melt, results in a strong seasonality of flow. Flood events are observed in spring due to snowmelt and in late summer/fall as a result of events with large to very large rainfall amounts (e.g. the so-called "Binn-Simplon" situations; OFEG, 2002). The URR hydrological regimes turns to pluvio-nival downstream, with the successive contributions of lower elevation areas, especially that of the Arve River. Flood events from the Arve River mainly result from extreme rainfall events in fall (e.g. the so-called "retour d'Est events"; Metzger, 2023).

| Category | Item | 1900 | 1961 | 2010 | 2015 |
|---|---|---|---|---|---|
| Atmospherical data | ERA-20C reanalysis | ██████████████████████████████████ | | | |
| Meteorological data | Precipitation | | | ███████████ | |
| | Temperature | | | ███████████ | |
| Hydrological data | Rhône@Brigue | | | *1965* ██████ | |
| | Rhône@Viège | *1922* ████████ | *1963* ▨▨▨▨▨ | | |
| | Rhône@Sion | *1916* ██████ | *1956* ▨▨▨▨▨ | | |
| | Rhône@Branson | *1941* ████ | *1956* ▨▨▨▨▨ | | |
| | Rhône@Porte-du-Scex | *1905* ████████ | *1956* ▨▨▨▨▨ | | |
| | Rhône@Genève, HDI | *1923* ██████ | *1956* ▨▨▨▨▨ | | |
| | Arve@Sallanches | | | *1965* ██████ | |
| | Arve@Taninges | *1948* ████████ | | | |
| | Arve@Genève, BDM | *1904* ████████████ | | | |
| | Rhône@Bognes | *1920* ████ *1947* ▨▨▨▨▨ | | | |
| Hydrological simulations | From MAR weather | *1902* ██████████████████ *2009* | | | |
| | From SCAMP weather | *1902* ██████████████████ *2009* | | | |

**Figure 2.** Data used. Periods for which daily variables are available are shown in green. Hatchings indicate periods for which the hydrological regime of the Upper Rhône River (URR) catchment is significantly altered by dams. Rhône@Genève, HDI: Rhône@Genève, Halle-de-l'Ile. Arve@Genève, BDM: Arve@Genève, Bout-du-Monde. MAR: dynamical downscaling model. SCAMP: statistical downscaling model.

Upstream to Lake Geneva, the URR hydrological regime is considered to be natural until the 1950s, before the construction of several large seasonal water reservoirs, mainly used for hydroelectricity production and flood protection. The reservoirs store the snow and glacier melt inflows from high elevation areas in spring and summer for hydroelectricity production in winter. The total storage capacity of all reservoirs in the catchment is $1200 \times 10^6$ m$^3$ or roughly 20 % of the annual catchment precipitation. Since the 1950s, the URR hydrological regime is thus significantly altered, with a reduced seasonality (higher river discharges in winter, smaller ones in spring/summer) and reduced flood discharges in summer and fall (Hingray et al., 2014). Downstream, Lake Geneva, which is the largest natural water reservoir in Western Europe (580 km$^2$), has a significant natural buffer effect on river flows. Its influence on flows has even been exacerbated since 1884 with the construction of a downstream regulation weir for flood protection and hydroelectricity production (Grandjean, 1990).

## 3    Data

### 3.1    Meteorological and hydrological data

The density of weather stations covering the URR catchment is rather high. Daily meteorological variables are available from January 1, 1961 to December 31, 2015 for 62 rain gauges and 39 temperature stations in the catchment and in the areas bordering it. The period for which observed daily discharges are available depends on the gauging station. It covers up to one century for the URR at Rhône@Porte-du-Scex, the outlet of the river to Lake Geneva (Fig. 2). Note that for some sub-basins, there is no observed meteorological data concomitant with "natural" discharge data. Weather and hydrological data were provided by i) the Office Fédéral de l'EnVironnement (OFEV) for the Swiss part, and ii) Météo France and the Banque Hydro for the French part. The observed level of Lake Geneva has also been available from the OFEV since 1886 at daily time step.

### 3.2    Atmospheric data

The global atmospheric reanalysis ERA-20C from the European Centre for Medium-Range Weather Forecasts (ECMWF; Poli et al., 2016) is used i) as boundary conditions for the dynamical downscaling model MAR, and ii) directly as input for the statistical downscaling model SCAMP. This reanalysis provides 6-hourly data over the 1900-2010 period at a $1.25°$ spatial resolution for a number of atmospheric variables (e.g. geopotential height, wind speed, temperature and humidity of air masses). The URR catchment is covered by 8 ERA-20C grid points.

## 4    Methods

### 4.1    Modeling chains

The modeling chains developed in this study are made up of i) a downscaling model, either dynamical (MAR) or statistical (SCAMP), to generate time series scenarios of regional weather from the global atmospheric reanalysis ERA-20C, and ii) the glacio-hydrological model GSM-SOCONT, to simulate the corresponding discharge time series scenarios at different gauging stations in the URR catchment.

In this work, we also assess the need for a bias correction of the downscaled weather scenarios prior to their use for hydrological simulations. In some of the configurations considered in the following, a bias correction step is additionally included in the modeling chains. This applies to precipitation, temperature and temperature lapse rates.

The different models considered in the simulation chains are described below, in Sect. 4.2 for the hydrological model, in Sect. 4.3 for the dynamical and statistical downscaling models, and in Sect. 4.4 for the bias correction model. Section 4.5 sums up the different experiments carried out with these modeling chains.

## 4.2 Hydrological model

### 4.2.1 The GSM-SOCONT model

The discharge simulations were performed with a semi-distributed and daily configuration of GSM-SOCONT (Glacier and SnowMelt SOil CONTribution model; Schaefli et al., 2005), a bucket-type model that uses time series of mean areal precipitation and temperature as inputs for each hydrological unit (see Fig. S1 in Supplementary Materials). The URR catchment is divided into 18 sub-basins. They were selected so that they are roughly the same size, and that a gauging station is located at the outlet of a sub-basin wherever possible (Obled et al., 2009). The ice-covered and the ice-free parts of each sub-basin, extracted from the Global Land Ice Measurements from Space (GLIMS, 2015), are considered separately and divided into 500 m elevation bands.

For each elevation band, further referred to as Relatively Homogeneous Hydrological Unit (RHHU), daily mean areal precipitation and temperature are estimated from neighboring weather stations using the Thiessen's weighting method and a regional and time-varying temperature-elevation relationship (Hingray et al., 2010). The choice of the Thiessen's weighting method is discussed in Sect. 6.4. Mean areal temperature is estimated for the mean RHHU elevation. Mean areal precipitation is supposed to be solid if mean areal temperature is smaller than a critical temperature $Tc_1$, liquid if mean areal temperature is higher than a critical temperature $Tc_2$, and mixed otherwise. The $Tc_1$ and $Tc_2$ values are fixed to $0\ ^\circ C$ and $2\ ^\circ C$ following Hingray et al. (2010) and Froidurot et al. (2014).

For each RHHU, the snowpack temporal evolution is computed based on mean areal precipitation and temperature time series. Each simulation day, solid and/or liquid precipitation are added to the snowpack storage and/or to its liquid water content respectively. The liquid water content is also increased by snowmelt if any and the snowpack is increased by refreezing water if any. Snowmelt (or refreezing water) is estimated with a degree-day model from the positive (respectively negative) temperature degrees of the day. When the liquid water content reaches the snowpack retention capacity, an "equivalent rainfall" is released. The retention capacity is assumed to be proportional to the snowpack water equivalent following Kuchment and Gelfan (1996). For glaciated RHHUs, ice melt can additionally occur, namely when the glacier surface is free of snow. Ice melt is also estimated with a degree-day model.

For glaciated RHHUs, the rainfall and meltwater-runoff transformation is completed through two linear reservoirs, one for ice melt and one for the rainfall or equivalent rainfall. For non-glaciated RHHUs, the rainfall or equivalent rainfall is separated into infiltration and effective rainfall. Infiltration feeds a non-linear soil reservoir which produces deep infiltration and sub-surface flow. Infiltration (resp. actual evapotranspiration) is estimated from equivalent rainfall (resp. potential evapotranspiration) and simulated soil moisture. Effective rainfall feeds a linear overland reservoir which produces direct runoff.

Potential evapotranspiration (PET) estimates were derived from PET estimates of the Climate Research Unit (CRU) produced at a $0.5^\circ$ spatial resolution from a variant of the Penman-Monteith formula (Harris et al., 2014). In order to derive PET estimates for high elevation RHHUs, PET was assumed to be a linear function of temperature T. The PET-T relationship was estimated for the region on a monthly basis from the CRU PET and T estimates produced for the 1900-2010 period.

The discharge simulated at the outlet of each sub-basin is the sum of the different discharge components produced by the different RHHUs from the glaciated and non-glaciated areas of the sub-basin. Discharges simulated for the sub-basins are summed to produce those at each gauging station and at the catchment outlet. The daily time step used in the simulations and the rather small size of the catchment area allow to disregard the routing of discharges through the river network.

Seven parameters have to be estimated for each ice-free RHHU (e.g. snowmelt degree-day factor, storage capacity and recession coefficient of reservoirs). Three additional parameters have to be estimated for glaciated RHHUs (ice melt degree-day factor, and recession coefficient of ice and snow reservoirs).

### 4.2.2 Hydrological model calibration

In most cases (as in here), hydrological models simulate the natural behavior of the considered catchments. One frequent approach for estimating the model parameters in such a configuration requires a time period with concomitant observations of weather and "natural" discharges, i.e. concomitant observations of weather and discharges not altered by anthropic activities or waterworks. For most sub-basins of the URR catchment, there is no such period. Daily weather observations are mainly available from the 1960s (here for period P* = 1961-2015) whereas their hydrological behavior was natural until the 1950s only, strongly altered after by dams.

The method used for parameters estimation was thus adapted to the sub-basins data configuration, depending i) on the "perturbation" level of the sub-basins hydrological behavior during the period P*, and ii) on the availability of flow observation records for the period prior the 1950s. For gauged sub-basins for which the hydrological behavior can be considered as natural (or at least not significantly altered) over the period P* (or at least over a significant sub-period of P*), parameters were estimated with a classical hydrological calibration approach, minimizing an objective function estimated from simulated and observed discharges time series on the same period, namely:

$$F_{natural} = 1 - NSE_{chrono} \tag{1}$$

where $NSE_{chrono}$ is the Nash-Sutcliffe Efficiency criterion (Nash and Sutcliffe, 1970) obtained from day-to-day deviations between observed and simulated daily discharges.

For gauged sub-basins for which the hydrological behavior is significantly altered over P* and for which "natural" flow observations are available prior to 1950, parameters were estimated based on hydrological signatures (Sivapalan et al., 2003; Winsemius et al., 2009). In the present case, parameters were calibrated so that simulated signatures reproduce at best observed ones but observed and simulated signatures come from different periods following Hingray et al. (2010) (e.g. 1961-2015 and 1922-1963 respectively for the Viège sub-basin).

The signatures considered here are the interannual daily regime (366 values) and the statistical distribution of the annual daily discharge maxima. The objective function is a combination of the $NSE$ criterion and the Kolmogorov-Smirnov distance $d_{KS}$, applied respectively between simulated and observed signatures:

$$F_{altered} = 0.5 \times (1 - NSE_{regime}) + 0.5 \times \frac{d_{KS}}{\max(Q_{obs_{KS}}, Q_{sim_{KS}})} \tag{2}$$

where $d_{KS} = |Q_{obs_{KS}} - Q_{sim_{KS}}|$ is the Kolmogorov-Smirnov distance with $Q_{obs_{KS}}$ and $Q_{sim_{KS}}$ the corresponding discharge percentiles in the observed and simulated distributions.

For ungauged sub-basins, parameters were obtained via regionalization, following the methodological recommendations of Bárdossy (2007) and Viviroli et al. (2009). All the ungauged sub-basins located upstream of a given gauging station were calibrated at the same time and forced to share the same parameters set. The discharge time series used for the calibration is the time series simulated with this multiple sub-basins configuration at the downstream gauging station, where observations are available. The ungauged sub-basins can be seen in Fig. 1. The sub-basins grouped together for the calibration of their

parameters are also listed in Table S2 in Supplementary Materials.

    In practice, whatever the calibration configuration and the calibration objective function, we used the automatic calibration algorithm DDS (Dynamically Dimensioned Search; Tolson and Shoemaker, 2007). The objective functions values and the results of the classical and signature-based calibrations are given for each sub-basin in Table S2, Fig. S3 and Fig. S4 respectively in Supplementary Materials.

Depending on available data, we thus considered different calibration approaches. Because of this data context, it is not always easy to assess the relevance of the calibration. For sub-basins where concomitant weather and "natural" discharge data are available over a sufficiently long period, this assessment can be made with a classical split-sample test. The split-sample tests carried out by Schaefli et al. (2005) for three URR sub-basins show that the classical calibration is efficient and robust in this context.

For sub-basins for which the hydrological behavior is significantly altered by dams, the split-sample test is not possible, as the data of the entire period have to be considered for the signature-based calibration. The analyses described below nevertheless suggest that the signature-based calibration remains efficient and robust in our context.

    To assess this, we recalibrated the parameters of the four URR sub-basins that present a natural (or at least not significantly altered) hydrological regime, using only the hydrological signatures. In a first analysis, the signatures were estimated using

the same data (period P0) than those considered for the classical calibration (see Fig. S5 in Supplementary Materials). The simulated time series remain in good agreement with the observed ones. The NSE coefficients are logically lower than those obtained with the classical calibration approach, but the differences are quite small (see Fig. S6 in Supplementary Materials).

    In a second analysis, we carried out a similar split-sample test for the signature-based calibration. To do this, we split the period into two (sub-periods P1 and P2), we recalibrated the parameters using only the hydrological signatures of period P1

with the weather data from P1 and, with this set of parameters, we simulated over period P2 the discharge time series from the weather data of P2. The simulated time series remain in good agreement with those observed. They are also in good agreement with those obtained with a signature-based calibration when signatures are derived from the entire period P0. The NSE coefficients are logically lower, but the differences are quite small (see Fig. S7 in Supplementary Materials).

### 4.2.3   Modeling the behavior of Lake Geneva

An ad hoc model was developed to simulate the influence of Lake Geneva on river flows. The lake has a natural buffer effect on flows, additionally altered with its regulation. From the 1990s, the outflows follow the main regulation objectives of the 1997

settlement (CERCG, 1997), namely i) a target water level in the lake for each calendar day, ii) the environmental low flow to be satisfied downstream, and iii) the discharge to not exceed except during periods of high water in the lake. The day-to-day variations in lake outflows are additionally driven by the hydroelectric production required from the plant at the lake outlet to
satisfy a part of the regional electricity demand.

    By lack of appropriate data, we here only accounted for the natural buffer effect of the lake and for the main regulation objectives mentioned above. The natural buffer effect is simply modeled with the classical three equations system needed to simulate the behavior of unregulated reservoirs, namely a water balance, a water level-storage and a reservoir flow equation (Hingray et al., 2014). Here, a simple linear water level-storage relationship is considered and the outflow that would be
obtained without regulation is assumed to be proportional to the volume stored in the reservoir. For the water balance, changes in storage are obtained each day from reservoir inflows (direct precipitation, upstream and lateral basins flows) and losses (evaporation, lake outflow), where evaporation is estimated with the Rohwer's equation (Rohwer, 1931).

## 4.3    Downscaling models

### 4.3.1    The dynamical downscaling model MAR

The Modèle Atmosphérique Régional (MAR; Gallée, 1995; Gallée and Schayes, 1994; Gallée et al., 1996) is a hydrostatic primitive equation model for regional atmospheric simulations. It includes a detailed scheme of clouds microphysics with six prognostic equations for specific humidity, cloud droplet concentration, cloud ice crystals (concentration and number), concentration of precipitating snow particles and rain drops. The convective adjustment is parameterized according to Bechtold et al. (2001). MAR is coupled to the one-dimensional land surface scheme SISVAT (Soil Ice Snow Vegetation Atmosphere
Transfer; De Ridder and Schayes, 1997; Gallée et al., 2001) that includes a snow multilayer scheme (Brun et al., 1992; Gallée and Duynkerke, 1997).

    First designed for polar regions, especially Antarctica and Greenland (e.g. Gallée et al., 1996), MAR was also applied over other regions worldwide, such as midlatitudes areas (e.g. Wyard et al., 2017; Doutreloup et al., 2019), mountainous areas (e.g. Himalaya; Ménégoz et al., 2013) and West Africa (e.g. Chagnaud et al., 2020). For the present study, we used the MAR
simulation forced by the global atmospheric reanalysis ERA-20C over the 1902-2009 period and produced at a 7 km resolution over the European Alps (Ménégoz et al., 2020; Beaumet et al., 2021). The URR catchment is covered by 281 MAR grid points.

### 4.3.2    The statistical downscaling model SCAMP

SCAMP (Sequential Constructive Atmospheric Analogues for Multivariate weather Predictions; Raynaud et al., 2020) is a statistical downscaling model based on atmospheric analogues (Lorenz, 1969). It assumes that similar large-scale atmospheric
configurations lead to similar local or regional weather situations (e.g. Obled et al., 2002; Chardon et al., 2014). The simulation process is partly stochastic: for each simulation day, one of the 30-nearest analogues is randomly selected and used as weather scenario for this day. SCAMP can thus be used to generate multiple weather scenarios. In the present work, SCAMP was used

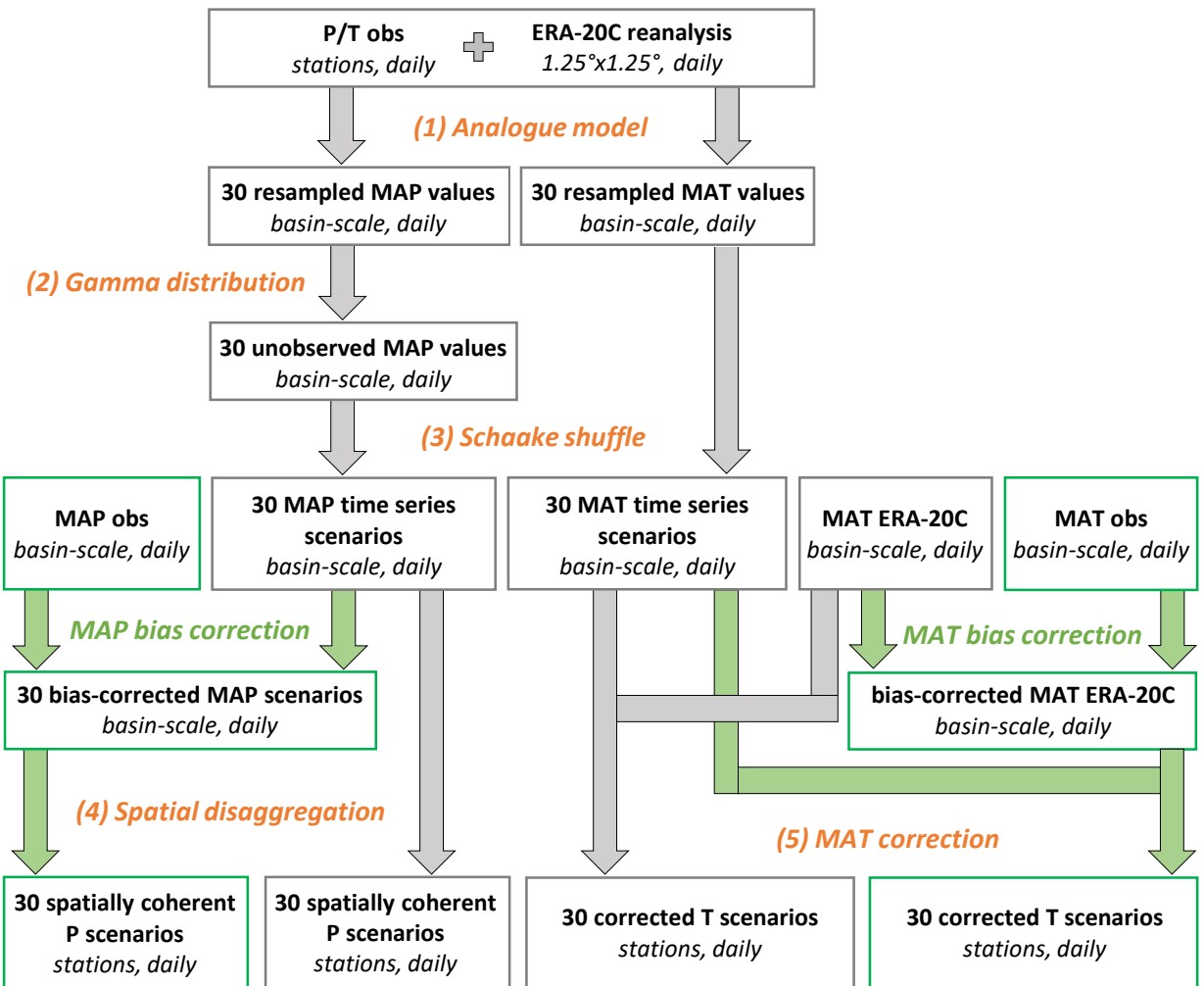

**Figure 3.** Scheme of the statistical downscaling model SCAMP. (**orange**) Components used in the SCAMP simulation. (**green**) Additional components used in the SCAMP bias-corrected simulation. (**bold**) Outputs obtained after each step. (**italics**) Spatial and temporal resolutions. P: Precipitation. T: Temperature. MAP: Mean Areal Precipitation. MAT: Mean Areal Temperature.

to generate 30 time series of daily spatial weather scenarios for the 1902-2009 period from the global atmospheric reanalysis ERA-20C outputs. The simulation process is described in the following and summarized in Fig. 3.

(1) For each day of the simulation period, the 30-nearest atmospheric analogue days are identified from candidate days available in the archive period (1961-2009 in the present case). The candidate days are the days found within a 61-days calendar window centered on the target day. A two-step analogue selection is considered. The 100-nearest analogues in terms of large-scale atmospheric circulation are first identified. The selection criterion is the Teweles-Wobus Score (Teweles and Wobus, 1954) applied to the daily geopotential heights at 1000 and 500 hPa. It quantifies the similarity between fields from their

shapes, informing thus on the origin of air masses. The 30-nearest analogs are sub-selected from these 100-nearest analogs, based on small-scale atmospheric features following Raynaud et al. (2020) (namely 600 hPa vertical velocities and large-scale temperature at 2 m from September to May, large-scale precipitation otherwise).

(2) Following Chardon et al. (2016), SCAMP is used to generate Mean Areal Precipitation (MAP) and Mean Areal Temperature (MAT) values for the whole URR catchment. To generate weather scenarios with values different from observations,
SCAMP extends the analogue method with a random generation process from a day-to-day adjusted statistical distribution (Chardon et al., 2018; Raynaud et al., 2020). For precipitation, a Gamma distribution is fitted each day to the 30 regional MAP values obtained for the whole URR catchment from the analogues. The distribution is then used to generate a new sample of 30 regional MAP values, thus unobserved (and possibly higher than observations). This adaptation was shown to improve the simulation of maximum regional MAP accumulations (Raynaud et al., 2020, Fig. 8 therein).

(3) 30 regional MAP and MAT time series scenarios are produced from the 30 regional MAP and MAT values generated each day. To improve the temporal consistency between consecutive days in each time series, the 30 regional MAP and MAT scenarios obtained for each day are paired with the 30 scenarios of the previous day with a Schaake shuffle reordering approach (Clark et al., 2004; Raynaud et al., 2020).

(4) Following Mezghani and Hingray (2009) and Viviroli et al. (2022), the regional MAP time series scenarios are finally
disaggregated with a non-parametric method of fragment to produce spatial MAP time series scenarios for the URR catchment (with one mean areal precipitation value for each RHHU of the hydrological model). In practice, the regional MAP scenario of each day is spatially disaggregated using the spatial pattern of an analog day for which weather observations are available.

(5) The large-scale atmospheric features considered for the identification of the analogues are not as informative for temperatures. The temperature additive correction method of Kuentz et al. (2015) was thus used to make each day the regional
MAT time series scenarios coherent with the regional MAT time series of the global atmospheric reanalysis ERA-20C. For each prediction day and each scenario, the correction factor is the difference between the regional MAT value of the ERA-20C reanalysis and the regional MAT value of the scenario. The correction factor is applied to correct the temperature of all stations for this scenario. For instance, if the daily regional MAT value of the scenario is 2 °C warmer than the regional MAT value of the ERA-20C reanalysis, then all local temperatures of the scenario are lowered by 2 °C.

## 4.4 Bias correction

As shown in Sect. 5.1, the simulated weather scenarios from the dynamical downscaling model MAR and the statistical downscaling model SCAMP can be significantly different from the observations. Corrected precipitation and temperature scenarios are also considered in the following. A quantile mapping bias correction (BC hereafter) was used for both variables and both models (Déqué, 2007). The different regional MAP and MAT time series were corrected using the observed regional MAP and
305 MAT time series as references. As commonly done, the correction is additive for MAT and multiplicative for MAP, and one BC function was estimated for each calendar month respectively.

For both models, an analytical BC function was used to ease interpolations of correction factors estimated between empirical percentiles (a degree 4 polynomial was fitted to the empirical correction factors). To avoid non relevant extrapolations of

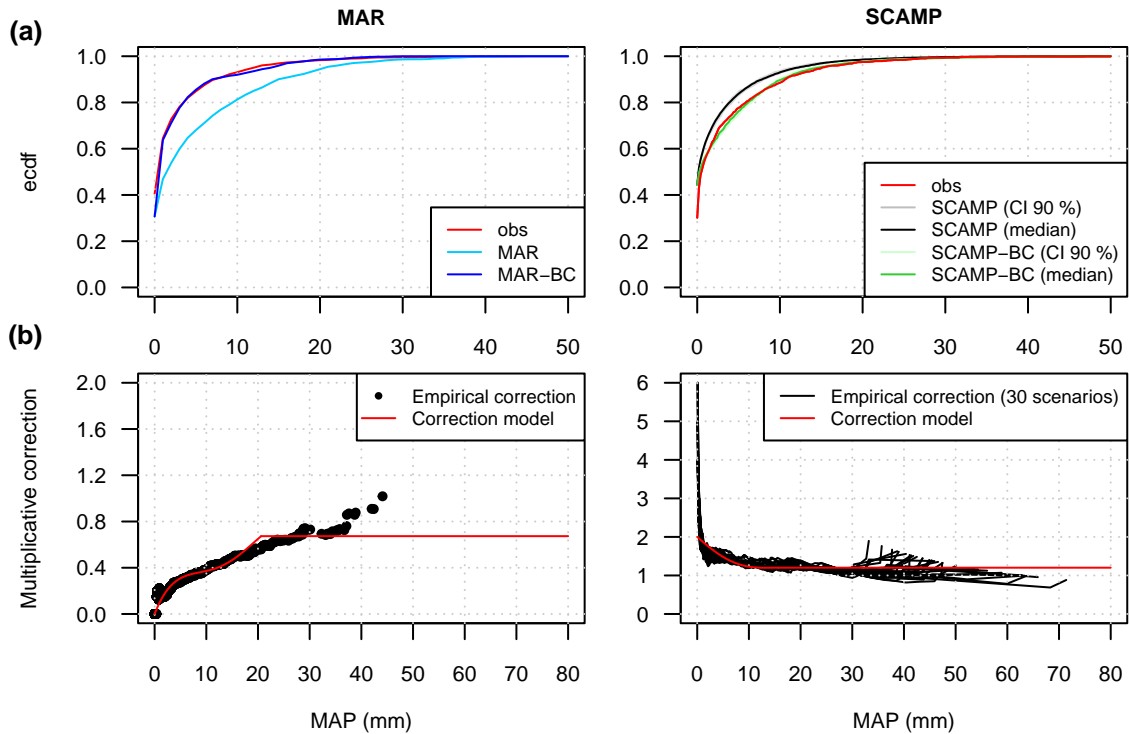

**Figure 4.** Examples of bias correction for Mean Areal Precipitation (MAP). MAR (dynamical downscaling model): sub-basins 1 to 4, January. SCAMP (statistical downscaling model): Upper Rhône River (URR) catchment, August. Control period: 1961-2009. (**a**) empirical cumulative distribution functions (ecdf). (**b**) Correction functions. For SCAMP, the grey and green bands represent the confidence intervals at 90 % level. The median scenarios are indicated by the black and green solid lines.

correction factors for extreme MAP values non simulated in the control period, the correction value was bounded. It was
310 bounded for the highest MAP percentiles (95 to 100 percentiles) to their mean empirical correction value. For MAT, the corrections for the lowest (0 to 5) and for the highest percentiles (95 to 100) were similarly set to their mean empirical correction values respectively.

For SCAMP, the MAP and MAT corrections could only be performed at the scale of the whole URR catchment. To keep the small-scale variability between the different scenarios produced by SCAMP, one single BC function was considered for the
315 30 time series. For MAR, specific BCs were applied for each of the five major sub-basins shown in Fig. 1. Examples of BC functions are shown for MAP in Fig. 4.

Note that for the MAR simulations, a BC of the temperature lapse rates was also necessary. For hydrological simulations, temperature estimates are required for each elevation band. In the present work, they are obtained by interpolation from temperature values available at neighboring locations, using a regional temperature-elevation relationship. For the MAR simulations,

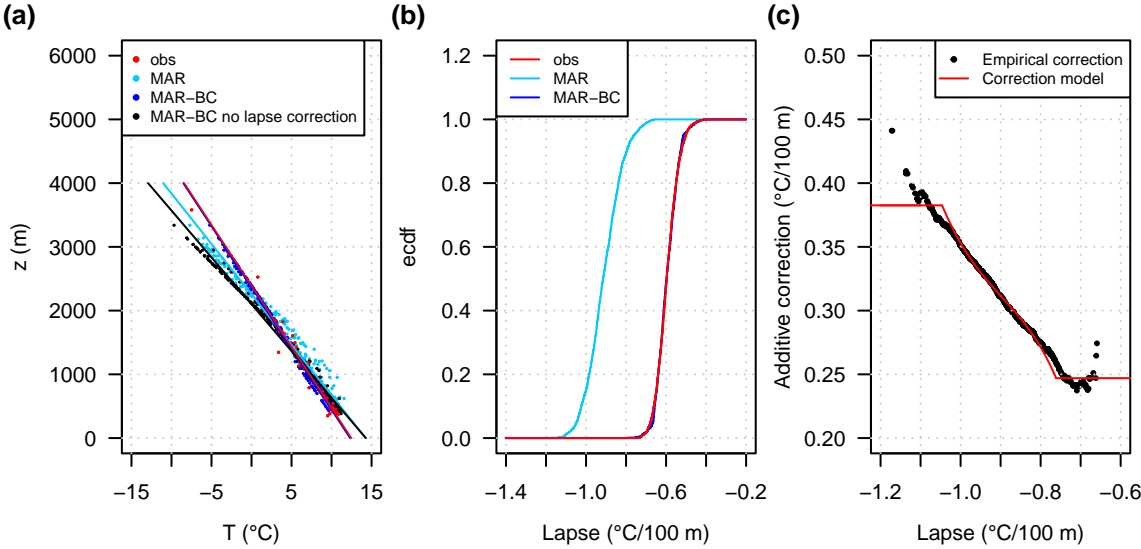

**Figure 5.** Illustration of the bias in the temperature-elevation relationship in the simulations of the dynamical downscaling model MAR. (**a**) Mean interannual relationship (and corresponding linear regressions) at the catchment-scale from observed temperatures (obs, red), raw MAR simulations (MAR, cyan), MAR simulations corrected for bias in mean temperature and lapse rate (MAR-BC, blue) and MAR simulations corrected for bias in mean temperature only (MAR-BC no lapse correction, black). (**b**) and (**c**) Examples of empirical cumulative distribution functions (ecdf) and correction model for temperature lapse rate (sub-basins 1 to 4, July). Note that the blue and red lines overlap in Fig. (**a**) and (**b**).

this relationship is assumed to be linear and its slope, the so-called "lapse rate", is estimated for each time step from simulated temperatures using all MAR grid cells in the URR catchment.

As illustrated by its statistical distribution in Fig. 5b, the lapse rate estimated from MAR simulations varies from one day to another, as does the lapse rate estimated from observations. However, it is on average higher than that of the observations, as also illustrated in Fig. 5a with the mean temperature-elevation relationships estimated from both data sets. The bias in the

lapse rate, which likely results from a warm bias in the lower atmospheric layers of the model, is quite large, of the order of 0.3 °C / 100 m (Fig. 5b).

The bias in temperature in dynamical downscaling models is recognized and corrected for a long time. To the best of our knowledge, the bias in the lapse rate is not. A common approach to use model temperatures for hydrological simulations is to identify the BC function for a given reference altitude and to use this function to correct model temperatures for all other

elevations. In this process, however, the lapse rates remain unchanged and the corrected MAT may still present residual biases for all elevations different from the reference one. This is the case for the present work. For the mountainous context considered here, as discussed in Sect. 6.2, this has important implications on the simulated hydrology.

**Table 1.** Summary of the different hydro-meteorological variables simulated and assessed in this study. MAP: Mean Areal Precipitation. MAT: Mean Areal Temperature. URR: Upper Rhône River.

| Assessment objective | Variables | Spatial resolution | Period |
|---|---|---|---|
| Year-to-year variations in weather variables | Annual MAP time series | Two major sub-basins | 1902-2009 |
| | Annual MAT time series | Two major sub-basins | 1902-2009 |
| Seasonality of weather variables | Seasonal cycle of monthly MAP | Two major sub-basins | 1961-2009 |
| | Seasonal cycle of monthly MAT | Two major sub-basins | 1961-2009 |
| Multi-scale variations in discharges | Daily discharge time series | Four gauging stations | 1981-1983 as example |
| | Mean monthly discharge time series | Four gauging stations | 1961-2009 |
| | Mean annual discharge time series | URR catchment | 1902-2009 |
| Hydrological extremes | Annual monthly discharge minima | Four gauging stations | 1961-2009 |
| | Annual daily discharge maxima | Four gauging stations | 1961-2009 |
| | Low flow activity | URR catchment | Three 30-year sub-periods |
| | Flood activity | URR catchment | Three 30-year sub-periods |

For MAR temperatures, we thus considered a two-part quantile mapping correction function that takes into account both biases, that of the MAT value simulated for a given reference elevation and that of the lapse rate. The corrected MAT value for a given elevation $z$ and a given time $t$ was obtained as follows:

$$\mathrm{MAT}_{\mathrm{MAR\text{-}BC}}\left(z,t\right) = \mathrm{MAT}_{\mathrm{MAR\text{-}BC}}\left(z_{ref},t\right) + \mathrm{lapse}_{\mathrm{MAR\text{-}BC}}(t) \times \left(z - z_{ref}\right) \tag{3}$$

where $\mathrm{MAT}_{\mathrm{MAR\text{-}BC}}\left(z_{ref},t\right)$ is the bias-corrected MAT value at the reference elevation $z_{ref}$ (the correction depends on the percentile of the MAT value at $t$) and $\mathrm{lapse}_{\mathrm{MAR\text{-}BC}}(t)$ is the bias-corrected lapse rate value at $t$ (the correction depends on the percentile of the lapse rate value estimated from MAR outputs at $t$).

In practice, we chose the mean elevation of the URR catchment (i.e. 1525 m) as reference elevation. The MAT and lapse rate corrections were carried out independently at the monthly scale and for the five major basins shown in Fig. 1. The results presented in the following for the MAR simulations were obtained with this two-part BC. The influence of the lapse rate correction is discussed in Sect. 6.2.

Note that for the SCAMP simulations, a BC of the temperature lapse rates was not necessary. In SCAMP simulations, the regional temperature-elevation relationship is estimated each day from the SCAMP temperatures at the observation stations. For each simulation day, as these temperatures are derived from the temperatures of an observed analog day, the relationship of the scenario always corresponds to an observed one. The temperature-elevation relationship and its variations over time are therefore coherent with the observations.

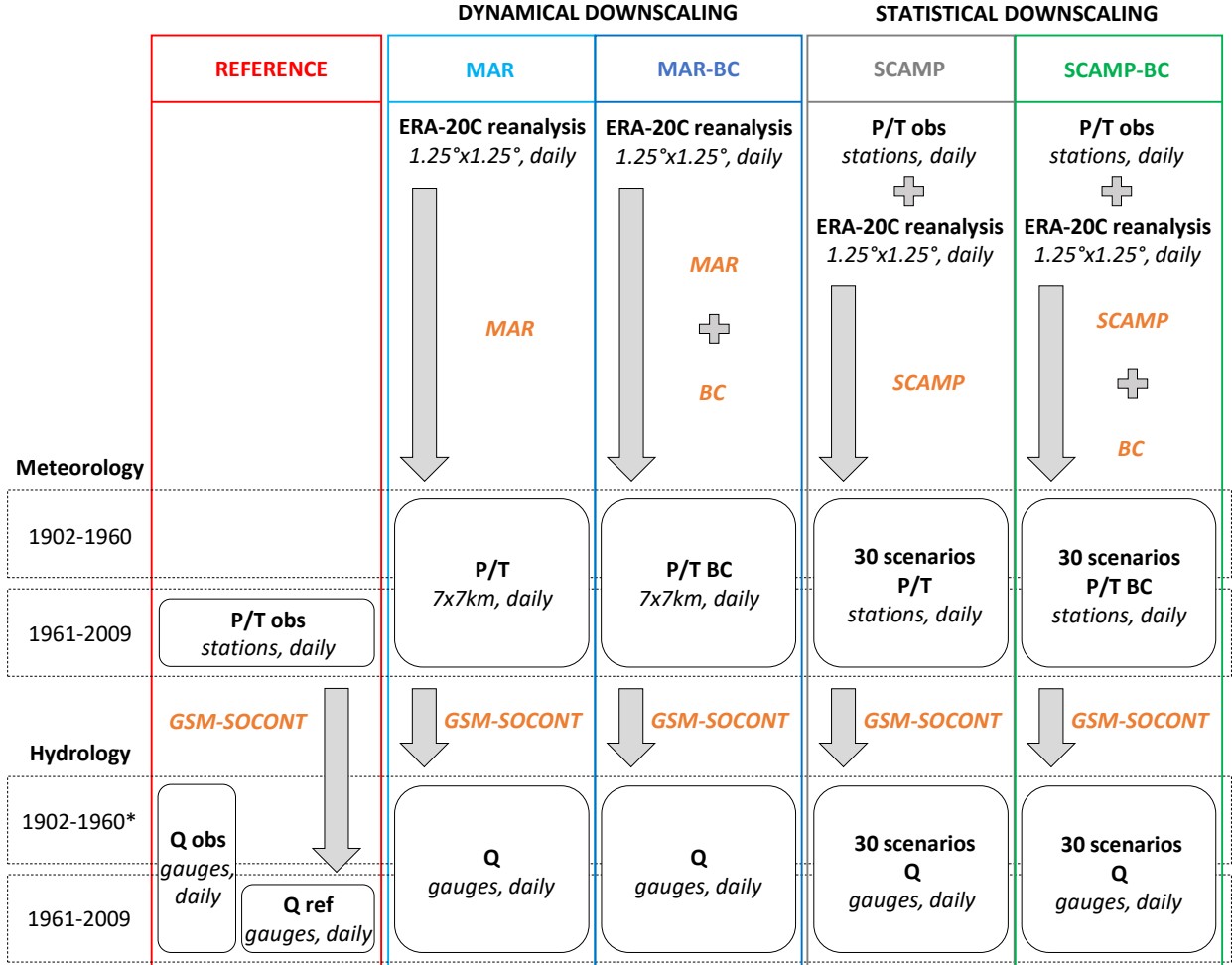

**Figure 6.** Summary of the different experiments. (**orange**) Models used: MAR (dynamical downscaling model), SCAMP (statistical downscaling model), BC (bias correction model), GSM-SOCONT (hydrological model). (**bold**) Outputs obtained after each step. (*italics*) Spatial and temporal resolutions. (\*) Time period depending on the considered gauge. P/T obs: observed precipitation/temperature. P/T: simulated precipitation/temperature. P/T BC: simulated bias-corrected precipitation/temperature. Q obs: observed discharge. Q ref: simulated discharge from observed weather variables. Q: simulated discharge.

## 4.5 Experimental setup

Four experiments are considered in the present work. They are summarized in Fig. 6. Weather scenarios are produced with either the dynamical downscaling model MAR or the statistical downscaling model SCAMP from large-scale atmospheric information (ERA-20C data). Weather scenarios obtained with each downscaling model are used to force the GSM-SOCONT model and simulate hydrological scenarios. Weather scenarios are first used with their raw values and then with their bias-corrected

values. In the following, the simulations with raw weather scenarios are referred to as "MAR" and "SCAMP" simulations.
The bias-corrected simulations are referred to as "MAR-BC" and "SCAMP-BC" simulations. Both weather and hydrological
scenarios can be compared to their counterpart references (observed or simulated as explained below). Among other things,
we will compare the ability of each hydro-meteorological modeling chain to reproduce the temporal variations in both weather
variables and discharges (Table 1).

As many sub-basins have altered hydrological regimes, the "hydrological reference" used for the comparison is the discharge
time series obtained via hydrological simulation with the observed weather variables as inputs. For some upstream sub-basins,
for which the hydrological behavior can be considered as natural, the evaluation could also rely on a comparison with discharge
observations. We however chose to use the simulated reference. This first makes the evaluation homogeneous for all URR sub-
basins. This additionally allows to focus only on the ability of the downscaling chains to simulate hydrologically relevant
weather scenarios. In other words, this allows to not distort the evaluation by intrinsic errors introduced by the hydrological
model. This point is further discussed in Sect. 6.4.

A multi-scale evaluation of simulations is carried out. We present here the meteorological evaluations carried out at the
scale of two of the five major sub-basins shown in Fig. 1 and hydrological evaluations at four illustrative gauging stations.
Rhône@Porte-du-Scex $\left(\text{gauged area of } 5{,}390\,\text{km}^2\right)$ is the outlet to Lake Geneva of the upstream part of the URR catchment.
Rhône@Genève, Halle-de-l'Ile $\left(7{,}945\,\text{km}^2\right)$ is the outlet of Lake Geneva. Arve@Genève, Bout-du-Monde is on the Arve
River before its confluence with the Rhône River $\left(1{,}990\,\text{km}^2\right)$, and Rhône@Bognes $\left(10{,}900\,\text{km}^2\right)$ is the outlet of the URR
catchment.

## 5  Results

### 5.1  Weather

Simulated weather variables are compared to observations in Fig. 7 and 8 for both downscaling models. Figure 7 shows
observed and simulated year-to-year variations in annual MAP and MAT and Fig. 8 the seasonal cycles of monthly MAP and
MAT. Results are presented for sub-basins in the upper part of the URR catchment (sub-basins 1 to 4, Fig. 7 and 8, left) and for
sub-basins downstream and around Lake Geneva (sub-basins 8 to 12, Fig. 7 and 8, right). For the statistical downscaling model
SCAMP, both figures present the dispersion between the 30 annual MAP values obtained from the 30 time series scenarios
(grey and green bands). Note that the dispersion is rather large for precipitation (e.g. up to 600 mm for annual MAP, Fig. 7),
illustrating the important uncertainty in the large-scale-small-scale relationship for this variable in this region.

For both downscaling models, simulated year-to-year variations in annual MAP and MAT are in good agreement with
observed ones, whatever the area considered (Fig. 7). The positive trend in temperature starting in 1980 is also adequately
reproduced, resulting from the global combination of the warming related to anthropogenic greenhouse gases and the reduced
anthropogenic aerosol cooling (Reid et al., 2016), especially pronounced over Europe (Nabat et al., 2014). However, the mean
simulated variables can be rather different from the observed ones. For the dynamical downscaling model MAR, for instance
(cyan lines), simulated annual MAPs are 12 % higher than observations for sub-basins around the lake and 58 % higher for

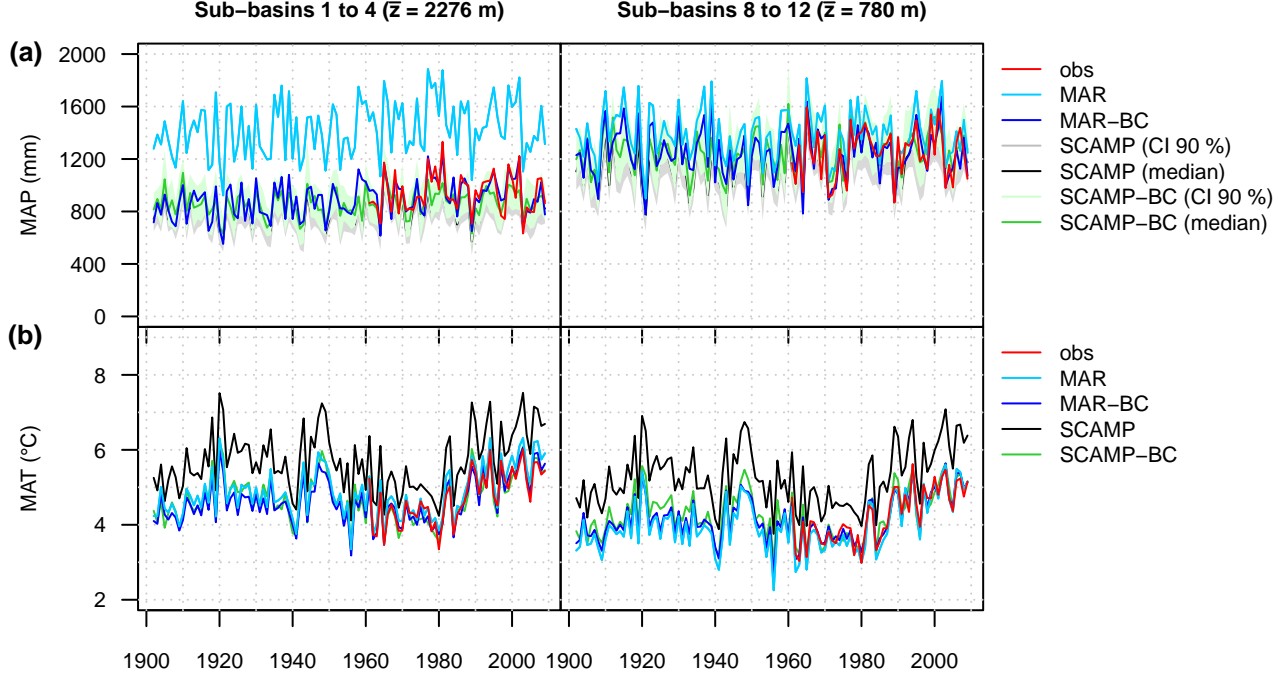

**Figure 7.** Time series of annual (**a**) Mean Areal Precipitation (MAP) and (**b**) Mean Areal Temperature (MAT) for two major basins (1902-2009). The average elevation of each major basin is indicated in brackets. For MAP, the grey and green bands represent the confidence intervals at 90 % level. The median scenarios are indicated by the black and green solid lines. For MAT, the 30 SCAMP time series scenarios are identical and correspond to the raw or to the bias-corrected time series of the ERA-20C reanalysis. obs: observed weather. MAR/MAR-BC: raw/bias-corrected weather scenario produced with the dynamical downscaling model MAR. SCAMP/SCAMP-BC: raw/bias-corrected weather scenarios produced with the statistical downscaling model SCAMP.

sub-basins in the upper part of the URR catchment. For the statistical downscaling model SCAMP (grey bands and black lines), the differences, although much smaller, are still significant. Depending on the SCAMP scenario, simulated annual MAPs are 2 % to 8 % smaller than observations around the lake and 10 % to 15 % smaller for the upper part of the URR catchment. Some

differences are also obtained for annual MATs. They are small for MAR (0.1 °C around the lake, 0.3 °C in the upper part of the URR catchment). For SCAMP, the simulations are roughly 1 °C warmer for the whole area (black lines).

As shown in Fig. 8, the deviations from observations can vary a lot from one season to another. In the upper part of the URR catchment, precipitation amounts simulated with MAR are similar to observations in summer but much larger in fall and spring (resp. +56 % and +71 %) and even larger in winter (+110 %). Around the lake, simulated precipitations are almost

similar to observations in fall (+8 %) and summer (-7 %) but are again significantly larger in winter and spring (+24 %). For SCAMP scenarios, differences are almost exclusively found in summer, when simulated MAPs are 15 % to 30 % smaller than observations for the whole area. A significant seasonality of the deviations is also found for MATs. For MAR, the difference is

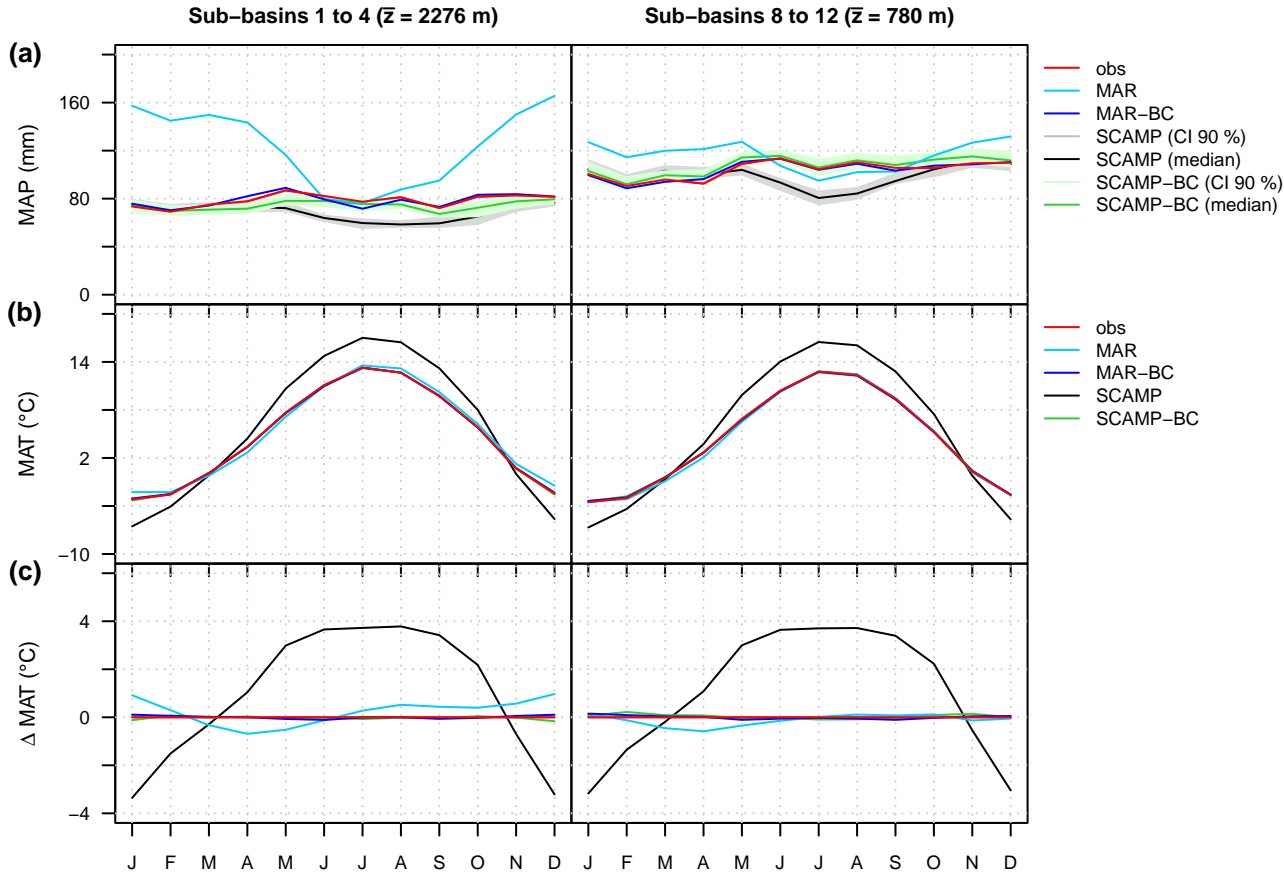

**Figure 8.** Seasonal cycles of (**a**) Mean Areal Precipitation (MAP) and (**b**, **c**) Mean Areal Temperature (MAT) for two major basins (1961-2009). The average elevation of each major basin is indicated in brackets. For MAP, the grey and green bands represent the confidence intervals at 90 % level. The median scenarios are indicated by the black and green solid lines. For MAT, the 30 SCAMP seasonal cycle scenarios are identical and correspond to the raw or to the bias-corrected seasonal cycle of the ERA-20C reanalysis. obs: observed weather. MAR/MAR-BC: raw/bias-corrected weather scenario produced with the dynamical downscaling model MAR. SCAMP/SCAMP-BC: raw/bias-corrected weather scenarios produced with the statistical downscaling model SCAMP.

-0.7 °C in spring and up to +1.1 °C in winter. For SCAMP, the seasonality is even larger: simulations are up to 3.7 °C warmer than observations in summer and, conversely, up to 2.7 °C colder in winter.

For both models, the differences mentioned above are significantly reduced and sometimes vanish completely after BC (blue lines for MAR-BC and green bands/solid lines for SCAMP-BC). This is the case for annual variables (Fig. 7) but also for monthly variables (Fig. 8) as the BC is performed on a monthly basis. For SCAMP, as the BC is performed at the catchment-scale, some biases can remain at the sub-basin scale. For annual MAPs, for instance, simulations are still 3 % to 9 % smaller

**Table 2.** Performance assessment of the modeling chains for the 1961-2009 period. For the dynamical downscaling model (MAR/MAR-BC), the NSE coefficient is calculated from the reference and the simulated discharge time series at each gauging station. For the statistical downscaling model (SCAMP/SCAMP-BC), the statistical index CRPSS (Continuous Ranked Probability Skill Score; Chardon et al., 2014) is used for the evaluation. For each gauging station, the CRPSS compares the probabilistic predictions of discharge obtained for each simulation day from i) the climatological distribution and ii) the 30 simulated time series scenarios. For a perfect model: NSE = 1 and CRPSS = 1. For a model worse than climatology: NSE < 0 and CRPSS < 0.

|  | MAR | MAR-BC | SCAMP | SCAMP-BC |
|---|---|---|---|---|
| Rhône@Porte-du-Scex | 0.84 | 0.90 | 0.10 | 0.68 |
| Rhône@Genève, Halle-de-l'Ile | 0.71 | 0.83 | 0.11 | 0.63 |
| Arve@Genève, Bout-du-Monde | 0.33 | 0.55 | 0.23 | 0.43 |
| Rhône@Bognes | 0.60 | 0.74 | 0.14 | 0.55 |

than observations in the upper part of the URR catchment, and 0 % to 6 % larger around the lake, depending on the scenario
(Fig. 7).

### 5.2 Discharge seasonality and variations

The discharges obtained via hydrological simulation from weather scenarios are compared with their reference counterparts, i.e. with the discharges obtained via hydrological simulation from observed weather variables. The comparison is applied to time series of discharges at daily, monthly and annual resolutions, and to time series of characteristic discharge variables (i.e. minimum monthly discharge observed each year and annual maximum daily discharge). Note that for SCAMP and SCAMP-BC simulations, for which 30 time series scenarios are simulated, the value of the reference discharge variable considered for a given time is compared to the 30 values obtained for that time from the 30 scenarios respectively.

At a daily time step, the discharge time series of the MAR-BC and SCAMP-BC simulations are in good agreement with the reference ones, as shown by the results obtained for the four illustrative gauging stations in Fig. 9. The agreement is even larger for time series of mean monthly discharges (Fig. 11, left). The large seasonality of flows, but also their daily and monthly temporal variations, are well reproduced, especially for the Rhône River upstream of Lake Geneva. These results are obtained for almost all gauging stations, even those located downstream of Lake Geneva, despite the significant influence of the lake regulation on flows and the rather crude regulation model used for its representation (Fig. 9e).

For both downscaling models, the BC of weather scenarios significantly improves the simulations (Fig. 9 versus Fig. 10, Table 2). The BC is required for both precipitation and temperature variables. This is especially visible in the results obtained for the upstream URR catchment. At Rhône@Porte-du-Scex (Fig. 10a), the discharge variations are rather well reproduced with raw weather scenarios but the discharges are overestimated with SCAMP and underestimated with MAR during the spring and summer. These deviations may be surprising, as raw SCAMP and MAR precipitation simulations are biased toward not enough summer precipitation in SCAMP and, conversely, much larger winter, spring and fall precipitation in MAR (Fig.

8). They actually derive from temperature scenarios that are too warm in SCAMP in summer and not warm enough in MAR. This point is further discussed in Sect. 6.2.

## 5.3 Floods and low flows

The hydrological relevance of simulated weather scenarios is further evaluated with simulations of floods and low flows. Note that the annual daily discharge maxima are used as flood proxy indicators and that the annual monthly discharge minima are used as low flows proxy indicators. Figure 11 (middle, right) presents scatter plots of simulated and reference values of annual monthly discharge minima and annual daily discharge maxima obtained for the 49 years of the 1961-2009 period. The same figure with results for the first half of the century is given in Fig. S8 in Supplementary Materials. For annual monthly discharge minima, the month with the lowest monthly flow in the reference discharge series is identified for each year of the period. The 49 lowest monthly flows are compared to their simulated counterparts for the same months. For annual daily discharge maxima, a similar comparison is made: for each year, the day with the highest reference daily flow is identified, and the corresponding discharge is compared to the maximum daily discharges obtained around that day in the simulations. The maximum discharge considered for the comparison in the simulation is identified from a 7-day window centered on the reference day.

For all gauging stations, the MAR-BC simulation leads to very satisfactory results for the annual monthly discharge minima. By contrast, the annual daily discharge maxima tend to be underestimated (at Rhône@Porte-du-Scex, Rhône@Genève, Halle-de-l'Ile and Rhône@Bognes) or poorly simulated (at Arve@Genève, Bout-du Monde). For the SCAMP-BC simulations, the annual monthly discharge minima and the annual daily discharge maxima are characterized by a very large inter-scenario variability, making the interpretation of these results more difficult. For annual monthly discharge minima, the medians of simulated scenarios are close to the reference ones at all gauging stations. For annual daily discharge maxima, the medians are also close to the reference values at Rhône@Porte-du-Scex. They are, however, off center downwards for the other three stations. The least well reproduced annual daily discharge maxima are those of the Arve basin and those at Rhône@Bognes. Conversely to the upper part of the URR catchment, the annual daily discharge maxima occur mainly in late summer and fall (see Fig. S9 in Supplementary Materials). The MAR-BC and SCAMP-BC simulations may thus fail to reproduce the large rainfall amounts in that season, especially convective events, known to generate the largest fall floods in this area.

For annual daily discharge maxima and annual monthly discharge minima, the added value of a BC for weather scenarios is again important (Fig. 11). Depending on the hydrological variable considered, the added value of a temperature BC is not necessarily equivalent to that of a precipitation BC. In the upstream parts of the URR catchment, for instance, low flows occur mainly in winter, due to the low temperatures in this season. The quality of simulated winter low flows thus depends to a great extent on the quality of temperature scenarios, and much less on the quality of precipitation scenarios. Conversely to precipitation corrections, temperatures corrections therefore lead to a significant improvement in low flow simulations. This is illustrated by the additional analyses presented in the discussion (Sect. 6.2).

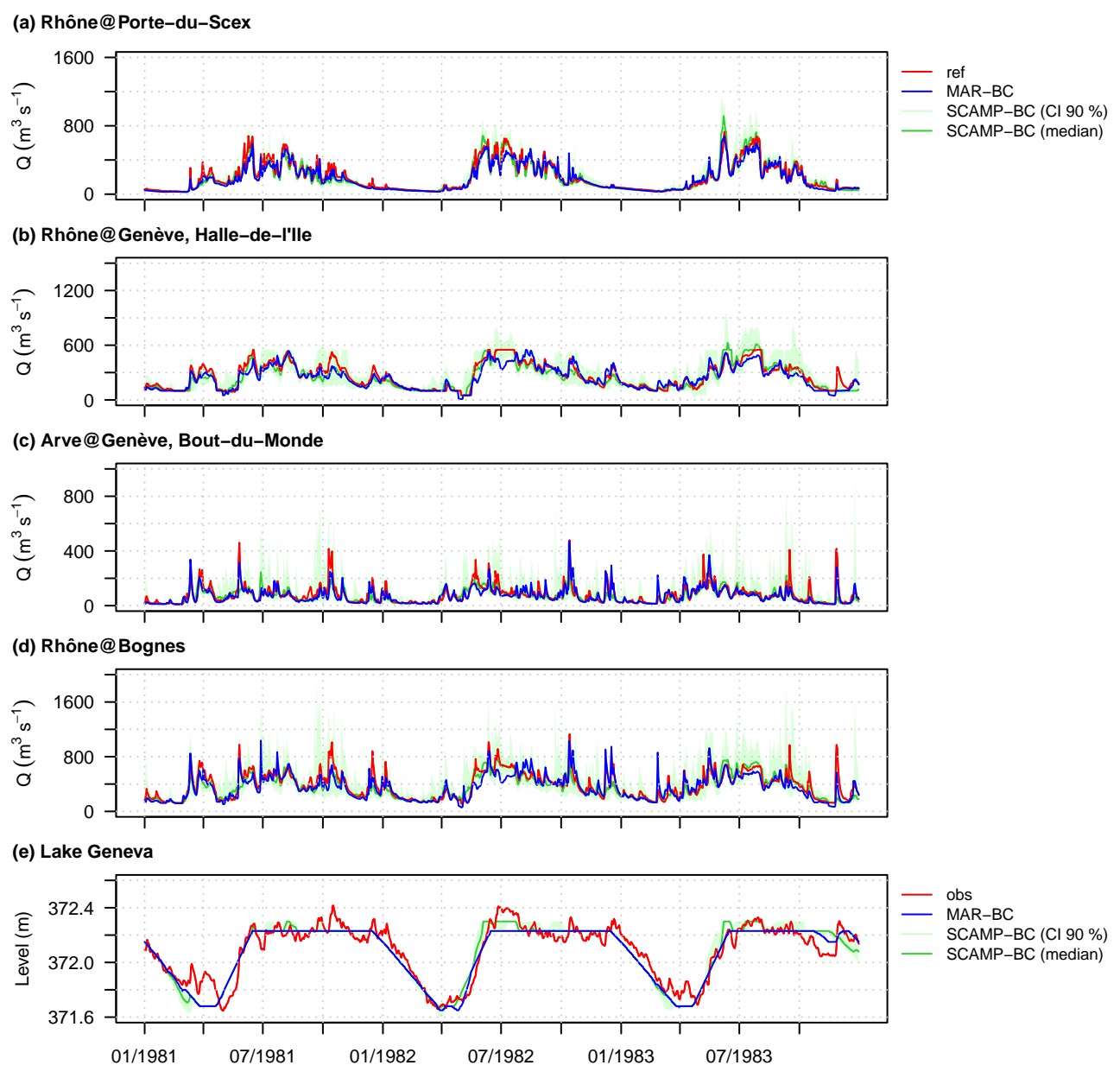

**Figure 9.** Time series of daily discharges at (**a**) Rhône@Porte-du-Scex, (**b**) Rhône@Genève, Halle-de-l'Ile, (**c**) Arve@Genève, Bout-du-Monde, and (**d**) Rhône@Bognes for the 1981-1983 period. (**e**) Time series of daily level of Lake Geneva for the same period. The green bands represent the confidence intervals at 90 % level. The median scenarios are indicated by the green solid lines. ref: simulated discharge from observed weather variables. MAR-BC: simulated discharge from the bias-corrected weather scenario produced with the dynamical downscaling model MAR. SCAMP-BC: simulated discharge from the bias-corrected weather scenarios produced with the statistical downscaling model SCAMP. obs: observed level of Lake Geneva.

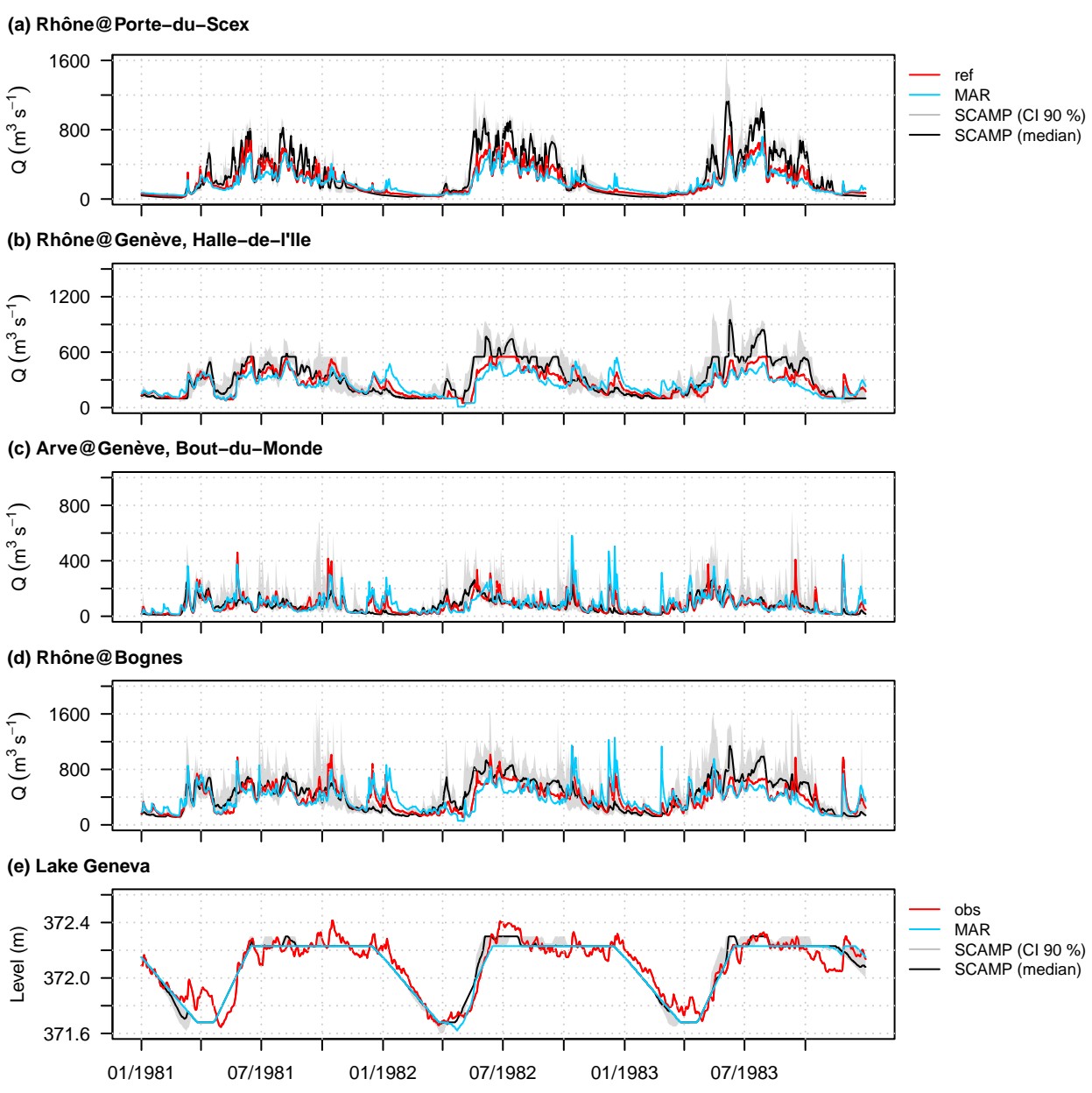

**Figure 10.** Time series of daily discharges at (**a**) Rhône@Porte-du-Scex, (**b**) Rhône@Genève, Halle-de-l'Ile, (**c**) Arve@Genève, Bout-du-Monde, and (**d**) Rhône@Bognes for the 1981-1983 period. (**e**) Time series of daily level of Lake Geneva for the same period. The grey bands represent the confidence intervals at 90 % level. The median scenarios are indicated by the black solid lines. ref: simulated discharge from observed weather variables. MAR: simulated discharge from the raw weather scenario produced with the dynamical downscaling model MAR. SCAMP: simulated discharge from the raw weather scenarios produced with the statistical downscaling model SCAMP. obs: observed level of Lake Geneva.

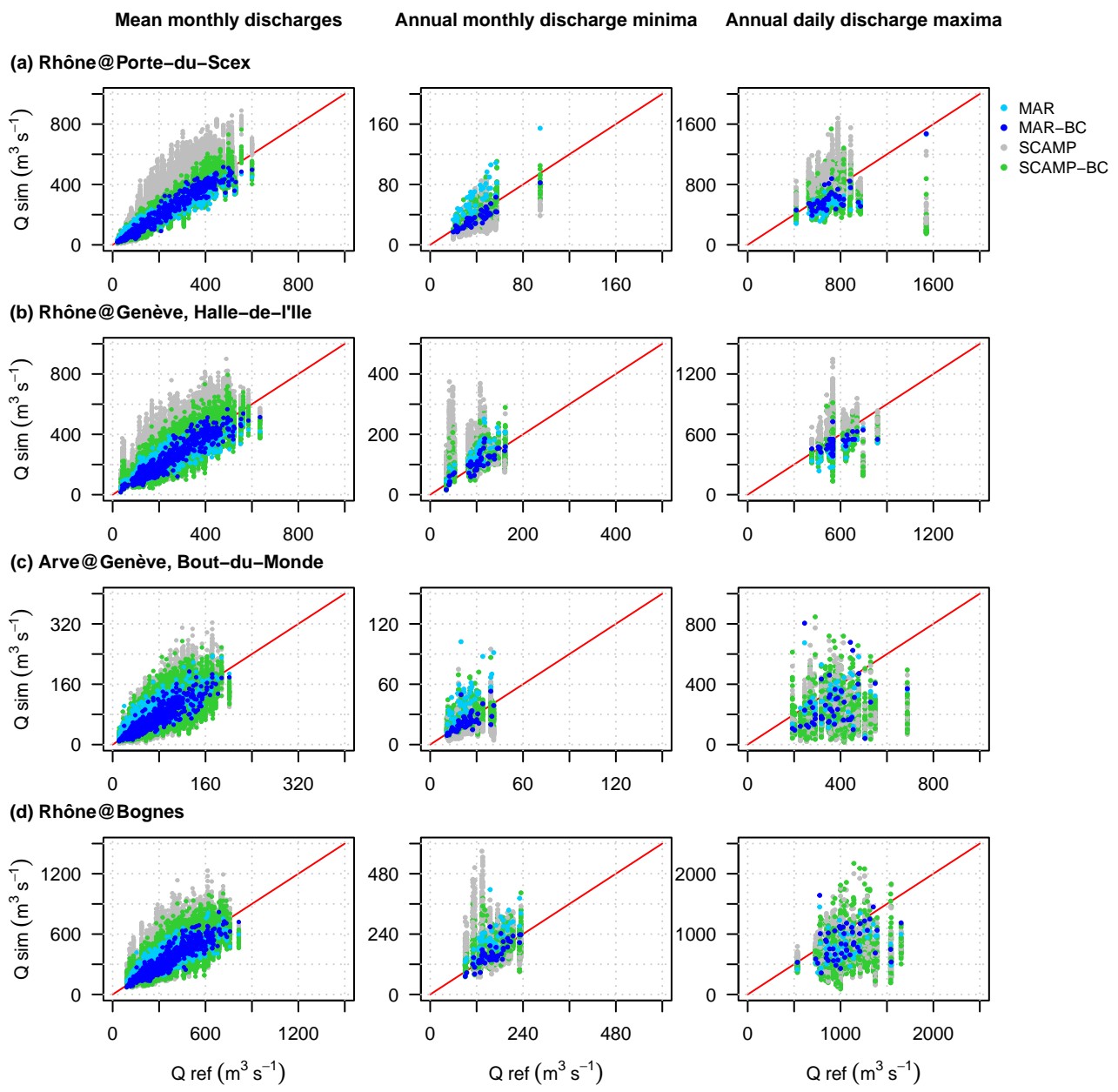

**Figure 11.** Scatter plots of mean monthly discharges, annual monthly discharge minima and annual daily discharge maxima at (**a**) Rhône@Porte-du-Scex, (**b**) Rhône@Genève, Halle-de-l'Ile, (**c**) Arve@Genève, Bout-du-Monde, and (**d**) Rhône@Bognes for the 1961-2009 period. Q ref: simulated discharge from observed weather variables. MAR/MAR-BC: simulated discharge from the raw/bias-corrected weather scenario produced with the dynamical downscaling model MAR. SCAMP/SCAMP-BC: simulated discharge from the raw/bias-corrected weather scenarios produced with the statistical downscaling model SCAMP.

## 5.4 Mean annual discharges, flood and low flow activities

The modeling chains considered previously are commonly used for hydroclimatic projections with large-scale atmospheric GCM outputs as forcing variables. The potential impact of climate change on hydrology is often assessed considering changes in the statistical characteristics of hydrological regimes (e.g. changes in seasonality and year-to-year variability, changes in flood and low flow activities). We therefore also assess the ability of the considered modeling chains to simulate the "reference" variations in different characteristics of the URR hydrological regime over the last century (1920-2009). In this section, as no better reference time series is available, the "references" are observed discharges for the 1920-1960 period and simulated discharges from observed weather variables for the 1961-2009 period. The results are therefore to be interpreted with caution.

The simulated year-to-year variations in mean annual discharges are first compared to the "reference" ones over the 1920-2009 period (Fig. 12 c, d). Due to the large interannual variability of discharges, it would be rather difficult to assess the ability of the modeling chains to catch any long-term trends in this variable. However, the year-to-year variations are well reproduced in both timing and amplitude, especially for the second half of the century. Recall especially that the hydrological model does not alter the comparison during this sub-period, as the two time series compared are obtained by hydrological simulation.

We then consider the variations in flood and low flow activities of the URR catchment. Flood activity is defined here as the average number of daily discharges per 30-year period above a given daily discharge threshold. Low flow activity is similarly defined as the average number of months per 30-year period during which the mean monthly discharge is below a given discharge threshold. The thresholds retained are the reference discharge values exceeded on average once a year over the entire 90-year simulation period (1920-2009). For flood activity, the threshold was calculated by considering the 90 largest daily discharges over the 90-year period. If the threshold was exceeded several times during five consecutive days, only the date of the maximum discharge was retained. For low flow activity, the threshold corresponds to the 10th percentile, i.e. the 90 lowest values of mean monthly reference discharges over the 90-year period. At Rhône@Bognes, the thresholds are respectively 962 $m^3$ $s^{-1}$ for floods and 167 $m^3$ $s^{-1}$ for low flows. The flood and low flow activities, estimated for each of the three 30-year sub-periods 1920-1949, 1950-1979 and 1980-2009, are presented in Fig. 12 (a, b).

In both BC simulations, the number of flood events exceeding the threshold and the variations in flood activity from one sub-period to another are in good agreement with the reference ones. The observed increase in flood activity is rather well reproduced (Fig. 12a). The results for low flow activity are less satisfactory (Fig. 12b). On the one hand, the number of mean monthly discharges below the threshold is, whatever the sub-period, significantly higher than that of the reference. It is more than three times higher for the 1920-1949 sub-period. This overestimation is mainly due to longer winter low flow durations (not shown). On the other hand, the variations in low flow activity from one sub-period to another are only partially reproduced. While the small decrease observed between the last two sub-periods is rather well captured, the large increase between the first two sub-periods is fully missed. The reasons for this are unclear. The large differences obtained with the raw and bias-corrected downscaling simulations suggest that some limitations remain in the bias-corrected weather scenarios (e.g. too low simulated winter temperatures). The large differences in the reference low flow activity obtained between the sub-periods also suggest an issue with the stationarity assumption (e.g. hydrological signatures) and/or with the low temporal homogeneity of the

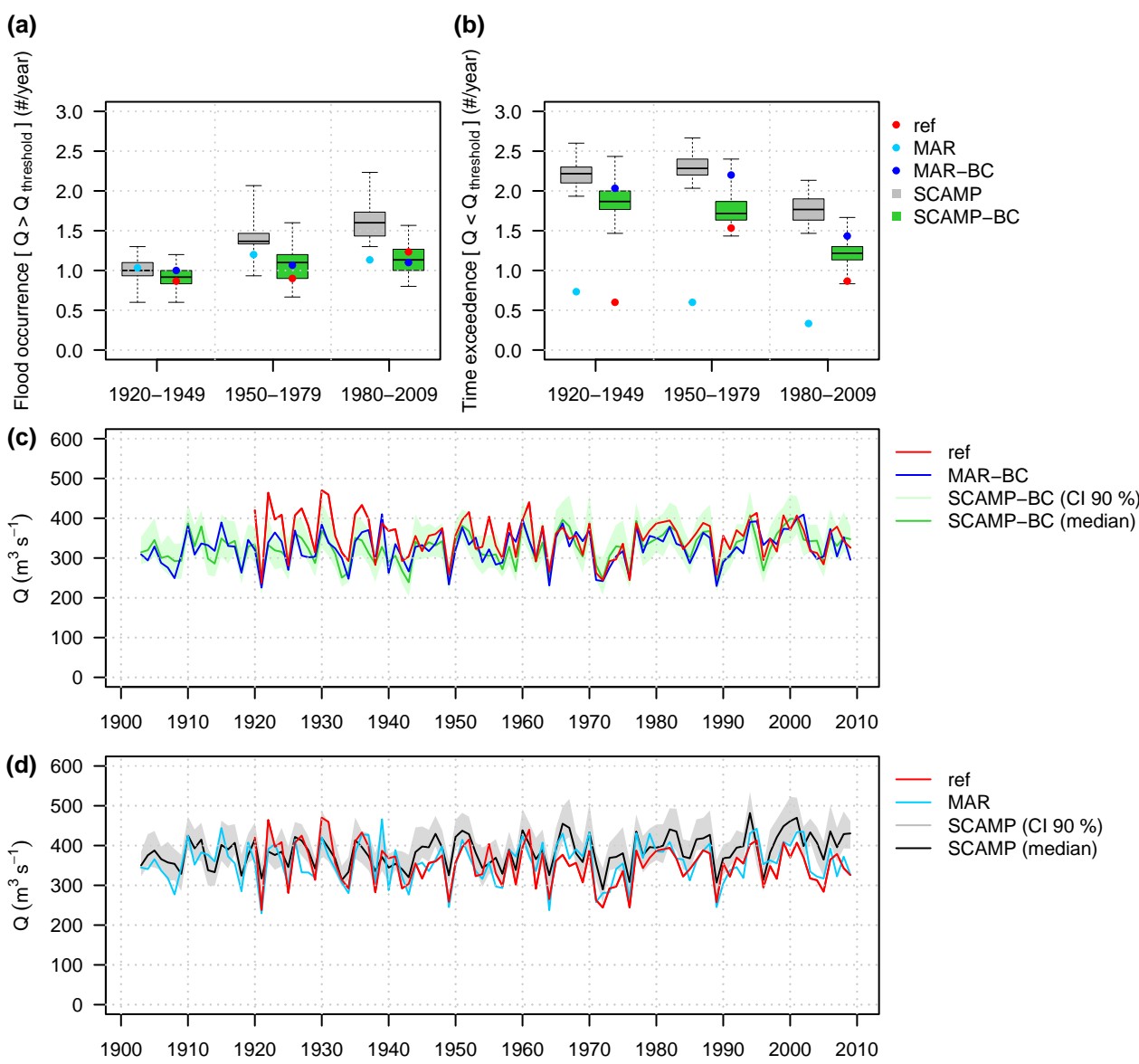

**Figure 12.** (**a**) Flood activity and (**b**) low flow activity at Rhône@Bognes for three 30-year sub-periods: 1920-1949, 1950-1979 and 1980-2009. See text for definitions of flood activity and low flow activity. (**c**) Mean annual discharge time series at Rhône@Bognes for the 1902-2009 period simulated with MAR-BC and SCAMP-BC. (**d**) The same for simulations with MAR and SCAMP. The grey and green bands represent the confidence intervals at 90 % level. The median scenarios are indicated by the black and green solid lines. As no better reference time series is available, note that the "references" are observed discharges for the 1920-1960 period and simulated discharges from observed weather variables for the 1961-2009 period. MAR/MAR-BC: simulated discharge from the raw/bias-corrected weather scenario produced with the dynamical downscaling model MAR. SCAMP/SCAMP-BC: simulated discharge from the raw/bias-corrected weather scenarios produced with the statistical downscaling model SCAMP.

data set considered in this work (e.g. the reference discharge time series made from observed/simulated discharges for the beginning/end of the period). These points will be worth further investigations in future works.

## 6  Discussion

All in all, and as already illustrated in a number of previous works (e.g. Boé et al., 2007; Kuentz et al., 2015; Bonnet et al., 2017; Caillouet et al., 2017; Weber et al., 2021), hydrologically relevant weather scenarios (or reconstructions) can be achieved
with either statistical or dynamical downscaling models from large-scale atmospheric information only. As also illustrated here, this may require some preliminary BCs to atmospheric model outputs.

As discussed in the following, the need for corrections can be attributed to some limitations of the models. It may be also attributed to the quality of the available "observations". We will discuss issues related to reanalysis data and to lapse rates for both temperature and precipitation. We will also discuss some issues related to the hydrological model considered in the
modeling chains.

### 6.1  Reanalysis data

The global atmospheric reanalysis ERA-20C considered in this study is used as pseudo-observation of the state and dynamics of the atmosphere over a large spatial area covering the European domain. As it is produced by assimilating only sea level pressure and wind measurements, ERA-20C data are not, however, free of limitations. The quality of the geopotential at 500
505  hPa and of other large-scale variables (such as vertical velocities at 600 hPa, temperature at 2 m, large-scale precipitation) may be rather low and may impact the skill of both downscaling models.

This is the case, for instance, for the regional MAT time series used to force the SCAMP time series scenarios. The large bias in the regional MAT SCAMP scenarios highlighted in Fig. 7 and 8 directly derives from the bias in the regional MAT of the ERA-20C reanalysis over the considered domain. Similar limitations were reported by Bonnet et al. (2017), who had to
correct the biases of a downscaled version of the ERA-20C reanalysis to simulate realistic mean river flows in France.

As shown by Horton and Brönnimann (2019), using a reanalysis assimilating more data, such as the recent ERA5 reanalysis (Hersbach et al., 2020), could lead to more relevant weather scenarios. Such reanalyses were not used in the present work, as they generally cover a much shorter period (around 60 years) preventing the simulation and evaluation of hydro-meteorological scenarios over a century.

### 6.2  Temperature lapse rate

As shown in Sect. 4.4 (Fig. 5a), the lapse rates estimated from MAR simulations are on average higher than those estimated from observations. For the mountainous context considered here, this bias has important implications on the simulated hydrology. A higher lapse rate leads to lower temperatures than those observed for high elevation bands in particular (where few observations are available) and vice versa for low elevation bands. This logically makes the simulated snowpack dynamics
significantly different from the observed one. Lower temperatures lead to more frequent solid precipitation, more snow accu-

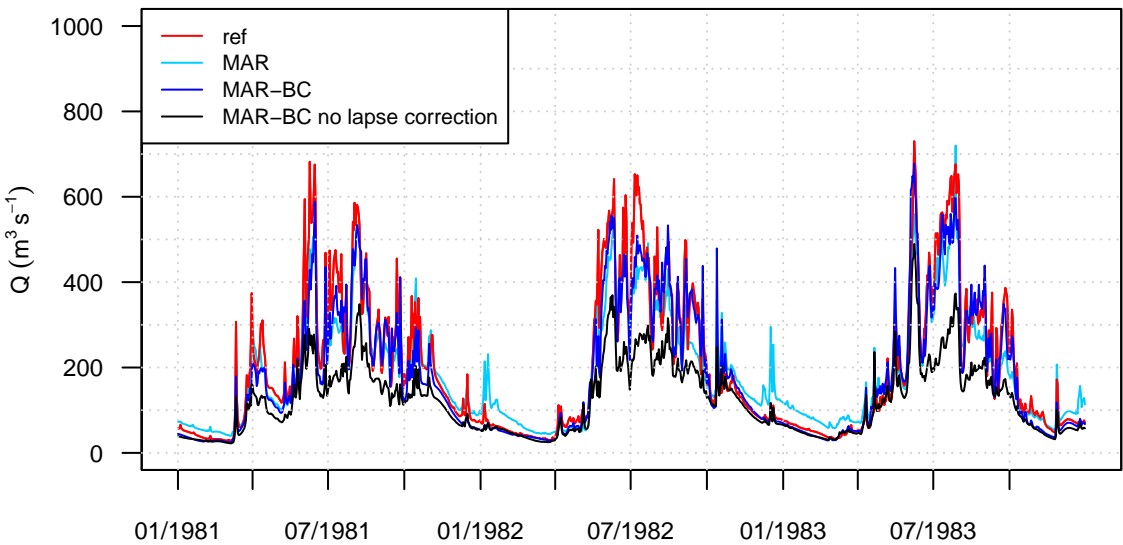

**Figure 13.** Influence on simulated hydrology of the bias in temperature lapse rate for the dynamical downscaling model MAR. Illustration with the time series of daily discharges at Rhône@Porte-du-Scex for the 1981-1983 period. ref: simulated discharge from observed weather variables. MAR: simulated discharge from the raw weather scenario. MAR-BC: simulated discharge from the bias-corrected weather scenario considering both the bias correction of the Mean Areal Temperature (MAT) for a given reference elevation and the bias correction of the temperature lapse rate. MAR-BC no lapse correction: simulated discharge from the bias-corrected weather scenario considering only the bias correction of the MAT for a given reference elevation.

mulation, less snowmelt in spring, snow cover for longer durations and over larger areas, etc. For the highest elevation bands, lower temperatures can even lead to a simulation of a perennial snowpack, preventing any simulation of ice melt (which can only occur in the model for elevation bands where the glacier is free of snow). All of this results in poorly simulated hydrological regimes. As already suggested in Sect. 5.4, a biased lapse rate is, for instance, expected to give a poor simulation of

525 winter low flow characteristics. A higher lapse rate is also expected to lead to delayed snowmelt floods and to low flows in late summer that are not sustained as they should be by ice melt.

If less evident, a biased lapse rate may also give a poor simulation of flood regimes. In the URR catchment, the largest floods often occur in fall due to large precipitation amounts. In fall, the temperatures can be low enough for precipitation to fall as snow in high elevation areas. Such situations lead to "reduced floods" compared to floods that would have occurred if all the

530 precipitation had fallen as rain. This situation was observed, for instance, during the flood of October 15, 2000, making the flood damage in the region much smaller than expected (Hingray et al., 2010). The temperature lapse rate is determinant, as it defines the elevation of the snowfall/rainfall limit and therefore the "effective area" of the catchment for these events. The overestimated lapse rate in the MAR model, which results in too cold weather in high elevations areas, is thus expected to lead to lower intensity floods in fall. This is illustrated by the differences between the two bias-corrected experiments produced with

535 MAR in Fig. 13. When MAR is not corrected for the lapse rate, the intensity of fall floods is significantly lower than when it is.

The added value of the temperature lapse rate correction for hydrological simulations is thus clearly significant, due to the direct effects of temperature on snow dynamics (snow/rain repartition, snowpack evolution). To the best of our knowledge, the issue of the temperature lapse rate has not received much attention in the past, but it should probably receive more, at least in

areas covering large elevation ranges and where highly non-linear behaviors with respect to temperature have to be simulated. These results should also lead scientists to integrate this issue when evaluating dynamical downscaling models, and to consider appropriate BC approaches before using model outputs to force impact models.

## 6.3 Orographic precipitation enhancement

A similar, but different, issue arises for precipitation. As mentioned in Sect. 5.1, significant differences are obtained between

545 MAPs estimated from downscaled simulations and from observations. This translates directly into significant differences in the simulated hydrology. However, the development of relevant MAP estimates is still a challenge in hydro-meteorology, particularly in mountainous areas (e.g. Ruelland, 2020).

In the present study, the MAPs estimated for the reference hydrological simulation are obtained from station observations using the Thiessen's weighting method. For the reasons mentioned in the following, no precipitation-elevation relationship

was considered in this estimation. However, the annual precipitation generally increases with elevation in the region. This dependency is rather clear from observations. It is also found in the MAR simulations, although the simulated precipitation lapse rates may overestimate the true ones (Ménégoz et al., 2020).

The "no precipitation lapse rate" assumption retained for our simulations is therefore not really valid. To illustrate the influence of this assumption, we carried out auxiliary "reference" simulations using a constant but elevation-dependant adjustment

factor for precipitation (e.g. Viviroli et al., 2022). In practice, all precipitation data of a given station are multiplied by a constant value, depending on the difference between the elevation of the station and that of the target hydrological unit, before application of the Thiessen's weighting process. Two "with precipitation lapse rate" experiments were performed. The adjustment factors were obtained assuming a linear increase of 5 % and 10 % respectively per 100 m of elevation.

As shown in Fig. 14, accounting for a precipitation lapse rate has contrasting effects depending on the area considered. It

significantly increases the annual MAP estimates for high elevation sub-basins (Fig. 14a, left), but has almost no influence on MAP estimates for low elevation sub-basins (Fig. 14a, right). This reflects the under-representation of high elevation stations in the region. All in all, accounting for a precipitation lapse rate significantly reduces the differences between annual amounts from observations and MAR simulations.

For hydrological simulations, however, the precipitation-elevation dependency is not trivial to take into account in a relevant

way. As shown by Ruelland (2020), while the orographic enhancement can be clearly identified from annual and seasonal means, it is no more evident at the event-scale, and for any given event, the spatial pattern of precipitation generally depends on where the precipitation event first occurs. This is likely the reason why the precipitation lapse rate is significantly lower in summer than in winter in the region (Fig. 14b), due to much more frequent convective events (e.g. Ménégoz et al., 2020).

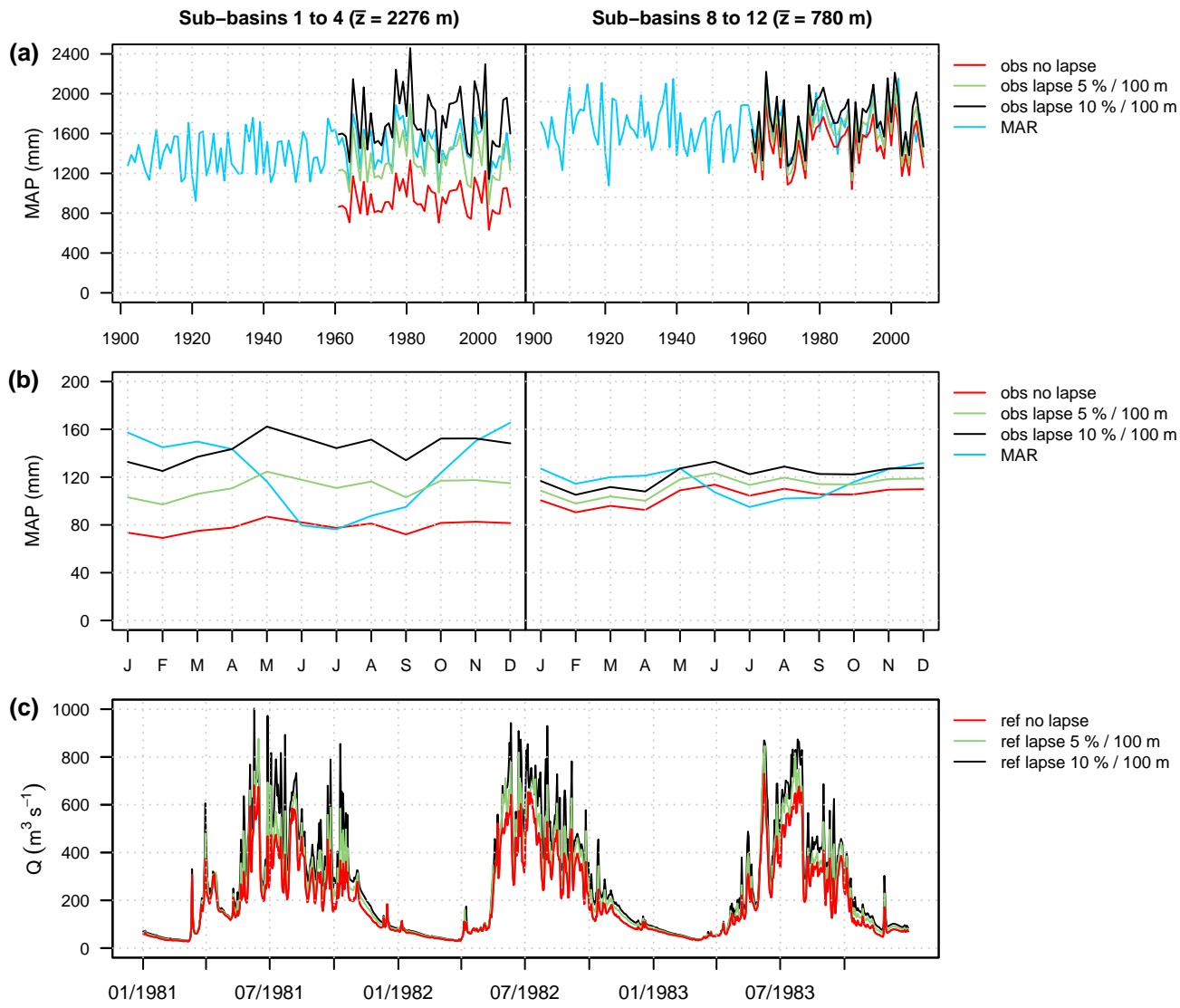

**Figure 14.** (a) Time series of annual Mean Areal Precipitation (MAP) (1902-2009) and (b) seasonal cycles of MAP (1961-2009) for two major basins. The average elevation of each major basin is indicated in brackets. obs: observed MAP considering different precipitation lapse rates. MAR: raw MAP scenario produced with the dynamical downscaling model MAR. (c) Time series of daily discharges at Rhône@Porte-du-Scex for the 1981-1983 period. ref: simulated discharge from observed weather variables considering different precipitation lapse rates.

According to Bárdossy and Pegram (2013), on a daily scale, the orographic effects generally contribute a small part to pre-
cipitation variations. In many cases, the orographic enhancement obtained for long accumulation durations is often related to

more frequent precipitation in high elevations rather than more intense precipitation. Although not verifiable from available observations, this is probably also the case in the region.

Although often determinant, orographic effects cannot be taken into account by a simple and constant adjustment factor, but for practical reasons and lack of better knowledge, this is the usual practice in hydrological modeling. Using a constant adjustment factor for all time steps of a given period is, however, likely to have significant implications on hydrological simulations. This may amplify the largest precipitation events in an unrealistic way, resulting in turn to huge and unrealistic floods (Hingray et al., 2010). For the URR catchment, hydrological simulations with a precipitation lapse rate produce significantly larger floods (Fig. 14c). With a lapse rate of 5 % / 100 m, the annual daily discharge maxima of the 1961-2009 period are increased by 3 to 106 % (+25 % on average). With a lapse rate of 10 % / 100 m, the increase is even larger (+56 % on average). It is up to 200 % for the largest flood ever observed in the area (October 15, 2000) (not shown).

The precipitation-elevation dependency was disregarded in the present work, to avoid such unrealistic simulations. This is not really satisfactory, especially regarding the annual water budget of sub-basins. The latter will be mis-represented as a result of the overall dry bias in the precipitation input. Without a better knowledge of how precipitation amounts depend on elevation at the event-scale, disregarding the elevation dependency was found to be an acceptable compromise option if relevant flood events have to be simulated.

Note that in our simulations, the dry bias is probably compensated during hydrological parameter optimization. Parameter optimization generally has the effect of forcing the model to close the water balance. Reservoir-based conceptual models, such as the one used here, can thus compensate for missing precipitation by reducing the evapotranspiration losses. This problem is well known, but rarely discussed. An example is the work of Minville et al. (2014).

## 6.4 The hydrological model: a powerful assessment tool despite its limitations

In this work, we assessed and compared the ability of two downscaling models to simulate hydrologically relevant weather scenarios. The hydrological model considered here for this assessment is obviously not free of limitations.

In the model, as mentioned in the previous section, the influence of orography on precipitation was disregarded to avoid the generation of irrelevant floods. This representation is obviously far from satisfactory, and more relevant representations, at the event-scale especially, would be worth considering in future years, when available.

The mean areal precipitation and temperature were estimated for each spatial unit of the model using the Thiessen's weighting method. This is a rather crude method, and other methods may provide better estimates (e.g. inverse distance weighting or kriging with external drift; Wagner et al., 2012). However, they have also important limitations in mountainous environments (e.g. difficulty to account for the influence of topography on weather spatial patterns at the event-scale). The Thiessen's weighting method was here retained for its simplicity and its ability to take into account the time-varying temperature-elevation relationship in a straightforward way.

The hydrological model relies also on assumptions, potentially crude. For instance, the characteristics of the sub-basins are assumed not to have changed over the last century. For the glacier cover, this assumption does not really hold. According to Huss (2011), however, the contribution of glacier melt in the region was relatively stable over the 20th century, with a similar

glacier contribution during the periods 1961-1990 and 1908-2008. The glacier retreat strengthened over the period 1988-2008, but the corresponding increase in the glacier contribution to the URR discharges was found to be rather limited (13% in August). While not fully satisfactory, the assumption of a constant glacier therefore seems reasonable.

The signature-based calibration of the model, used for sub-basins with altered discharge data, is also not optimal. The objective function considered for the calibration, for instance, gives considerable weight to the statistical distribution of annual daily discharge maxima. Other results may be obtained adding criteria for low flows (e.g. distribution of annual monthly discharge minima). This will be worth further investigations. On the other hand, parameters were calibrated so that simulated signatures reproduce at best observed ones, but observed and simulated signatures come from different periods. We thus implicitly assume that the weather regimes and the natural hydrological behavior of the URR sub-basins have not changed significantly over the last century. Both assumptions seem to be reasonable in first approximation. Indeed, over the last century, no significant trends in precipitation have been observed in the region (Masson and Frei, 2016) and the hydrological regimes of the natural sub-basins have remained almost unchanged (see Fig. S10 in Supplementary Materials).

The hydrological model is thus not optimal for a number of reasons. However, this is not expected to influence the main results of our study. Indeed, the model is mainly used here as a complex and non-linear filter to assess the ability of the downscaling chains to simulate hydrologically relevant weather scenarios. In this impact-oriented assessment context, the hydrological model can be imperfect, as it can also be used to produce the hydrological reference against which the hydrological scenarios will be compared. In the context of the URR catchment, the model is a powerful tool to achieve an impact-oriented assessment of downscaled weather scenarios in contrasting and demanding hydro-meteorological configurations, where the interplay between weather variables, both in space and time, is determinant.

## 7  Conclusion

In this study, two hydro-meteorological modeling chains were used to simulate the past variations in discharge at several stations of the Upper Rhône River catchment, a mesoscale catchment in the western Alps. The discharges were simulated with the glacio-hydrological GSM-SOCONT model using weather scenarios downscaled with the MAR and SCAMP models from the data of the global atmospheric reanalysis ERA-20C. MAR is a dynamical downscaling model, SCAMP a statistical one providing an ensemble of downscaled scenarios.

The originality of this study is fourfold. i) We evaluated the modeling chains in contrasting and demanding hydro-meteorological configurations where the interplay between weather variables, both in space and time, is determinant. ii) The spatio-temporal relevance of the weather scenarios is assessed by their hydrological responses, simulated using an ad hoc hydrological model at several gauging stations. iii) The simulations cover the entire 20th century, a period long enough to assess the ability of the modeling chains to reproduce daily variations in observed discharges, low frequency events, and variations in low flow and flood activities. iv) For both downscaling models, we evaluated the need for additional bias correction of the weather scenarios, including that of temperature lapse rates.

This framework allowed to highlight important criteria to be met for the simulation of relevant hydrological scenarios for the Upper Rhône River catchment. The alpine configuration of the Upper Rhône River catchment (unknown effects of the large upstream dams and of the regulation of Lake Geneva, scarcity of concomitant weather/discharges observations) made the calibration of the hydrological model rather difficult. This required the development of an original multiple calibration strategy, based on both observed discharge time series and hydrological signatures.

For both modeling chains, given this difficult modeling context and the fact that the weather scenarios are only produced from large-scale atmospheric information, the simulated discharges are globally in good agreement with the reference ones. For the 1961-2009 period, the multi-scale variations in reference discharges (daily, seasonal and interannual) are well reproduced. To some extent, the simulations also reproduce the annual monthly discharge minima and the annual daily discharge maxima quite well. For the first half of the century, the agreement with the reference discharges is lower (but still reasonable), likely due to lower data quality (ERA-20C and discharges data) and/or to some modeling assumptions and choices (e.g. signature-based calibration, stationarity assumption). Nevertheless, both modeling chains are able to accurately reproduce the variations in flood activity over the last century. The results for low flow activity are less satisfactory.

Both modeling chains are likely to be appropriate for the generation of relevant regional weather scenarios for different climate contexts, from outputs of ad hoc GCM experiments. Thanks to its much lower computational cost, the SCAMP model is to be favored when large ensembles of climate simulations have to be downscaled. The statistical nature of SCAMP also allows to account for the uncertainty in the downscaling relationship. The ensemble of weather scenarios generated by SCAMP for any large-scale scenario allows thus to simulate and account for the small-scale internal variability of weather. This is another advantage over MAR, allowing a more robust assessment of possible low-frequency changes in hydro-meteorological regimes (e.g. Lafaysse et al., 2014). In a future study, we will force the SCAMP/GSM-SOCONT chain with a CMIP6-PMIP4 paleosimulation ensemble (Jungclaus et al., 2017; Kageyama et al., 2018) to assess the variations in hydro-meteorological regimes of the Upper Rhône River catchment over the last millenium. We also expect to confront the simulated variations in flood activity with those obtained in previous works from the sediments archives of Lake Bourget (Jenny et al., 2014; Evin et al., 2019; Wilhelm et al., 2022). Moreover, this could also corroborate the influence of warming and atmospheric circulation changes on multi-decadal flood activity over the last two centuries, recently highlighted by Brönnimann et al. (2022) for different European rivers.

As already shown in previous works, the hydrological behavior of river basins can be simulated from large-scale atmospheric information only. In this study, we also showed that the simulation of hydrologically relevant weather scenarios required a bias correction of the downscaled weather scenarios. The highly non-linear behavior of hydrological systems does not bear biased weather. This was made evident here for temperature, due to the highly non-linear thermal sensitivity of snow variables. If it seeks to be relevant, the bias correction step is not necessarily straightforward (Maraun, 2016; Switanek et al., 2022). For temperature, the bias correction is commonly applied for the temperature of reference stations. For the Upper Rhône River catchment, the bias correction was also needed for the temperature lapse rate simulated by the dynamical downscaling model MAR. The temperature lapse rate correction was determinant to avoid irrelevant simulations of the snowpack dynamics at high elevations, and consequently of the hydrology.

The bias correction was also required for precipitation. Significant differences were also found between the reference and the downscaled precipitation, particularly for MAR in winter. The bias correction of downscaled precipitation was thus also applied so that the statistical distribution of precipitation scenarios fits that of reference. But the quality of the reference is questionable. It was indeed developed disregarding the significant precipitation-elevation relationship in the area, to avoid the simulation of irrelevant flood events. The much larger precipitation amounts simulated by MAR for high elevation sub-basins are thus not necessarily irrelevant, considering the snowfall undercatch issues (Kochendorfer et al., 2017). An interesting perspective of this work would be to recalibrate the hydrological model using MAR data that provide these larger precipitation amounts at high elevations, to study their physical realism. Overall, a better understanding of the precipitation-elevation relationship in mountainous areas, especially its likely variations in time and dependency on event types, would improve hydro-meteorological analyses and simulations and make them more relevant. This would benefit from more observations in high altitude areas, a critical issue pointed out for a long time (e.g. Hingray et al., 2012). Despite their multiple limitations in mountainous areas, radar data could also provide valuable insights into orographic drivers of precipitation (e.g. Germann et al., 2022).

If downscaling models clearly need to be refined in the future, scientists should also consider improving the bias correction methods for such challenging configurations. They should also consider the possibility of better understanding and accounting for the precipitation orographic enhancement at the event-scale. Dynamical downscaling models such as MAR are likely promising tools for such analyses, but their value for areas with marked relief should be better estimated.

*Author contributions.* This study is part of CL's PhD thesis. BW and BH supervised the PhD. All authors contributed to the design of the study and the analysis framework. CL performed the simulations and produced the figures presented in this study. CL, MM and BH contributed to the writing of the document and to the editing of the paper.

*Competing interests.* The authors declare that they have no conflict of interest.

*Acknowledgements.* This work was supported by the French National Research Agency in the framework of the "Investissements d'avenir" program (ANR-15-IDEX-02). A part of the simulations could be performed from the Grenoble Alpes Research Data and Computing Infrastructure GRICAD (https://gricad.univ-grenoble-alpes.fr/, last access: 20 February 2023). We would like to thank Mondher Chekki for his precious technical and informatic support, and Julien Beaumet for providing the outputs of the MAR simulation. We also thank the editor and four anonymous reviewers for their constructive comments, which helped us to improve the manuscript.

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
