# Peer review of "Assessing downscaling methods to simulate hydrologically relevant weather scenarios from a global atmospheric reanalysis: case study of the Upper Rhône River (1902-2009)"

_Hydrology and Earth System Sciences, 2023_

## Author Comment (AC1)

**Response to Anonymous Referee #1**

We thank Referee #1 for reading the manuscript carefully and providing thoughtful and constructive comments. The comments are noted with RC, our responses with AC, and the intended additions or changes in the manuscript are underlined.

General comments

**RC1.1** This study evaluates two approaches for simulating over one century-long daily streamflow at different Upper Rhône River catchment locations. The analysis examines the potential of atmospheric reanalyses simulations (ERA-20C), conceptual glacier-hydrological model and two downscaling approaches for providing temporal variations of river discharges, low flow sequences and flood events. The results indicate that bias correction is crucial for both precipitation and air temperature and both downscaling models. While the observed multi-scale variations of discharges (daily, seasonal and inter-annual) are well reproduced, the results for low flows are less satisfactory.

The manuscript is within the scope of the journal and has a good structure. Still, the novel scientific contribution can be better formulated. The Introduction says that the comparison and evaluation of different downscaling approaches have been the subject of multiple previous studies, so it is not clear what are still the research gaps and how this manuscript contributes to some novel findings/knowledge. The manuscript will also benefit from some more detailed justification of why such one-century runs are needed and/or beneficial.

> **AC1.1** Thank you for this comment. This point was also highlighted by Referee #3 (RC3.1) and Referee #4 (RC4.1).
>
> As mentioned in the introduction, a large number of previous studies described downscaling approaches to adapt climate model outputs used to force hydrological models. Several studies highlighted that the choice of the downscaling method can strongly modulate the hydrological regime simulated with the same model (Wood et al., 2004; Quintana Seguí et al., 2010; Chen et al., 2011).
>
> Realistic hydrological simulations based on a specific downscaling approach and fine model calibration at the basin-scale are often used for relatively short hydrological simulations, lasting from one year to one decade (e.g. Habets et al., 1999; Boscarello et al., 2014), while longer periods are required to study trends in precipitation (Ménégoz et al., 2020) and river flows (Brönnimann et al., 2022).
>
> Bonnet et al. (2017) simulated the water cycle at the scale of the entire French hydrological system over the 20th century, a period long enough to discuss decadal variability and long-term trends. However, their model shows local deficiencies, particularly in mountainous areas where the snow cover plays a crucial role, and where the dams - not considered in their study - have significantly affected the river flows since the 1950s.
>
> The novelty of our study relies on three combined features: (i) we used a fine model calibration of the hydrological model to accurately simulate snowmelt, including the impacts of the dams at the scale of the Upper Rhône River catchment; (ii) we applied this model over a long enough period - the entire 20th century - to allow a complete evaluation of the model to simulate the

hydrological regime in terms of mean, variability, long-term trend and extremes; (iii) we compared two configurations, one based on statistical downscaling and the other one using dynamical downscaling, highlighting the need to accurately simulate the temperature and precipitation lapse rates. Overall, we confirm the need for an additional bias correction of atmospheric variables after the downscaling step and before the application of the hydrological model.

The novelty of our study will be deeply described in the revised version of the article.

A discussion of the value of the proposed approach with some alternative approaches (e.g. stochastic weather generator) will be useful as well.

Thank you for this comment. Yes, we could use a stochastic weather generator (WGEN). We have currently developed a multi-site WGEN for the simulation of long time series (> 10,000 years) of weather scenarios for the large Aare catchment in Switzerland (GWEX). The entire simulation framework is described in the article by Viviroli et al, 2022 published last year in HESS.

However, WGENs are developed to generate plausible weather sequences. They do not aim to reproduce the spatio-temporal variations in weather observed in a given area. The main reason for this is that they are not forced by variations in large-scale atmospheric information. Unlike WGENs, downscaling models are. This is an important advantage of downscaling models when scenarios have to be generated for other climate contexts, where changes in the circulation and state of the atmosphere are likely to lead to changes in local weather conditions.

We agree that this point could be clarified in the manuscript, although it is not a major issue here. If lack of space is not an issue, we will clarify this point.

The proposed modelling chains are quite complex, so a rigorous description of the methods is challenging, resulting in difficulty in reproducing the proposed experiments exactly.

We recognize that the modeling chains are quite complex. These chains are quite common when weather conditions are produced from downscaling models for climate change impact studies. We tried to give as much detail as possible, and where this was not possible, we provided references to articles in which the methods are described in more detail. We hope the reader will feel comfortable with the actual details of the description.

For ease of understanding, we will include in the revised version of the article, at the beginning of the section devoted to the methods, a synthetic summary of the model chains.

**RC1.2** My second suggestion is to add more process-related interpretations of results and findings. For example, to discuss and present which processes lead to low flows and floods in the study area and to demonstrate that the suggested chains represent well such runoff generation processes in the (sub) catchments. For example, I'm not sure whether individual correction of the bias for air temperature and precipitation does not result in some artificial combination which can affect, e.g. low flow simulations of the model. If the low flows occur in winter, the impact of air temperature bias correction will be more important than precipitation corrections or vice versa. So it will be interesting to provide more process interpretations in the results.

**AC1.2** Thank you for this very interesting suggestion. Yes, you are right, for the upstream part of the Upper Rhône River catchment, low flows occur mainly in winter, under the effect of low temperatures. So, for precipitation, the lapse rate issue has little impact on low flows (see Fig. 13c). For temperatures, it conversely has a significant impact (see Fig. 12d). For low flows, it is therefore unlikely that the bias correction for precipitation and temperature results in an artificial combination that affects low flow simulations. This is clear for high elevations sub-basins (see Fig. 12d and 13c for Rhône@Porte-du-Scex). It is also observed in the lower part of the Upper Rhône River catchment, but this is less evident.

With regard to its influence on floods, a similar discussion can be proposed for the precipitation lapse rate. Increased precipitation leads to increased snowfall in winter, followed by increased snowmelt and runoff in spring. Snowmelt floods are more or less proportional. The impact on autumn discharges is different. Larger precipitation directly leads to larger floods during this season. The increase in flood intensity is expected to be much larger than the increase in precipitation amount, due to the non-linear runoff generation process that increases with precipitation intensity.

For floods, the influence of the temperature lapse rate is much less evident. In this region, the largest floods often occur in autumn, due to the large amounts of precipitation. In autumn, however, the temperatures can be low enough for precipitation in high elevation areas to fall as snow (as this was the case with the 2000 flood in the Wallis canton, see Hingray et al., 2010), leading to "reduced floods" compared to floods that would have occurred with the full amount of precipitation as rain. A wrong temperature lapse rate in the model will definitively determine the elevation of the 1°C isotherm (elevation of the snowfall/rainfall limit) and therefore the "effective catchment area" for these events. In MAR, the temperature lapse rate is overestimated, resulting in too cold weather in high elevations areas. When MAR is corrected for mean temperature only (and not for the lapse), this leads to a reduction in the intensity of autumn flood (Fig. 12d).

We will add a comment on these points in the text.

**RC1.3** My third comment is on the validation of the results. It will be interesting to see how the procedures work in some independent (validation) time periods.

**AC1.3** Thank you for this important comment. It would obviously be important to assess it.

For basins for which we have concomitant time series of weather and discharge observations, not perturbed by dams, a classical split-sample test would be easy to perform. This has been done in many other works in this mountainous context. This work has been carried out for the GSM-SOCONT model by Schaefli et al. (2005) for 3 sub-catchments of the Upper Rhône River basin. This shows that the calibration is solid and robust.

For all other basins, this split-sample test is not possible. This is because all data of all the period have to be considered for the signature-based calibration. We are convinced that the signature-based calibration is also reasonable. For catchments for which a classical calibration was possible, we split the period in two. We applied the signature-based calibration on the discharge signatures of period P1 with weather data from P1. We then simulated the discharge

time series of period P2. The simulated time series remain in good agreement with the observed flows. The NSE coefficients are logically lower, but the difference is quite small.

We will add a few comments on this.

I wonder what is the relative contribution of each individual member of the proposed chain on the final results.

The main objective of our article is to assess the ability of two downscaling models, SCAMP and MAR, to produce hydrologically relevant precipitation and temperature scenarios from large-scale atmospheric information only. For this evaluation, we compare hydrological scenarios simulated from weather scenarios with reference hydrological scenarios also obtained by simulation, but from observed weather.

In other words, to evaluate weather scenarios in relation to observed weather, we use a filter, the hydrological filter of the considered Upper Rhône River basin. The interest of this filter is that it takes the hydrological point of view, which is a very demanding point of view due to the highly non-linear nature of many hydrological processes.

We therefore do not seek to estimate the relative contribution of each individual member of the simulation chain to the final results.

Specific comments

**RC1.4** Abstract: Please make the description of the results consistent with the conclusions. In the abstracts is written: "… the hydrological situations of low frequency (low flow sequences and flood events) are reasonably well reproduced." However in the conclusions is written: "The results for low flow activity are less satisfactory."

**AC1.4** Thank you for this comment. We will modify the abstract accordingly.

**RC1.5** Figure captions. The abbreviations used in the figures are not always explained in the captions, so it is sometimes difficult to understand them (without a detailed reading of the text).

**AC1.5** The following abbreviations will be explained in the captions: URR, P, T, MAP, MAT, Q, obs, ref, MAM, JJA, SON, DJF, BC, MAR, MAR-BC, SCAMP, SCAMP-BC, GSM-SOCONT.

**References**

Bonnet, R., Boé, J., Dayon, G., and Martin, E.: Twentieth-Century Hydrometeorological Reconstructions to Study the Multidecadal Variations of the Water Cycle Over France, Water Resour. Res., 53, 8366–8382, https://doi.org/10.1002/2017WR020596, 2017.

Brönnimann, S., Stucki, P., Franke, J., Valler, V., Brugnara, Y., Hand, R., Slivinski, L. C., Compo, G. P., Sardeshmukh, P. D., Lang, M., and Schaefli, B.: Influence of warming and atmospheric circulation changes on multidecadal European flood variability, Clim. Past, 18, 919–933, https://doi.org/10.5194/cp-18-919-2022, 2022.

Boscarello, L., Ravazzani, G., Rabuffetti, D., and Mancini, M.: Integrating glaciers raster-based modelling in large catchments hydrological balance: the Rhone case study, Hydrol. Process., 28, 496–508, https://doi.org/10.1002/hyp.9588, 2014.

Chen, J., Brissette, F. P., and Leconte, R.: Uncertainty of downscaling method in quantifying the impact of climate change on hydrology, J. Hydrol., 401, 190–202, https://doi.org/10.1016/j.jhydrol.2011.02.020, 2011.

Habets, F., Etchevers, P., Golaz, C., Leblois, E., Ledoux, E., Martin, E., Noilhan, J., and Ottlé, C.: Simulation of the water budget and the river flows of the Rhone basin, J. Geophys. Res.-Atmos., 104, 31 145–31 172, https://doi.org/10.1029/1999JD901008, 1999.

Hingray, B., Schaefli, B., Mezghani, A., and Hamdi, Y.: Signature-based model calibration for hydrological prediction in mesoscale Alpine catchments, Hydrolog. Sci. J., 55, 1002–1016, https://doi.org/10.1080/02626667.2010.505572, 2010.

Ménégoz, M., Valla, E., Jourdain, N. C., Blanchet, J., Beaumet, J., Wilhelm, B., Gallée, H., Fettweis, X., Morin, S., and Anquetin, S.: Contrasting seasonal changes in total and intense precipitation in the European Alps from 1903 to 2010, Hydrol. Earth Syst. Sci., 24, 5355–5377, https://doi.org/10.5194/hess-24-5355-2020, 2020.

Quintana Seguí, P., Ribes, A., Martin, E., Habets, F., and Boé, J.: Comparison of three downscaling methods in simulating the impact of climate change on the hydrology of Mediterranean basins, J. Hydrol., 383, 111–124, https://doi.org/10.1016/j.jhydrol.2009.09.050, 2010.

Schaefli, B., Hingray, B., Niggli, M., and Musy, A.: A conceptual glacio-hydrological model for high mountainous catchments, Hydrol. Earth Syst. Sci., 9, 95–109, https://doi.org/10.5194/hess-9-95-2005, 2005.

Viviroli, D., Sikorska-Senoner, A. E., Evin, G., Staudinger, M., Kauzlaric, M., Chardon, J., Favre, A.-C., Hingray, B., Nicolet, G., Raynaud, D., Seibert, J., Weingartner, R., and Whealton, C.: Comprehensive space–time hydrometeorological simulations for estimating very rare floods at multiple sites in a large river basin, Nat. Hazards Earth Syst. Sci., 22, 2891–2920, https://doi.org/10.5194/nhess-22-2891-2022, 2022.

Wood, A. W., Leung, L. R., Sridhar, V., and Lettenmaier, D. P.: Hydrologic Implications of Dynamical and Statistical Approaches to Downscaling Climate Model Outputs, Clim. Change, 62, 189–216, https://doi.org/10.1023/B:CLIM.0000013685.99609.9e, 2004.

---

## Author Comment (AC2)

**Response to Anonymous Referee #2**

We thank Referee #2 for reading the manuscript carefully and providing thoughtful and constructive comments. The comments are noted with RC, our responses with AC, and the intended additions or changes in the manuscript are underlined.

**RC2.1** The manuscript addresses the important topic of downscaling and bias-correction for hydrological modeling by applying different configurations of a "simulation chain" over an alpine river over the last century. I found the paper interesting and including meaningful results. Nevertheless, the paper lacks both important details on the hydrological modeling and discussion on the uncertainty related to the hydrological modeling within the applied simulation chains.

**AC2.1** Thank you for this comment. The conceptual hydrological model is mainly used here as a complex and non-linear filter to assess the weather scenarios. Note that the model can be imperfect as we also use it to produce the hydrological reference - against which hydrological scenarios derived from weather scenarios will be compared. The hydrological model is used as a filter to assess the relevance of the downscaling models.

As mentioned in our article, our downscaling models are forced with time series of large-scale atmospheric circulation. Perfect downscaling models would allow to perfectly reproduce the multi-scale spatio-temporal dynamics of weather over the time period used for the simulation. Downscaling models can be assessed, at several scales, for their ability to reproduce observed spatio-temporal dynamics of weather variables. This assessment is rather difficult, especially with regard to the covariability between weather conditions in different sub-basins of the Upper Rhône River basin, or with regard to the covariability between precipitation and temperature.

The hydrological filter is a powerful alternative filter for making this assessment, as the hydrological behavior of sub-basins is highly non-linear, and depends on all the covariability and dependency structures of meteorological conditions, both spatially and between variables. In brief, we need to assess the meteorological variability and covariability features that are important for hydrology and the hydrological model does this job well.

**Regarding the hydrological modeling methodology:**

**RC2.2** L118: please consider adding a diagram of the GSM-SOCONT model in the paper or at least in the Appendix.

**AC2.2 A scheme of the glacio-hydrological model GSM-SOCONT will be added as an appendix.**

**RC2.3** L122: are these "ice-covered" and "ice-free" parts of the studied catchments dynamics in time? Indeed, the ice-covered part of the upper catchments may have significantly changed over the last century.

**AC2.3** No, the hydrological model GSM-SOCONT assumes that glacier extent does not vary over time, and that glacier thickness is infinite. We recognize that this is a strong assumption as the ice-covered part of the upper catchments may have changed significantly over the last century.

This will be mentioned in the text.

**RC2.4 L212: what are these additional criteria considering streamflow availability?**

**AC2.4** We guess you are referring to line 121 and not line 212. The entire catchment was divided into 18 sub-basins. They were selected so that they are roughly the same size, and so that a gauging station is located at the outlet of a sub-basin wherever possible (see Fig. 1).

We will change the formulation in the text and specify that.

**RC2.5** L124 to 126: the calculation of the potential evapotranspiration (PE) time series needs to be more deeply presented in the paper: which CRU dataset has been used for this calculation, what is the spatial resolution of this CRU dataset, and how relevant is this database over this region and in this hydroclimatic context? What about this regional temperature-PE relationship in a non-stationary context?

**AC2.5** We acknowledge that estimating potential evapotranspiration (PET) in such a mountainous area is a critical issue, and that the CRU data set may give a rough and poor estimate of this variable. However, to the best of our knowledge, PET observations are very scarce in the region, making an evaluation of CRU products almost impossible.

To develop our T-PET model, we use monthly CRU values of PET and temperature T at a resolution of 0.5° x 0.5° for the 1900-2010 period. PET is assumed to be a linear function of temperature T:  $PET(t) = a \times T(t) + b$ . The coefficient b depends on the calendar month. This model, estimated for the entire Upper Rhône River catchment, is then used at a daily time step to produce daily PET for all its hydrological units. The PET values of the CRU were calculated from a variant of the Penman-Monteith formula (Harris et al., 2014).

We assume that this relationship is the same for the entire period. Variations may have occurred during the period, for instance due to changes in land use and vegetation cover. This will be worth dedicated investigations in future works.

This point will be specified in the text.

**RC2.6** L147 to 148: the routing part of the hydrological model needs to be more deeply presented in the paper. What are the hypotheses? How many parameters are devoted to the routing part of the model?

**AC2.6** We did not use a routing module. The discharge at a given point in the catchment is simply the sum of the discharges in the upstream sub-basins. In previous work, we used a Muskingum routing model (Hingray et al., 2010), but it was not really necessary here. The size of the catchment is not very large and we use a daily time step for hydrological simulations.

We will change the formulation in the text and specify that.

**RC2.7** L149 to 151: please state in the paper the total number of parameters that needs to be calibrated for each sub-catchment and add a list of these parameter in the paper (in the Appendix?).

**AC2.7** Please note that the number of parameters (and their names) to be calibrated is already mentioned on lines 149 to 151.

**RC2.8** L183 to 184: please give more details on the regionalization procedure applied, and list the ungauged catchment studied here.

**AC2.8** Thank you for this comment. All the ungauged sub-basins upstream a given gauging station are calibrated at the same time and are forced to share the same parameters set. The discharge time series used for the calibration is the time series simulated with this multiple sub-basins configuration at the downstream gauging station, where observations are available.

The ungauged sub-basins can be seen in Fig. 1. The sub-basins grouped together for the calibration of their parameters are also listed in Table 2 in Supplementary Materials.

**We will clarify this point in the text.**

Regarding the discussion about the hydrological modelling uncertainty within the applied chains:

**RC2.9** L172 to 174: "For gauged catchments for which the hydrological behavior is significantly altered over P1 and for which "natural" flow observations are available prior to 1950, parameters were estimated based on hydrological signatures (Sivapalan et al., 2003; Winsemius et al., 2009). In the present case, parameters are calibrated so that simulated signatures reproduce at best observed ones but observed and simulated signatures come from different periods following Hingray et al. (2010) (e.g. 1961-2015 and 1922-1963 respectively for the Viège basin). We thus assume that the weather regimes and the natural hydrological behavior of the catchment have not significantly changed over the last century, which seems a reasonable assumption to make in first approximation." This strong hypothesis needs to be more deeply investigated in the paper: what about potential significant interannual / decadal hydro-climatic variability on the hydrological model parameters? What about using adjustment on long-term climatic information (e.g. Nicolle et al., 2013, doi: 10.1002/2012WR012940)?

**AC2.9** The stationarity assumption is indeed a strong one. Unfortunately, there is no other choice for the calibration of some sub-basins of the Upper Rhône River catchment, due to the flow data available here, which have been strongly perturbed by dams since the 1950s. See the figure below from Hingray et al. (2010).

It is difficult to assess the relevance of the stationarity assumption for hydrological regimes. This assumption implies that the following two independent assumptions are valid: the stationarity of precipitation regimes and the stationarity of the natural hydrological behavior of the catchment.

Apart from the impact of dams, the hydrological behavior of sub-basins could have changed over the period, due to changes in land cover (glacier and vegetation). About glacier retreat over the last century, Huss (2011) explains that (i) the contribution of glacier melt is relatively stable over the 20th century, with similar glacier contribution over the periods 1961-1990 and 1908-2008, and (ii) glacier retreat strengthened over the period 1988-2008, with glacier contributions on the Rhône River increasing by 13% for the month of August. Considering a stationary glacier contribution over the whole period is therefore not fully satisfactory for the

last two decades, but acceptable over the whole period. On the other hand, the impact of vegetation changes on hydrology is often difficult to highlight. Moreover, in the region, a large part of the sub-basins is free of vegetation (due to elevation). Assuming a stationary natural hydrological behavior seems thus to be reasonable here.

The stationarity of the precipitation regime may be an issue. Significant interannual variability in annual precipitation generally exists for precipitation. It is much lower when the mean interannual precipitation is considered, as it is the case here (we compare the hydrological signatures of two periods of several decades each). To the best of our knowledge, no significant precipitation trends have been observed in the region over the last century (e.g. Masson and Frei, 2016; Scherrer et al., 2016).

Fig. 1 Effects of water works on the URR basin hydrology: (a) dam storage capacity evolution over the 1905–2005 period; (b) and (c) 10, 50 and 90% percentiles of the mean monthly discharge at Porte du Scex for 1905–1955 and 1955–2005, respectively; and (d) annual maximum peak discharge. For 1987, 1993 and 2000 floods:  $\blacklozenge$ : observed discharge;  $\Box$ : reconstructed discharge (from Hingray *et al.*, 2009).

**Using adjustment on long-term climatic information.**

Using a blending neighbor approach for the calibration, as in Nicolle et al. (2013), is indeed an attractive alternative. However, this would not be straightforward. In the upstream sub-basins of the Upper Rhône River basin, the hydrological regime is mainly "glacial" (dominated by glacier melt, according to the Swiss classification). It shifts to a "glacio-nival" regime (glacier/snow melt), a "nival" regime (dominated by snow) and a "pluvio-nival" regime (dominated by a mixture of snow and precipitation) as we move downstream. The flow signatures (such as the mean interannual cycle) at different locations along the river present clearly different patterns. It would therefore probably be difficult to find neighboring basins to consider as donors of the signatures used for the signature-based calibration.

**RC2.10** L180: Table S1 needs to be presented in the paper and not in the Appendix, and more deeply discussed, since the performance obtained on several subcatchments are poor (e.g. Arve@Genève, BDM). What are the potential reasons for these differences in performance between the studied catchments? Please also consider producing the Figure S2 for every studied catchment in the Appendix.

**AC2.10** Thank you for this suggestion. Table S1 could be moved into the article, but it is already quite long. We agree that performance is rather poor for some sub-basins. However, data "quality" varies considerably from one sub-basin to another. For some of them, daily meteorological data and daily non perturbed discharge data are not available. This requires a signature-based approach which, logically leads to lower "performance".

To assess the decrease in performance due to the signature-based approach, we carried out in the meantime the following experiment. For four sub-basins with concomitant meteorological and flow data, we first applied a classical calibration, followed by a signature-based calibration. The signature-based calibration is quite effective, but is still less efficient than the classical calibration.

We will comment this point in the manuscript and include the results of this experiment in Supplementary Materials.

Even when concomitant series are available, the classical calibration is unfortunately not always very effective. This is indeed the case for the gauging station Arve@Genève, Bout-du-Monde. For this basin, the hydrological regime is much less snow-dominated than elsewhere in the Upper Rhône River catchment. The density of the precipitation network is perhaps not dense enough to provide a good estimate of precipitation inputs.

As suggested by the reviewer, the results of the classical and signature-based calibrations will be given for the other sub-basins in Supplementary Materials.

These will be Fig. S3 and S4 below:

---

## Author Comment (AC3)

**Response to Anonymous Referee #3**

We thank Referee #3 for reading the manuscript carefully and providing thoughtful and constructive comments. The comments are noted with RC, our responses with AC, and the intended additions or changes in the manuscript are underlined.

This is a very well written paper (with high quality figures) on discharge simulations based on climate model outputs, with "a typical simulations chain" (abstract first line). The paper was easy to read, which is a great achievement for such a complex modelling paper.

**We thank Referee #3 for his/her encouraging general comments.**

**RC3.1** The topic is probably undersold: it is not just a case study but one of the few attempts to produce long simulations of the past, with a high number of challenges to overcome. I am not aware of long simulation studies back in the past in mountainous environments with climate model outputs (I might not be on top of that literature but I suspect there are very few).

Accordingly, the added value to the literature should be formulated in a much more straightforward way as early as possible in the introduction and again in the conclusion.

The method is not well captured in the abstract or in the paper overall and calling this a "typical simulation chain" is perhaps underselling. A critical step in such simulation chains is how to combine reanalysis data with a hydrological model, especially in mountainous areas. What does this study do differently than other works? Why is the approach more interesting?

**AC3.1** Thank you for this comment. This point was also highlighted by Referee #1 (RC1.1) and Referee #4 (RC4.1).

As mentioned in the introduction, a large number of previous studies described downscaling approaches to adapt climate model outputs used to force hydrological models. Several studies highlighted that the choice of the downscaling method can strongly modulate the hydrological regime simulated with the same model (Wood et al., 2004; Quintana Seguí et al., 2010; Chen et al., 2011).

Realistic hydrological simulations based on a specific downscaling approach and fine model calibration at the basin-scale are often used for relatively short hydrological simulations, lasting from one year to one decade (e.g. Habets et al., 1999; Boscarello et al., 2014), while longer periods are required to study trends in precipitation (Ménégoz et al., 2020) and river flows (Brönnimann et al., 2022).

Bonnet et al. (2017) simulated the water cycle at the scale of the entire French hydrological system over the 20th century, a period long enough to discuss decadal variability and long-term trends. However, their model shows local deficiencies, particularly in mountainous areas where the snow cover plays a crucial role, and where the dams - not considered in their study - have significantly affected the river flows since the 1950s.

The novelty of our study relies on three combined features: (i) we used a fine model calibration of the hydrological model to accurately simulate snowmelt, including the impacts of the dams

at the scale of the Upper Rhône River catchment; (ii) we applied this model over a long enough period - the entire 20th century - to allow a complete evaluation of the model to simulate the hydrological regime in terms of mean, variability, long-term trend and extremes; (iii) we compared two configurations, one based on statistical downscaling and the other one using dynamical downscaling, highlighting the need to accurately simulate the temperature and precipitation lapse rates. Overall, we confirm the need for an additional bias correction of atmospheric variables after the downscaling step and before the application of the hydrological model.

The novelty of our study will be deeply described in the revised version of the article.

**RC3.2** A concise summary of the method would help (take a hydro model, calibrate on observed meteo and streamflow data, run with meteo scenarios generated based on reanalysis data; contrary to many other works, no simple spatial downscaling of reanalysis data but use of a weather generator to produce mean areal precipitation scenarios).

**AC3.2** Thank you for this suggestion. In the first version of the article, each step of the model chain was described, but a general description of the method was clearly missing.

We will include in the revised version of the article, at the beginning of the section devoted to the methods, a synthetic summary of the model chains.

Details:

**RC3.3** Method: it should be made very clear at what scale MAP (mean areal precip) and MAT (temperature) are defined. I see from the sentence "bucket-type model that uses time series of Mean Areal Precipitation (MAP) and Temperature (MAT) as inputs for each hydrological unit." that MAP and MAT is defined at the elevation band scale (?)

**AC3.3** Yes, MAP and MAT are defined at the scale of elevation bands. The MAP and MAT time series are different from one elevation band to another, due to the different elevations and Thiessen weights for the neighboring stations. This point will be clarified.

Note that MAP and MAT are also estimated for other spatial scales. In our evaluations, we compare simulated and observed MAP and MAT at the scale of the 5 major sub-basins shown in Fig. 1. We will specify the spatial scale at which the different MAP and MAT are considered in each section.

**RC3.4 "a daily potential evapotranspiration time series is derived from temperature", how? Linear?**

**AC3.4** To develop our T-PET model, we use monthly CRU values of PET and temperature T at a resolution of  $0.5^{\circ} \times 0.5^{\circ}$  for the 1900-2010 period. PET is assumed to be a linear function of temperature T: PET(t) = a x T(t) + b. The coefficient b depends on the calendar month. This model, estimated for the entire Upper Rhône River catchment, is then used at a daily time step to produce daily PET for all its hydrological units.

This point will be specified in the article.

**RC3.5** "For each RHHU, daily MAP and MAT are estimated from neighboring weather stations using the Thiessen's weighting", why Thiessen? Probably very inappropriate for mountain environments? And how is the Thiessen obtained between points (weather station locations) and elevation bands?

**AC3.5** As shown on the two maps in Fig. A, the Thiessen polygons are derived from the network of available stations in the region (within the catchment and in its neighborhood).

Figure A. Thiessen polygons for (left) the precipitation stations and (right) the temperature stations considered for the study.

The Thiessen weights of the different stations for any given surface area are estimated as follows. Let's call Ai, the surface area of the considered unit covered by the ith station. For most stations, Ai will be zero. The weight associated to station i is simply Ai / A where A is the surface area of the considered spatial unit. Whatever the spatial unit (elevation band, subbasin, URR basin, ...), the sum of the weights is always equal to one for this unit. The weights are always different from one spatial unit to another, from one elevation band to another, depending on the proximity of the stations to that unit.

Note that the URR basin is divided into 18 sub-basins. For each of them, we always have 5 to 10 stations for which the weights are non-zero for precipitation. If we disregard the issue of elevation dependency, which is widely discussed in the article, we considered that this number provides a reasonable estimate of MAP for these areas. Note that this issue of elevation dependency would not be better addressed by other spatialisation approaches (inverse distance, kriging).

For temperature, the number of stations with non-zero weights is lower, but this variable, when considered at a given reference elevation, varies much less spatially. The way temperature is estimated for each unit is as follows: apply a weighted Thiessen sum of station temperatures, corrected in a previous step to correspond to the reference elevation for that unit (thanks to the temperature lapse rate discussed in the article). For most time steps, the linear temperature-elevation relationship is very robust, allowing a very reasonable temperature estimate for altitudes different from those of the stations.

**RC3.6** What is the storage equation in this model? Is it the storage-volume relationship? Where do you get if from?

**AC3.6** This is the water level-storage equation mentioned in the next sentence. This point will be clarified.**

**RC3.7** How do you construct the linear outflow relationship? As far as I see from one year of observed data (extract from the pdfs https://www.hydrodaten.admin.ch/en/seen-und-fluesse/stations/2028 and https://www.hydrodaten.admin.ch/en/seen-und-fluesse/stations/2606) any outflow is possible for low water levels (see figures below) and a linear relationship only holds for high water levels and flow beyond 500 m3/s

**AC3.7** We could not find these figures on the above-mentioned websites. However, the data used for these figures are likely those of the last few years, decades when a regulation applies. The water level-outflow relationship has thus no reason to be univocal, since the operations for a given day (hour) must be defined according to the state of electricity demand and electricity prices for that day (hour). An exception is made for high water levels, when operations aim to limit the water level in the lake to avoid riparian flooding. In this case, the only decision variable for operation is the lake water level, which explains the univocal relationship found in the data.

If the lake were unregulated, we would probably have a univocal relationship in first approximation, due to its very large surface area and the topographical configuration of its outlet. The discharge would be a function of the hydraulic head. As the water level does not vary a lot in the lake (much less than 2 m), we considered the relationship was to be linear.

**RC3.8** Can you say something about the evaporation equation, what does it include? I have not seen the Rohwer's equation (Rohwer, 1931) before.**

**AC3.8** We used Rohwer's equation (see equation below) as it uses an additional variable, namely atmospheric pressure at 10 m, and this was found of potential interest for further use of the model in other climate contexts. Note that Rohwer's equation is very close to the better-known Penman (1948) equation (see below), and that the difference of the mean annual evaporated water over the 1961-2010 period between Rohwer's and Penman's estimates is only 3%. This choice is thus expected to have little impact on the main conclusions of our study.

**Rohwer (1931) equation:**

 $E(t) = 0.77 \times (1.465 - 0.0186 \times p_{10m}(t)) \times (0.44 + 0.118 \times U_{10m}(t)) \times (es_surf(t) - ea_{10m}(t))$  where :

- E(t) (in j-1) is the evaporation from Lake Geneva at time step t.
- p\_10m(t) (inHg) is the atmospheric pressure at 10 m at time step t.
- U\_10m(t) (mph) is the wind speed at 10 m at time step t.
- es\_surf(t) (inHg) is the vapor pressure of saturated air at lake surface temperature T\_surf(t) at time step t.
- ea\_10m (t) (inHg) is the current vapor pressure in the air at 10 m at time step t.

**Penman (1948) equation: E(t) = 0.35 x (1+0.24 x U\_10m(t)) x (es\_surf(t)-ea\_10m(t))**

**RC3.9** "In the present work, SCAMP was used to generate 30 time series of daily spatial weather scenarios for the 1902-2009 period from ERA-20C reanalysis outputs." At what spatial scale? At the scale of the 7 km x 7km of the MAR model? This would be in contradiction with Fig. 3 that says that you generate MAP (mean areal precip) estimates and not first spatial estimates? So, do you first produce spatial estimates, i.e. daily time series per pixel? If yes, how do you combine the pixels into MAP estimates? By taking the pixels within the elevation band? Does this make any sense?

**AC3.9** Thank you for this comment. There was indeed a mistake in Fig. 5, where disaggregated data are shown for MAR and SCAMP. SCAMP indeed produces precipitation and temperature time series at station level, and the stations at the end of the process are exactly the same as the observation stations.

**Fig. 5 will be corrected: the spatial resolution of P/T is "stations" for obs and SCAMP, and "7x7 km" for MAR.**

Sorry for this. We understand it was almost impossible to understand the spatial aggregation/disaggregation process. We hope it is now clear.

**RC3.10** Method overall: how do you have confidence that the conceptual model calibrated on observed streamflow data with observed station meteo does a good job if used with weather generator scenarios produced at a different scale? What part of your method ensures this?

**AC3.10** Thank you for this comment. The conceptual hydrological model is mainly used here as a complex and non-linear filter to assess the weather scenarios. Note that the model can be imperfect as we also use it to produce the hydrological reference - against which hydrological scenarios derived from weather scenarios will be compared. The hydrological model is used as a filter to assess the relevance of the downscaling models.

As mentioned in our article, our downscaling models are forced with time series of large-scale atmospheric circulation. Perfect downscaling models would allow to perfectly reproduce the multi-scale spatio-temporal dynamics of weather over the time period used for the simulation. Downscaling models can be assessed, at several scales, for their ability to reproduce observed spatio-temporal dynamics of weather variables. This assessment is rather difficult, especially with regard to the covariability between weather conditions in different sub-basins of the Upper Rhône River basin, or with regard to the covariability between precipitation and temperature.

The hydrological filter is a powerful alternative filter for making this assessment, as the hydrological behavior of sub-basins is highly non-linear, and depends on all the covariability and dependency structures of meteorological conditions, both spatially and between variables. In short, we need to assess the meteorological variability and covariability features that are important for hydrology and the hydrological model does this job well.

**RC3.11** Model calibration on signatures: did you check for the catchments with observed concomitant streamflow if the signature calibration gives good results?**

**AC3.11** Thank you for this comment. To assess the relevance of the signature-based calibration, we carried out the following experiment. We recalibrated the parameters of subbasins with a natural (or at least not significantly altered) hydrological regime, using only the hydrological signatures (Fig. S5).

We then examined the time series obtained with this signature-based calibration. The simulated time series remain in very good agreement with the observed flows (Fig. S6). The NSE coefficients are logically lower than those obtained with the classical calibration approach, but the difference is quite small (Fig. S6).

We will add comments on these analyses in the article and provide Fig. S5 and S6 in Supplementary Materials.

Interannual daily regime

Gumbel plot for annual discharge maxima

---

## Author Comment (AC4)

**Response to Anonymous Referee #4**

We thank Referee #4 for reading the manuscript carefully and providing thoughtful and constructive comments. The comments are noted with RC, our responses with AC, and the intended additions or changes in the manuscript are underlined.

This paper by Legrand et al. presents and discusses the results of a modelling chain that uses largescale atmospheric information (ERA-20C reanalysis) to produce continuous daily river discharge at several stations of the Upper Rhône river catchment (~ 11000 km2). This modelling chain includes a semi-distributed hydrological model (GSM-SOCONT) and a downscaling model. Two downscaling models (statistical or dynamical) are actually used and tested. The results are analysed over several periods (1902-2009, 1961-2009, 1981-1983, 1920-1949, 1950-1979, 1980-2009) and according to various indicators (daily discharges, mean monthly and annual discharges, low and high flows).

This paper clearly corresponds to a huge amount of work. A lot of material and results are presented, the Figures are very rich.

**We thank Referee #4 for his/her encouraging general comments.**

The drawback of this very rich content is that it is not always easy for the reader to navigate and appreciate the results in the present form of the manuscript. My main remarks and recommendations to improve the paper are listed below :

Title and focus.

**RC4.1** The paper is entitled "Simulating one century (1902-2009) of river discharges, low flow sequences and flood events of an alpine river from large-scale atmospheric information" which is a very general title and does not really correspond to the real focus of the paper. In my opinion, as stated by the authors themselves, the objective of the work presented here is to assess the ability of downscaling chains to simulate hydrologically relevant weather scenarios (p12, l. 282-284). This is consistent with the detailed description of the different steps of the downscaling models, and the presentation of the results as comparison to a reference that is not the observed discharge (but rather the discharge simulated by the hydrological model forced by reference observations). I therefore suggest to make this objective more clear in the Introduction, to change the title accordingly and to trim the parts of the text that do not contribute directly to this objective (a lot of details about the hydrological model description and set up could be placed in supplementary material for example).

**AC4.1** Thank you for this comment. In the introduction, we will explain more clearly our aim to better understand the crucial steps required to simulate hydrologically relevant weather scenarios.

Note that a description of the novelty of our study will be included in the introduction, as this point was also highlighted by Referee #1 and Referee #3. Please see our answer RC1.1 or RC3.1, which is the same.

As suggested, we propose to replace the title of the study with the following words: "Hydrological implications of different downscaling approaches and bias correction of a global atmospheric reanalysis: case study of the Upper Rhône River (1902-2009)" We decided to keep the part devoted to the hydrological model in the main article, as it contains some crucial and innovative steps, particularly those related to the presence of dams along the river, as well as the discussion linked to lapse rates.

**Figures.**

The Figures contain a lot of information, maybe too much compared to what the reader is able to see and to what is necessary to illustrate the author's point. They would gain a lot from a bit of trimming. As examples :

**RC4.2** Fig 1 : the names of the gauging stations on the map are not necessary (they are also in Fig 2) and prevent the reading.

**AC4.2** We prefer to keep the names of the gauging stations in Fig. 1, as this helps to identify ungauged sub-basins and to locate the gauging stations where hydrological assessments will be made later. For reasons of legibility, we will however use shorter names.

**RC4.3 Fig 7 : is it really useful to present both MAT and DMAT ?**

**AC4.3** In Fig. 7b for MAT, it is rather easy to see the monthly biases obtained between observations and SCAMP simulations. However, the bias for the MAR simulation is difficult to identify due to the high seasonality of the raw variable. Fig. 7c is given to highlight this "small" bias, which nevertheless has strong implications on simulated hydrology. We therefore think it is important to keep both figures.

**RC4.4** Figs 8 and 9 : these Figs are more illustrative, maybe present more synthetic results before (Fig 10) + it could be worth presenting a Table with a few synthetic values (bias, KGE) so that we can have a general idea of how the simulation chains perform compared to reference. Why presenting simulations with BC before without BC ? + is it really interesting to present the time series of lake level ? + add the names of the stations directly on the Figures

**AC4.4 Presenting a Table with a few synthetic values.**

Thank you for this suggestion. A Table with the NSE coefficients for the dynamical downscaling model (MAR/MAR-BC) and the statistical index CRPSS (Continuous Ranked Probability Skill Score) for the statistical downscaling model (SCAMP/SCAMP-BC) will be added to show the performance gain associated with bias correction.

**Why presenting simulations with BC before without BC?**

In the section devoted to the evaluation of SCAMP and MAR weather scenarios, we highlight the significant biases for both MAP and MAT variables. It was therefore logical for us to consider that hydrological simulations have to be performed with bias-corrected weather scenarios. This is why we present these results first. However, the presentation of the simulations with the raw scenarios follows. This allows us to show that the discrepancies obtained on discharges do not correspond to what we could have expected from the MAP biases, and this because of the MAT biases. In our opinion, this makes the demonstration of the importance of MAT biases more effective. This is the main reason why simulations without bias correction follow simulations with bias correction.

**Is it really interesting to present the time series of lake level?**

As the lake has a significant buffering effect on downstream hydrology, due to storage cycles, it was also important to check that the model is able to simulate the main features of storage variations.

Add the names of the stations directly on the Figures. As suggested, the name of the gauging stations will be added on the figures.

**RC4.5** Fig 11 : too many lines on Fig c) + Figs a) and b) don't work. It is not clear that there is a "column" for each period + the only thing we see is the comparison between SCAMP and SCAMP-BC for each period

**AC4.5** Thank you for this comment. The columns for the 3 periods will be more clearly separated in Fig. 11a and b, and for greater clarity, Fig. 11c will be divided as follows:

Figure 11. (a) Flood activity and (b) low flow activity at Rhône@Bognes for three 30-year sub-periods: 1920-1949, 1950-1979 and 1980-2009. (c) Mean annual discharges at Rhône@Bognes for the period 1902-2009 simulated with MAR-BC and SCAMP-BC. (d) The same for simulation with MAR and SCAMP. The grey and green bands represent the confidence interval at 90 % level. The median scenarios are indicated by the black and green solid lines. The "references" are observed discharges for the 1920-1960 period and simulated discharges from observed weather for the 1961-2009 period. MAR/MAR-BC: hydrological simulation forced by the raw/bias-corrected weather scenario produced with the dynamical downscaling model. SCAMP/SCAMP-BC: hydrological simulations forced by the raw/bias-corrected weather scenarios produced with the statistical downscaling model.

**RC4.6** Fig 12 : Fig a) not legible. What am I supposed to see on Fig b) ? Why not removing MAR simulation (since lapse correction is only done to the MAR-BC simulation) ?

**AC4.6** The size of the dots in Fig. 12a will be reduced for greater clarity. In Fig. 12a and b, note that the blue and red lines overlap. For greater clarity, this point will be added to the caption.

In Fig. 12d, we show that the simulation with a classical bias correction of the Mean Areal Temperature (MAT) (bias correction of the mean for a reference altitude - black curve) leads to worse simulated discharges than those obtained without any bias correction (cyan curve). It is therefore useful to show the MAR simulation. The problem is clearly the bias in the temperature lapse rate. Simulations with double bias correction, i.e. 1) bias correction of MAT at a reference altitude, and 2) bias correction of temperature lapse rates, perform well (dark blue curve). Fig.12 a, b and c are different ways to highlight the importance of the bias for the temperature lapse rate. To the best of our knowledge, the discrepancy between the observed and simulated statistical distributions of the temperature lapse rate has never been recognized or illustrated before. In our opinion, it is therefore useful to highlight this issue here (Fig. 12b).

RC4.7 Fig 13 : Fig b) : why lines and crosses on this Fig for the different simulations ?

**AC4.7** Thank you for this comment. We will also use lines for the MAP estimates with the two different precipitation lapse rates.

Acronyms and notations.

**RC4.8** The paper uses many acronyms that are sometimes confusing and could be simplified. I particularly struggled with MAP / MAR / MAT (the name of the model can't be changed, but the names of the variables probably can), with basin-scale, RHHU, grid points, station (Fig 3 and accompanying text), see next remark. BC (Bias correction) is also not clear as it is used for different corrections (not the same for MAR and SCAMP, which I can understand, but also for temperature lapse-rate for MAR. About this last correction, if it is included in the results presented in section 5 (which is the case from what I can understand), why is it not presented along with the other corrections in section 4 ?

**AC4.8** Thank you for this suggestion. For the notations, we will specify that we are referring to the dynamical downscaling model when we use the notation MAR. We will also specify the spatial scale at which the different MAP and MAT are considered in each section.

Sect. 4.3 "Bias Correction" will be clarified and the equation for the lapse rate correction will be moved there, where we discuss the bias correction for MAP and for MAT.

Note that the principle of bias correction is the same for both models (MAR and SCAMP) and for all variables, i.e. MAP, MAT and temperature lapse rates. It is a quantile mapping bias correction. We simply estimate the correction function on different time series, depending on the model and variable considered.

**Spatial resolution**

**RC4.9** I really struggled (and did not succeed) at understanding exactly what was done in terms of spatial aggregation / disaggregation for the weather scenarios in SCAMP. In step (1) how do to change from station (= point) observations to basin-scale resampled values (basin-scale = whole catchment or subcatchments?). Are the stations at the end of the process the same as the observation stations ? (I understand rather RHHU = subcatchment but later in the text it is still referred as neighbouring stations ad inputs to the hydrological model, see p 22 I 410-411). Similarly Fig 5 presents the numerical experimentation plan with the hydrological model forced by P and T at 7\*7 km resolution. Please, make all this clearer !!

**AC4.9** Thank you for this comment. There was indeed a mistake in Fig. 5, where disaggregated data are shown for MAR and SCAMP. SCAMP indeed produces precipitation and temperature time series at station level, and the stations at the end of the process are exactly the same as the observation stations.

Fig. 5 will be corrected: the spatial resolution of P/T is "stations" for obs and SCAMP, and "7x7 km" for MAR.

Sorry for this. We understand it was almost impossible to understand the spatial aggregation/disaggregation process. We hope it is now clear.

Time periods and indicators for the results.

**RC4.10** As said before, there are a lot of results, presented for various times periods and various indicators that change from Figure to Figure. This does not make the reader's task easy. I would be nice, along with the experimental setup, to define in a Table the various indicators and time periods used and explain why they were selected.

**AC4.10** Thank you for the suggestion. We will add a table to the Experimental setup section, with the indicators, the assessment objectives, the time periods and the areas considered for the evaluations.

**RC4.11** If I am not mistaken, the results of the calibration of the hydrological model for the reference simulation are not presented. It would be nice to add a few elements about this, just to make sure that the reference model is not completely off the track.

**AC4.11** The calibration efficiency is already mentioned in Supplementary Materials (Table S1):

- For all sub-basins for which a classical calibration was carried out, we give the NSE coefficient calculated from observed and simulated time series of discharge.
- For all sub-basins for which a signature-based calibration was carried out, we give the NSE coefficient calculated from observed and simulated regimes and the Kolmogorov-

Smirnov coefficient calculated from observed and simulated distributions of annual discharge maxima.

Reference MAP and MAT.

**RC4.12** I am not a specialist of estimation of weather variables in montainous areas but I am surprised by the methodology used here, which consists of Thiessen polygons with a density of stations that is not so high (62 raingauges for ~ 11000 km2, even less temperature stations). There are several other methods for the estimation of areal P and T, from very simple such as inverse-distance weighting to more complicated (kriging). What is the reason of choosing such a simplistic approach ? How confident can we be with these "reference" values ? This should at least be discussed thoroughly.

**AC4.12** As mentioned previously, spatial variations in weather in a mountainous environment can be obviously significant, often due to the effect of topography. A satisfactory estimate of MAT and MAP is however impossible to achieve from stations only, even in this Upper Rhône River configuration where the network of precipitation and temperature stations is denser than in many basins worldwide. In our experience, it is even rather inextricable.

The inverse distance is of course interesting, but has other limitations: the distance to be considered in a mountainous environment is not trivial at all. Two stations distant by 5 km in two different neighboring valleys are probably much more distant than 2 stations distant by 10 km in the same valley. A topographical distance has been proposed in some papers, but the choice of weight linked to topography has to be estimated and is always based on different assumptions and choices that are rarely justified (at least from a physical point of view).

Using an inverse distance approach to estimate MAP is not necessarily straightforward either. It would be necessary to estimate the weights of all stations for all (grid) points in the spatial unit considered. In most cases, the weights are estimated from the distances to the centroid of the spatial unit, which is also an important and not really satisfactory assumption. Let's consider 2 basins A and B of the same shape, very elongated (this is often the case in the Upper Rhône River catchment), with the same surface area and centroid. The only difference between the two would be their orientation: basin A is oriented in a West-East direction, basin B in a North-South direction. With an inverse distance approach, all weather stations would have exactly the same weights for both basins, which would obviously not be the case in reality.

Kriging would also be interesting. We could try Kriging with external drift, where the drift is linked to the topography. A number of works have been presented on this subject. However, they generally apply to interannual mean variables, i.e. mean annual precipitation, mean annual temperature. At these aggregated scales, there is indeed a clear relationship between meteorological conditions and altitude. However, this relationship is no longer valid for each individual time step of the period to be considered. As mentioned in the article, the elevation-temperature relationship (if any) varies from day to day. Kriging is much more computationally demanding than Thiessen polygons. It also requires the calibration of a functional relationship (the variogram model). The quality of this model must be verified. It must be verified for each time step, which would be really difficult to do.

For precipitation, as mentioned in the article, the nonexistent precipitation-elevation relationship at high resolution and at the scale of precipitation events is another important limitation.

For these reasons, and for the sake of simplicity, we have chosen to work with the Thiessen method. Note that estimates of MAP and MAT are obtained from a rather high number of stations (5 to 10 for each of the 18 sub-basins for example for precipitation). We therefore expect not to produce very poor estimates of MAP and MAT.

Note also that this choice should not influence the main results of our work, notably the critical issue of bias in downscaling models, especially that of the temperature lapse rate for dynamical downscaling, and the critical issue of the precipitation lapse rate at the event scale.

We agree that this point should ideally be discussed in depth. However, the article is rather long and covers many other issues. We will just mention it briefly.

**RC4.13** p 24 the authors write "no precipitation-elevation relationship was considered" although p 6: "and a regional and time-varying elevation-temperature relationship".

**AC4.13** The issue of elevation dependency is different for precipitation and temperature.

For temperatures, the dependency is very strong and robust for most time steps, and is accounted for in (likely) all hydrological models (for mountainous areas at least, where snow is important) with a linear elevation-temperature relationship. The slope (the lapse rate) of this relationship can vary from one time to another. We have accounted for this temporal variability, estimated from station observations (see Sect. 4.1.1).

For precipitation, as discussed in Sect. 6.3, there is no clear precipitation-elevation relationship at the event scale, and the inclusion of the mean relationship observed for a long time period is problematic for flood simulation. This is why our simulations do not take into account a precipitation lapse rate. To the best of our knowledge, the precipitation-elevation issue was not described in previous works using hydrological models. However, it is potentially critical. We hope that our work will lead scientists to pay more attention to this issue in future studies.

**Floods.**

**RC4.14** I have a few comments on the "flood" part of the paper. First, I do not really agree with the use of the term "flood" given what is presented here. In 5.3 (Fig 10) what is done is picking the maximum daily flow for each year, which does not necessarily correspond to a flood. Similarly, flood events can be expected to last several days given the size of the catchment so I don't think that the max daily discharge can be called a flood event.

**AC4.14** Thank you for this comment. We agree with the reviewer's comment: we analyze the annual maxima of daily flows and they do not always correspond to floods. We will specify in the article that, for this part, we use the annual discharge maxima as flood proxy indicators and replace the term "Flood events" with "Annual discharge maxima" in Fig. 10.

**RC4.15** In 5.4 (Fig 11), I suppose that the "flood activity" is defined as the number of days over the threshold, which again does not necessarily correspond to single flood events. The way of calculating the thresholds should also be precised (p 21, I 373-374 : are they defined through a flood frequency analysis as the value of the flood return period 1 year, or from the flow duration curve ? Therefore I would be more cautious with the term "flood" and would preferrably use terms like high flow indicators or maybe flood proxy indicators.

**AC4.15** For flood activity, the threshold was calculated by considering the 90 largest flood events over a 90-year period. If the threshold was exceeded on several consecutive days, only the date of the maximum annual discharge over a 5-day window was retained. In this analysis, several flood events could be identified in the same year. Here we come close to the definition of "flood" referred to by the reviewer.

For low flow activity, the threshold was calculated by considering the median of the 90 minimum annual reference flows at a monthly time step. Here again, several low flow sequences could be identified in a single year.

These points will be clarified in the text.

**RC4.16** p 26, I 483-488 : the unrealistic simulated discharges obtained with altitude-elevation correction can also be due to the calibration of the hydrological model, considering in particular that the model was "high flow calibrated" (4.1.2). A rigorous way to test that would be to re-calibrate the hydrological model with the new corrections added to the observations before testing them on the downscaling models.

**AC4.16** Thank you for this comment. A recalibration would obviously be interesting, but based on previous analyses, we argue that it would not allow to fix this issue in a relevant way.

As mentioned in the article, if an elevation-precipitation relationship can be identified from aggregated (e.g. annual) precipitation data, this is not the case at the scale of individual events. This is illustrated by the daily rainfall amounts observed at the different stations in the catchment area for the 4 largest floods recorded during the period. As shown in Fig. A below, there is no dependency on elevation. For event 2, precipitations at the highest elevation are even lower than in the lowlands.

As shown in Fig. B below, a precipitation lapse rate of 10%/100m significantly increases the precipitation amounts, even for the heaviest precipitation events. On average, the annual MAP maxima at the Upper Rhône River basin scale are increased by 20%. Due to the high nonlinearity of the production processes, this 20% increase leads to a much higher increase in peak flows. A recalibration of the model would reduce the "productivity" of the catchments, i.e. it would increase the storage capacity of the soil reservoir. The filling rate of this reservoir determines the runoff coefficient of the catchment for that period, and the lower the capacity, the easier it is to fill the reservoir. To avoid huge and unrealistic flo

---

## Author Response (AR1)

We thank the 4 referees for reading the manuscript carefully and providing thoughtful and constructive comments. The comments are noted with RC, our responses with AC, and the additions or changes in the manuscript are underlined. We made significant modifications to our initial work to account for both comments and suggestions of each referee. They allowed strengthening the core of the analysis we presented in the former version of the manuscript.

**Response to Anonymous Referee #1**

General comments

**RC1.1** This study evaluates two approaches for simulating over one century-long daily streamflow at different Upper Rhône River catchment locations. The analysis examines the potential of atmospheric reanalyses simulations (ERA-20C), conceptual glacier-hydrological model and two downscaling approaches for providing temporal variations of river discharges, low flow sequences and flood events. The results indicate that bias correction is crucial for both precipitation and air temperature and both downscaling models. While the observed multi-scale variations of discharges (daily, seasonal and inter-annual) are well reproduced, the results for low flows are less satisfactory.

The manuscript is within the scope of the journal and has a good structure. Still, the novel scientific contribution can be better formulated. The Introduction says that the comparison and evaluation of different downscaling approaches have been the subject of multiple previous studies, so it is not clear what are still the research gaps and how this manuscript contributes to some novel findings/knowledge. The manuscript will also benefit from some more detailed justification of why such one-century runs are needed and/or beneficial.

> **AC1.1** Thank you for this comment. This point was also highlighted by Referee #3 (RC3.1) and Referee #4 (RC4.1).
>
> A large number of studies have set up modeling chains to simulate river discharges in response to large-scale atmospheric trajectories over specific periods. As shown by Wood et al. (2004) and Quintana Seguí et al. (2010), for instance, the choice of the downscaling approach can strongly influence the simulation results. Not all downscaling approaches are necessarily relevant for the targeted simulations, but impact-oriented assessments can guide the model selection. For climate impact analyses focusing on hydrology, for instance, the modeling chains have to be able to reproduce in a relevant way the multi-scale hydrological variations that result from the large-scale atmospheric trajectories of the considered period (e.g. Lafaysse et al., 2014).
>
> In this work, we assess and compare the ability of two modeling chains to reproduce, from large-scale atmospheric information only, the observed temporal variations in discharges in the Upper Rhône River (URR) catchment.
>
> We combine three innovative features. i) We evaluate the modeling chains in contrasting and demanding hydro-meteorological configurations where the interplay between weather variables, both in space and time, is determinant. The URR catchment, a mesoscale alpine catchment straddling France and Switzerland, indeed presents a number of different hydrological regimes, ranging from highly glaciated ones in its upper part to mixed ones dominated by snow and rain downstream. ii) We evaluate the modeling chains over the entire

20th century, a period long enough to assess the ability of the modeling chains to reproduce daily variations in observed discharges, low frequency events (low flow sequences and annual discharge maxima), and variations in low flow and flood activities. iii) For both downscaling models, we evaluate the need for additional bias correction prior to application of the hydrological model for precipitation, temperature and temperature lapse rates.

Overall, we confirm the interest of such modeling chains, provided that additional bias correction of atmospheric variables is applied to downscaled weather scenarios prior to their use in the hydrological model.

The novelty of our study is now described in the introduction to the revised version of the article. It is also mentioned in the conclusion.

A discussion of the value of the proposed approach with some alternative approaches (e.g. stochastic weather generator) will be useful as well.

Thank you for this comment. Yes, we could use a stochastic weather generator (WGEN). We have currently developed a multi-site WGEN for the simulation of long time series (> 10,000 years) of weather scenarios for the large Aare catchment in Switzerland (GWEX). The entire simulation framework is described in the article by Viviroli et al. (2022) published last year in HESS.

However, WGENs are developed to generate plausible weather sequences. They do not aim to reproduce the spatio-temporal variations in weather observed in a given area. The main reason for this is that they are not forced by variations in large-scale atmospheric information. Unlike WGENs, downscaling models are. This is an important advantage of downscaling models when scenarios have to be generated for other climate contexts, where changes in the circulation and state of the atmosphere are likely to lead to changes in local weather conditions.

We agree that this point could be clarified in the manuscript, although it is not a major issue here. We have, however, added a rather large amount of new material to account for a number of comments made by the 4 reviewers. We have therefore not discussed this point in the revised version of the article.

The proposed modelling chains are quite complex, so a rigorous description of the methods is challenging, resulting in difficulty in reproducing the proposed experiments exactly.

We recognize that the modeling chains are quite complex. These modeling chains are quite common when weather conditions are produced from downscaling models for climate change impact studies. We tried to give as much detail as possible, and where this was not possible, we provided references to articles in which the methods are described in more detail. We hope the reader will feel comfortable with the actual details of the description.

For ease of understanding, we have included in the revised version of the article, at the beginning of the Methods section, a synthetic summary of the modeling chains.

**RC1.2** My second suggestion is to add more process-related interpretations of results and findings. For example, to discuss and present which processes lead to low flows and floods in the study area and to

demonstrate that the suggested chains represent well such runoff generation processes in the (sub) catchments. For example, I'm not sure whether individual correction of the bias for air temperature and precipitation does not result in some artificial combination which can affect, e.g. low flow simulations of the model. If the low flows occur in winter, the impact of air temperature bias correction will be more important than precipitation corrections or vice versa. So it will be interesting to provide more process interpretations in the results.

AC1.2 Thank you for this very interesting suggestion and yes, you are right.

For flood and low flow events, the added value of a bias correction for weather scenarios is important (Fig. 11). Depending on the hydrological variable considered, the added value of a temperature bias correction is not necessarily equivalent to that of a precipitation bias correction. In the upstream parts of the URR catchment, for instance, low flows occur mainly in winter, due to the low temperatures in this season. The quality of simulated winter low flows thus depends to a great extent on the quality of temperature scenarios, and much less on the quality of precipitation scenarios. Conversely to precipitation corrections, temperatures corrections therefore lead to a significant improvement in low flow simulations. This is illustrated by the additional analyses presented in the discussion (Sect. 6.2).

Note also that winter low flows are expected to mainly come from the discharge contributions of low elevations areas. As most temperature stations are located at low elevations, the influence of temperature bias on low flows is more related to the bias in the mean temperature than to the bias in the temperature lapse rate. This is clear for high elevations sub-basins (see Fig. 13 and 14c for Rhône@Porte-du-Scex). It is also observed in the lower part of the Upper Rhône River catchment, but this is less evident.

With regard to its influence on floods, a similar discussion can be proposed for the precipitation lapse rate. Increased precipitation leads to increased snowfall in winter, followed by increased snowmelt and runoff in spring. Snowmelt floods are more or less proportional. The impact on fall discharges is different. Larger precipitation directly leads to larger floods during this season. The increase in flood intensity is expected to be much larger than the increase in precipitation amount, due to the non-linear runoff generation process that increases with precipitation intensity.

If less evident, a biased temperature lapse rate may also give a poor simulation of flood regimes. In the URR catchment, the largest floods often occur in fall due to large precipitation amounts. In fall, the temperatures can be low enough for precipitation to fall as snow in high elevation areas. Such situations lead to "reduced floods" compared to floods that would have occurred if all the precipitation had fallen as rain. This situation was observed, for instance, during the flood of October 15, 2000, making the flood damage in the region much smaller than expected (Hingray et al., 2010). The temperature lapse rate is determinant, as it defines the elevation of the snowfall/rainfall limit and therefore the "effective area" of the catchment for these events. The overestimated lapse rate in the MAR model, which results in too cold weather in high elevations areas, is thus expected to lead to lower intensity floods in fall. This is illustrated by the differences between the two bias-corrected experiments produced with MAR in Fig. 13. When MAR is not corrected for the lapse rate, the intensity of fall floods is significantly lower than when it is.

We have added comments on these points in the Results and the Discussion sections.

**RC1.3** My third comment is on the validation of the results. It will be interesting to see how the procedures work in some independent (validation) time periods.

**AC1.3** Thank you for this important comment. It would obviously be important to assess it.

Because of the data context, it is not always easy to assess the relevance of the calibration. For sub-basins where concomitant weather and "natural" discharge data are available over a sufficiently long period, this assessment can be made with a classical split-sample test. The split-sample tests carried out by Schaefli et al. (2005) for three URR sub-basins show that the classical calibration is efficient and robust in this context.

For sub-basins for which the hydrological behavior is significantly altered by dams, the split-sample test is not possible, as the data of the entire period have to be considered for the signature-based calibration. The analyses described below nevertheless suggest that the signature-based calibration remains efficient and robust in our context.

To assess this, we recalibrated the parameters of the four URR sub-basins with a natural (or at least not significantly altered) hydrological regime, using only the hydrological signatures. In a first analysis, the signatures were estimated using the same data (period P0) than those considered for the classical calibration (see Fig. S5 in Supplementary Materials). The simulated time series remain in good agreement with the observed ones. The NSE coefficients are logically lower than those obtained with the classical calibration approach, but the differences are quite small (see Fig. S6 in Supplementary Materials).

In a second analysis, we carried out a similar split-sample test for the signature-based calibration. To do this, we split the period into two (sub-periods P1 and P2), we recalibrated the parameters using only the hydrological signatures of period P1 with the weather data from P1 and, with this set of parameters, we simulated over period P2 the discharge time series from the weather data of P2. The simulated time series remain in good agreement with those observed. They are also in good agreement with those obtained with a signature-based calibration when signatures are derived from the entire period P0. The NSE coefficients are logically lower, but the differences are quite small (see Fig. S7 in Supplementary Materials).

We have added these points in the Methods section and Fig. S5, S6, S7 in Supplementary Materials.

I wonder what is the relative contribution of each individual member of the proposed chain on the final results.

Thank you for this comment. The main objective of our article is to assess the ability of two downscaling models, SCAMP and MAR, to produce hydrologically relevant precipitation and temperature scenarios from large-scale atmospheric information only. For this evaluation, we compare hydrological scenarios simulated from weather scenarios with reference hydrological scenarios also obtained by simulation, but from observed weather.

> In other words, to evaluate weather scenarios in relation to observed weather, we use a filter, the hydrological filter of the considered URR basin. The interest of this filter is that it takes the hydrological point of view, which is a very demanding point of view due to the highly non-linear nature of many hydrological processes.
>
> We therefore do not seek to estimate the relative contribution of each individual member of the simulation chain to the final results.

Specific comments

**RC1.4** Abstract: Please make the description of the results consistent with the conclusions. In the abstracts is written: "… the hydrological situations of low frequency (low flow sequences and flood events) are reasonably well reproduced." However in the conclusions is written: "The results for low flow activity are less satisfactory."

> **AC1.4** Thank you for this comment. We have modified the abstract accordingly.

**RC1.5** Figure captions. The abbreviations used in the figures are not always explained in the captions, so it is sometimes difficult to understand them (without a detailed reading of the text).

> **AC1.5** The following abbreviations are now explained in the captions: URR, P, T, MAP, MAT, Q, obs, ref, MAM, JJA, SON, DJF, BC, MAR, MAR-BC, SCAMP, SCAMP-BC, GSM-SOCONT.

**References**

Hingray, B., Schaefli, B., Mezghani, A., and Hamdi, Y.: Signature-based model calibration for hydrological prediction in mesoscale Alpine catchments, Hydrolog. Sci. J., 55, 1002–1016, https://doi.org/10.1080/02626667.2010.505572, 2010.

Quintana Seguí, P., Ribes, A., Martin, E., Habets, F., and Boé, J.: Comparison of three downscaling methods in simulating the impact of climate change on the hydrology of Mediterranean basins, J. Hydrol., 383, 111–124, https://doi.org/10.1016/j.jhydrol.2009.09.050, 2010.

Schaefli, B., Hingray, B., Niggli, M., and Musy, A.: A conceptual glacio-hydrological model for high mountainous catchments, Hydrol. Earth Syst. Sci., 9, 95–109, https://doi.org/10.5194/hess-9-95-2005, 2005.

Viviroli, D., Sikorska-Senoner, A. E., Evin, G., Staudinger, M., Kauzlaric, M., Chardon, J., Favre, A.-C., Hingray, B., Nicolet, G., Raynaud, D., Seibert, J., Weingartner, R., and Whealton, C.: Comprehensive space–time hydrometeorological simulations for estimating very rare floods at multiple sites in a large river basin, Nat. Hazards Earth Syst. Sci., 22, 2891–2920, https://doi.org/10.5194/nhess-22-2891-2022, 2022.

Wood, A. W., Leung, L. R., Sridhar, V., and Lettenmaier, D. P.: Hydrologic Implications of Dynamical and Statistical Approaches to Downscaling Climate Model Outputs, Clim. Change, 62, 189–216, https://doi.org/10.1023/B:CLIM.0000013685.99609.9e, 2004.

**Response to Anonymous Referee #2**

**RC2.1** The manuscript addresses the important topic of downscaling and bias-correction for hydrological modeling by applying different configurations of a "simulation chain" over an alpine river over the last century. I found the paper interesting and including meaningful results. Nevertheless, the paper lacks both important details on the hydrological modeling and discussion on the uncertainty related to the hydrological modeling within the applied simulation chains.

> **AC2.1** Thank you for this comment. The conceptual hydrological model is mainly used here as a complex and non-linear filter to assess the ability of the downscaling chains to simulate hydrologically relevant weather scenarios. Note that the hydrological model can be imperfect, as it can also be used to produce the hydrological reference against which the hydrological scenarios will be compared. The hydrological model is used as a filter to assess the relevance of the downscaling models.
>
> As mentioned in our article, downscaling models are forced with time series of large-scale atmospheric circulation. Perfect downscaling models would allow to perfectly reproduce the multi-scale spatio-temporal dynamics of weather over the time period used for the simulation. Downscaling models can be assessed, at several scales, for their ability to reproduce observed spatio-temporal dynamics of weather variables. This assessment is rather difficult, especially with regard to the covariability between weather conditions in different sub-basins of the Upper Rhône River (URR) basin, or with regard to the covariability between precipitation and temperature.
>
> The hydrological filter is a powerful alternative filter for making this assessment, as the hydrological behavior of sub-basins is highly non-linear, and depends on all the covariability and dependency structures of meteorological conditions, both spatially and between variables. In brief, we need to assess the meteorological variability and covariability features that are important for hydrology and the hydrological model does this job well.
>
> This impact-oriented evaluation of downscaled weather scenarios is not very widespread. It is, however, very informative. We have mentioned this point in the Introduction. We have also added a section to the discussion to address this point.

Regarding the hydrological modeling methodology:

**RC2.2** L118: please consider adding a diagram of the GSM-SOCONT model in the paper or at least in the Appendix.

> **AC2.2** A scheme of the glacio-hydrological model GSM-SOCONT has been added as an appendix (Fig. S1).

**RC2.3** L122: are these "ice-covered" and "ice-free" parts of the studied catchments dynamics in time? Indeed, the ice-covered part of the upper catchments may have significantly changed over the last century.

> **AC2.3** No, the hydrological model GSM-SOCONT assumes that glacier extent does not vary over time. We recognize that this is a strong assumption.

According to Huss (2011), however, the contribution of glacier melt in the region was relatively stable over the 20th century, with a similar glacier contribution during the periods 1961-1990 and 1908-2008. The glacier retreat strengthened over the period 1988-2008, but the corresponding increase in the glacier contribution to the URR discharges was found to be rather limited (13% in August). While not fully satisfactory, the assumption of a constant glacier therefore seems reasonable.

We have added this point in the discussion.

**RC2.4** L212: what are these additional criteria considering streamflow availability?

**AC2.4** We guess you are referring to line 121 and not line 212. The entire catchment was divided into 18 sub-basins. They were selected so that they are roughly the same size, and so that a gauging station is located at the outlet of a sub-basin wherever possible (see Fig. 1).

We have clarified this point in the text.

**RC2.5** L124 to 126: the calculation of the potential evapotranspiration (PE) time series needs to be more deeply presented in the paper: which CRU dataset has been used for this calculation, what is the spatial resolution of this CRU dataset, and how relevant is this database over this region and in this hydroclimatic context? What about this regional temperature-PE relationship in a non-stationary context?

**AC2.5** We acknowledge that estimating potential evapotranspiration (PET) in such a mountainous area is a critical issue, and that the CRU data set may give a rough and poor estimate of this variable. However, to the best of our knowledge, PET observations are very scarce in the region, making an evaluation of CRU products almost impossible.

To develop our PET-T model, we use monthly CRU values of PET and temperature T at a 0.5° spatial resolution for the 1900-2010 period. PET is assumed to be a linear function of temperature T: PET(t) = a x T(t) + b. The coefficient b depends on the calendar month. This model, estimated for the entire URR catchment, is then used at a daily time step to produce daily PET for all its hydrological units. This relationship allows us to estimate PET for high elevation areas. The PET values of the CRU were calculated from a variant of the Penman-Monteith formula (Harris et al., 2014).

We assume that this relationship is the same for the entire period. Variations may have occurred during the period, for instance due to changes in land use and vegetation cover. This will be worth dedicated investigations in future works.

The PET estimation has been clarified in the text.

**RC2.6** L147 to 148: the routing part of the hydrological model needs to be more deeply presented in the paper. What are the hypotheses? How many parameters are devoted to the routing part of the model?

**AC2.6** Thank you for this comment. Our formulation was not appropriate. We did not use a routing module. The discharge at a given point in the catchment is simply the sum of the discharges in the upstream sub-basins. In previous work, we used a Muskingum routing model (Hingray et al., 2010), but it was not really necessary here. The size of the catchment is not very large and we use a daily time step for hydrological simulations.

We have changed the formulation in the text.

**RC2.7** L149 to 151: please state in the paper the total number of parameters that needs to be calibrated for each sub-catchment and add a list of these parameter in the paper (in the Appendix?).

**AC2.7** Please note that the number of parameters (and their names) to be calibrated is already mentioned on lines 149 to 151 (lines 171 to 173 in the revised version of the article).

**RC2.8** L183 to 184: please give more details on the regionalization procedure applied, and list the ungauged catchment studied here.

**AC2.8** Thank you for this comment. All the ungauged sub-basins located upstream of a given gauging station were calibrated at the same time and forced to share the same parameters set. The discharge time series used for the calibration is the time series simulated with this multiple sub-basins configuration at the downstream gauging station, where observations are available. The ungauged sub-basins can be seen in Fig. 1. The sub-basins grouped together for the calibration of their parameters are also listed in Table S2 in Supplementary Materials.

We have clarified this point in the text.

Regarding the discussion about the hydrological modelling uncertainty within the applied chains:

**RC2.9** L172 to 174: *"For gauged catchments for which the hydrological behavior is significantly altered over P1 and for which "natural" flow observations are available prior to 1950, parameters were estimated based on hydrological signatures (Sivapalan et al., 2003; Winsemius et al., 2009). In the present case, parameters are calibrated so that simulated signatures reproduce at best observed ones but observed and simulated signatures come from different periods following Hingray et al. (2010) (e.g. 1961-2015 and 1922-1963 respectively for the Viège basin). We thus assume that the weather regimes and the natural hydrological behavior of the catchment have not significantly changed over the last century, which seems a reasonable assumption to make in first approximation."* This strong hypothesis needs to be more deeply investigated in the paper: what about potential significant interannual / decadal hydro-climatic variability in the region? What about potential impacts of such interannual / decadal hydro-climatic variability on the hydrological model parameters? What about using adjustment on long-term climatic information (e.g. Nicolle et al., 2013, doi: 10.1002/2012WR012940)?

**AC2.9** The stationarity assumption is indeed a strong one. Unfortunately, there is no other choice for the calibration of some sub-basins of the URR catchment, due to the flow data available here, which have been strongly perturbed by dams since the 1950s. See the figure below from Hingray et al. (2010).

It is difficult to assess the relevance of the stationarity assumption for hydrological regimes. This assumption implies that the following two independent assumptions are valid: the

stationarity of precipitation regimes and the stationarity of the natural hydrological behavior of the catchment.

Apart from the impact of dams, the hydrological behavior of sub-basins could have changed over the period, due to changes in land cover (glacier and vegetation). About glacier retreat over the last century, Huss (2011) explains that (i) the contribution of glacier melt was relatively stable over the 20th century, with similar glacier contribution over the periods 1961-1990 and 1908-2008, and (ii) glacier retreat strengthened over the period 1988-2008, with glacier contributions on the Rhône River increasing by 13% for the month of August. Considering a stationary glacier contribution over the whole period is therefore not fully satisfactory for the last two decades, but acceptable over the whole period. On the other hand, the impact of vegetation changes on hydrology is often difficult to highlight. Moreover, in the region, a large part of the sub-basins is free of vegetation (due to elevation). Assuming a stationary natural hydrological behavior seems thus to be reasonable here (see also our analysis in Fig. S10 in Supplementary Materials).

The stationarity of the precipitation regime may be an issue. Significant interannual variability in annual precipitation generally exists for precipitation. It is much lower when the mean interannual precipitation is considered, as it is the case here (we compare the hydrological signatures of two periods of several decades each). To the best of our knowledge, no significant trends in precipitation have been observed in the region over the last century (Masson and Frei, 2016).

We have mentioned this issue in the discussion.

[Figure]

**Fig. 1** Effects of water works on the URR basin hydrology: (a) dam storage capacity evolution over the 1905–2005 period; (b) and (c) 10, 50 and 90% percentiles of the mean monthly discharge at Porte du Scex for 1905–1955 and 1955–2005, respectively; and (d) annual maximum peak discharge. For 1987, 1993 and 2000 floods: ◆: observed discharge; ☐: reconstructed discharge (from Hingray *et al.*, 2009).

Using adjustment on long-term climatic information.

Using a blending neighbor approach for the calibration, as in Nicolle et al. (2013), is indeed an attractive alternative. However, this would not be straightforward. In the upstream sub-basins of the URR basin, the hydrological regime is mainly "glacial" (dominated by glacier melt, according to the Swiss classification). It shifts to a "glacio-nival" regime (glacier/snow melt), a "nival" regime (dominated by snow) and a "pluvio-nival" regime (dominated by a mixture of snow and precipitation) as we move downstream. The flow signatures (such as the mean interannual cycle) at different locations along the river present clearly different patterns. It would therefore probably be difficult to find neighboring basins to consider as donors of the signatures used for the signature-based calibration.

**RC2.10** L180: Table S1 needs to be presented in the paper and not in the Appendix, and more deeply discussed, since the performance obtained on several subcatchments are poor (e.g. Arve@Genève, BDM). What are the potential reasons for these differences in performance between the studied catchments? Please also consider producing the Figure S2 for every studied catchment in the Appendix.

**AC2.10** Thank you for this suggestion. Table S2 could be moved into the article, but the article is already quite long. We agree that performance is rather poor for some sub-basins. However, data *"quality"* varies considerably from one sub-basin to another. For some of them, daily meteorological data and daily non perturbed discharge data are not available. This requires a signature-based approach which, logically leads to lower *"performance"*.

Even when concomitant series are available, the classical calibration is unfortunately not always very effective. This is indeed the case for the gauging station Arve@Genève, Bout-du-Monde. For this basin, the hydrological regime is much less snow-dominated than elsewhere in the Upper Rhône River catchment. The density of the precipitation network is perhaps not dense enough to provide a good estimate of precipitation inputs.

As suggested by the reviewer, the results of the classical and signature-based calibrations are now given for the other sub-basins in Supplementary Materials (Fig. S3 and S4).

Because of the data context, it is not always easy to assess the relevance of the calibration. For sub-basins where concomitant weather and "natural" discharge data are available over a sufficiently long period, this assessment can be made with a classical split-sample test. The split-sample tests carried out by Schaefli et al. (2005) for three URR sub-basins show that the classical calibration is efficient and robust in this context.

For sub-basins for which the hydrological behavior is significantly altered by dams, the split-sample test is not possible, as the data of the entire period have to be considered for the signature-based calibration. The analyses described below nevertheless suggest that the signature-based calibration remains efficient and robust in our context.

To assess this, we recalibrated the parameters of the four URR sub-basins with a natural (or at least not significantly altered) hydrological regime, using only the hydrological signatures. In a first analysis, the signatures were estimated using the same data (period P0) than those considered for the classical calibration (see Fig. S5 in Supplementary Materials). The simulated time series remain in good agreement with the observed ones. The NSE coefficients are logically lower than those obtained with the classical calibration approach, but the differences are quite small (see Fig. S6 in Supplementary Materials).

In a second analysis, we carried out a similar split-sample test for the signature-based calibration. To do this, we split the period into two (sub-periods P1 and P2), we recalibrated the parameters using only the hydrological signatures of period P1 with the weather data from P1 and, with this set of parameters, we simulated over period P2 the discharge time series from the weather data of P2. The simulated time series remain in good agreement with those observed. They are also in good agreement with those obtained with a signature-based calibration when signatures are derived from the entire period P0. The NSE coefficients are logically lower, but the differences are quite small (see Fig. S7 in Supplementary Materials).

We have added these points in the Methods section and Fig. S5, S6, S7 in Supplementary Materials.

**RC2.11** L279 to 284: *"As many sub-basins have altered hydrological regimes, the "hydrological reference" used for the comparison is the discharge time series obtained via hydrological simulation with the "observed weather" as input. For some upstream sub-basins, which hydrological behavior can be considered as roughly natural, the evaluation could also rely on a comparison with discharge observations. We however choose to use the simulated reference. This first makes the evaluation homogeneous for all URR sub-basins. This additionally allows to only focus on the ability of downscaling chains to simulate hydrologically relevant weather scenarios. In other words, this allows to not distort the evaluation by intrinsic errors introduced by the hydrological model."* This point is critical and needs to be more deeply discussed: what about "simulated reference" that are not hydrologically relevant, i.e., highly different from observed streamflows (e.g. Arve@Genève, BDM catchment, with a NSE after calibration equal to 0.44)? What are "intrinsic errors introduced by the hydrological model" in this context?

> **AC2.11** Please see our response to comment RC2.1. The conceptual hydrological model is mainly used here as a very complex and non-linear filter to assess the weather scenarios. Note that the model can be imperfect as we also use it to produce the hydrological reference against which hydrological scenarios derived from weather scenarios will be compared.

**RC2.12** L424 to 429: this delayed dynamics between air temperature, snow accumulation / melting and thus simulated streamflows might be compensated (for good or bad reasons) by the hydrological model parameters during calibration. This point has to be discussed in the paper.

> **AC2.12** Thank you for this comment. We argue that the temperature lapse rate of the hydrological model has to be relevant first. It would not really be satisfactory and relevant to calibrate a model with known biased weather forcings. In this respect, compensation on certain model parameters would not be very relevant either. Despite this, we could indeed imagine that some kind of compensation exists. However, this compensation is unlikely to be large. The reason is due to the following.
>
> Snowfall at a given location in the model is estimated from precipitation and temperature estimated for that location. The temperature threshold parameters of the snow/rain repartition model (below which precipitation is entirely snow, and above which it is entirely rainfall) are fixed. They are not calibrated. They were estimated from weather stations throughout Switzerland, where observations of the nature of precipitation (rain, snow or mixed precipitation) were estimated each day at 7:30 am, 1:30 pm and 7:30 pm by a MeteoSwiss operator (Hingray et al., 2010; Froidurot et al., 2014). The two temperature thresholds of the snow/rain distribution model are the same everywhere, for high altitudes, low altitudes, etc.
>
> The same applies to snowmelt. The snowmelt threshold is not calibrated (it is set at 0°C). A negatively biased temperature lapse rate will lead to a rather good snow/rain repartition at mid-elevations, but to an overestimation of snowfall (and snowpack size/duration) at high elevations, and an underestimation of snowfall (and snowpack size/duration) at low elevations. A biased temperature lapse rate would therefore lead to an irrelevant simulation of altitudinal dependency of the snowpack dynamics.

This is not expected to be easily compensated in the model. For these reasons, but also for reasons of space limitation, we have not discussed this point in the article.

**RC2.13** L440 to 450: what are the potential impacts of these changes (on the air temperature lapse rate) on the calibration of the hydrological model parameters? Did you try to do another calibration of the model considering these changes? Please discuss this point in the paper.

**AC2.13** Please see our response to comment RC2.12.

**RC2.14** L461 to 466: what are the potential impacts of these changes (on the orographic precipitation enhancement) on the calibration of the hydrological model parameters? Did you try to do another calibration of the model considering these changes? Please discuss this point in the paper.

**AC2.14** Thank you for this comment. A recalibration would obviously be interesting, but based on previous analyses, we argue that it would not allow to fix this issue in a relevant way.

As mentioned in the article, if a precipitation-elevation relationship can be identified from aggregated (e.g. annual) precipitation data, this is not the case at the scale of individual events. This is illustrated by the daily rainfall amounts observed at the different stations in the catchment area for the 4 largest floods recorded during the period. As shown in the figure below, there is no dependency on elevation. For event 2, precipitations at the highest elevation are even lower than in the lowlands.

[Figure]

As shown in the figure below, a precipitation lapse rate of 10 %/100 m significantly increases the precipitation amounts, even for the heaviest precipitation events. On average, the annual MAP maxima at the URR basin scale are increased by 20 %. Due to the high non-linearity of the production processes, this 20 % increase leads to a much higher increase in peak flows. A recalibration of the model would reduce the "productivity" of the catchments, i.e. it would increase the storage capacity of the soil reservoir. The filling rate of this reservoir determines the runoff coefficient of the catchment for that period, and the lower the capacity, the easier it is to fill the reservoir. To avoid huge and unrealistic floods in a +10 %/100 m lapse rate configuration, a much higher soil capacity is therefore required. Previous analyses show, however, that this can be detrimental for smaller (and more numerous) rainfall-runoff events, leading to a number of minor floods being underestimated.

MAP observed at D for annual discharge maxima

**RC2.15** L481 to 488: did you check these hypotheses by comparing simulated streamflows with observed ones (and not "simulated references")?

**AC2.15** No, and we acknowledge that this is not an ideal configuration, but unfortunately we did not find a better way to carry out the evaluation. Ideally, the reference should be the observations. However, for the 1961-2009 period, observed discharges are significantly perturbed by dams (most of which were built in the 1950s). See the figure from Hingray et al. (2010) in answer AC2.9. Comparing simulated discharges from meteorological scenarios with observed discharges is therefore meaningless. The observation was thus replaced by a proxy: the discharges simulated from observed weather.

Other specific comments:

**RC2.16** L90: what is a "Binn-Simplon" situation?

**AC2.16** For the URR catchment, the highest regional precipitation amounts are induced by the "Binn-Simplon" weather situations, where warm and humid air masses from the Mediterranean Sea cross the southern Alps (OFEG, 2002). These situations present a typical spatial pattern, with higher precipitation amounts in the southern and eastern parts of the catchment.

We have added the following reference:
OFEG: Les crues 2000 - Analyse des événements/cas exemplaires, Rapports de l'OFEG, Série Eaux, 2, Office Fédéral des Eaux et de la Géologie, Berne, Suisse, 2002.

**RC2.17** L92: what is a "retour d'Est" situation?

**AC2.17** The "retour d'Est" situations are similar to the "Binn-Simplon" ones, but the warm and moist air fluxes from the Mediterranean Sea are finally oriented eastwards, from Italy towards France. They mainly affect the Piémont in Italy and the French Alps behind the border, resulting in heavy to very heavy precipitation amounts one to three times a year in these regions, particularly in the east-facing massifs.

We have added the following reference:
Metzger, A.: Retour d'est - La géochronique du temps qu'il fait, La Géographie, 1588, 64–65, https://doi.org/10.3917/geo.1588.0064, 2023.

Please see also:
http://pluiesextremes.meteo.fr/france-metropole/Retours-d-est-sur-le-Queyras.html

**RC2.18** L114 : please consider highlighting the spatial resolution of the ERA-20C dataset on Figure 1 (in the topleft panel?)

**AC2.18** In our opinion, the addition of the ERA-20C grid would render the topleft panel of Fig. 1 rather illegible. The spatial resolution of the ERA-20C reanalysis is indicated on line 114 (line 121 in the revised version of the article).

We have added in the text that the URR catchment is covered by 8 ERA-20C grid points.

**RC2.19** L213: please consider highlighting the spatial resolution of the MAR model on Figure 1 (in the topleft panel?)

**AC2.19** The same issue arises as above. Adding the MAR grid would make the Fig. 1 rather illegible. The spatial resolution of the MAR model and the number of grid points covering the URR catchment are indicated on lines 212 to 215 (lines 256 to 257 in the revised version of the article).

**RC2.20** L240: how many previous days are considered? Please add details on this point.

**AC2.20** One previous day is considered for the pairing. This point has been specified.

**RC2.21** L243: please details what is a "non-parametric method of fragment" in this context.

**AC2.21** The method of fragment is a non-parametric method. Whatever the context, the observed spatio-temporal structure of the observed variable is used as the structure of the day for which the structure is unknown and required. Here, the structure is defined by the spatial pattern of precipitation observed from precipitation stations. This structure is used to determine how the mean areal precipitation amount of a given day is distributed in space for that day. Mathematically, the observed precipitation of each station is multiplied by a

coefficient MAP*/MAPa, where MAP* is the mean areal precipitation at the regional scale for the target day in the scenario and MAPa is the mean areal precipitation at the regional scale for the day used as a reference for the structure. The reference day can be selected on the basis of an analogy criterion (see Mezghani and Hingray, 2009, for an example).

A short clarification has been given in the text.

**RC2.22** L336: the comparison of the Figure 8 and 9 is not easy: please consider adding another figure that present flow differences on each catchment and configuration to ease the comparison.

**AC2.22** We acknowledge that the comparison is not easy. For reasons of space, however, we cannot add a figure showing the differences.

To show the performance gain associated with bias correction, we have added a Table with the NSE coefficients for the dynamical downscaling model (MAR/MAR-BC) and the statistical index CRPSS (Continuous Ranked Probability Skill Score) for the statistical downscaling model (SCAMP/SCAMP-BC). This is Table 2.

**RC2.23** Figure 8: please add the observed flows when available. What is the plateau observed for the "reference" streamflow series of the Rhône@Geneve in 07/1982 (and for other dates)?

**AC2.23** Unfortunately, the comparison with observed flows is meaningless for this period because i) the hydrological regime of the URR catchment has been altered by dams since the 1950s, and ii) the aim of our study is to simulate flows in a natural hydrological regime.

As mentioned in the article, the regulation rules of the lake are specified in the 1997 settlement. One of the main objectives is that the flow at Genève, Halle-de-l'Ile must not exceed 550 m3 s-1 except during high-water periods in the lake Geneva. The plateau observed for the "reference" streamflow series of the gauging station Rhône@ Genève, Halle-de-l'Ile in 07/1982 corresponds to the 550 m3 s-1 threshold imposed in our simplified modeling of the behavior of Lake Geneva. The other plateaus correspond to the environmental low flow to be satisfied downstream (100 m3 s-1 from May 1st to September 30th and 50 m3 s-1 from October 1st to April 30th). We make the simulated water lake rather far from the observations. However, it is not possible to make a better simulation because the precise operations are not known (they follow daily variations in electricity demand in the region and its price).

**RC2.24** L398 to 402: please add some references on the use of ERA-20C for hydrology or other references related to this potential ERA-20c drawback.

**AC2.24** As recommended, we have included further references to hydrological studies based on ERA-20C, in Sect. 6.1 (where limitations related to ERA-20C are discussed).

**References**

Froidurot, S., Zin, I., Hingray, B., and Gautheron, A.: Sensitivity of Precipitation Phase over the Swiss Alps to Different Meteorological Variables, J. Hydrometeorol., 15, 685–696, https://doi.org/10.1175/JHM-D-13-073.1, 2014.

Harris, I., Jones, P., Osborn, T., and Lister, D.: Updated high-resolution grids of monthly climatic observations - the CRU TS3.10 Dataset, Int. J. Climatol., 34, 623–642, https://doi.org/10.1002/joc.3711, 2014.

Hingray, B., Schaefli, B., Mezghani, A., and Hamdi, Y.: Signature-based model calibration for hydrological prediction in mesoscale Alpine catchments, Hydrolog. Sci. J., 55, 1002–1016, https://doi.org/10.1080/02626667.2010.505572, 2010.

Huss, M.: Present and future contribution of glacier storage change to runoff from macroscale drainage basins in Europe, Water Resour. Res., 47, W07 511, https://doi.org/10.1029/2010WR010299, 2011.

Masson, D. and Frei, C.: Long-term variations and trends of mesoscale precipitation in the Alps: recalculation and update for 1901-2008, Int. J. Climatol., 36, 492–500, https://doi.org/10.1002/joc.4343, 2016.

Metzger, A.: Retour d'est - La géochronique du temps qu'il fait, La Géographie, 1588, 64–65, https://doi.org/10.3917/geo.1588.0064, 2023.

Mezghani, A. and Hingray, B.: A combined downscaling-disaggregation weather generator for stochastic generation of multisite hourly weather variables over complex terrain: Development and multi-scale validation for the Upper Rhone River basin, J. Hydrol., 377, 245-260, https://doi.org/10.1016/j.jhydrol.2009.08.033, 2009.

OFEG: Les crues 2000 - Analyse des événements/cas exemplaires. Rapports de l'OFEG. Série Eaux, n°2, Office Fédéral des Eaux et de la Géologie, Berne, Suisse, 2002.

**Response to Anonymous Referee #3**

This is a very well written paper (with high quality figures) on discharge simulations based on climate model outputs, with "a typical simulations chain" (abstract first line). The paper was easy to read, which is a great achievement for such a complex modelling paper.

We thank Referee #3 for his/her encouraging general comments.

**RC3.1** The topic is probably undersold: it is not just a case study but one of the few attempts to produce long simulations of the past, with a high number of challenges to overcome. I am not aware of long simulation studies back in the past in mountainous environments with climate model outputs (I might not be on top of that literature but I suspect there are very few).

Accordingly, the added value to the literature should be formulated in a much more straightforward way as early as possible in the introduction and again in the conclusion.

The method is not well captured in the abstract or in the paper overall and calling this a "typical simulation chain" is perhaps underselling. A critical step in such simulation chains is how to combine reanalysis data with a hydrological model, especially in mountainous areas. What does this study do differently than other works? Why is the approach more interesting?

> **AC3.1** Thank you for this comment. This point was also highlighted by Referee #1 (RC1.1) and Referee #4 (RC4.1).
>
> A large number of studies have set up modeling chains to simulate river discharges in response to large-scale atmospheric trajectories over specific periods. As shown by Wood et al. (2004) and Quintana Seguí et al. (2010), for instance, the choice of the downscaling approach can strongly influence the simulation results. Not all downscaling approaches are necessarily relevant for the targeted simulations, but impact-oriented assessments can guide the model selection. For climate impact analyses focusing on hydrology, for instance, the modeling chains have to be able to reproduce in a relevant way the multi-scale hydrological variations that result from the large-scale atmospheric trajectories of the considered period (e.g. Lafaysse et al., 2014).
>
> In this work, we assess and compare the ability of two modeling chains to reproduce, from large-scale atmospheric information only, the observed temporal variations in discharges in the Upper Rhône River (URR) catchment.
>
> We combine three innovative features. i) We evaluate the modeling chains in contrasting and demanding hydro-meteorological configurations where the interplay between weather variables, both in space and time, is determinant. The URR catchment, a mesoscale alpine catchment straddling France and Switzerland, indeed presents a number of different hydrological regimes, ranging from highly glaciated ones in its upper part to mixed ones dominated by snow and rain downstream. ii) We evaluate the modeling chains over the entire 20th century, a period long enough to assess the ability of the modeling chains to reproduce daily variations in observed discharges, low frequency events (low flow sequences and annual discharge maxima), and variations in low flow and flood activities. iii) For both downscaling

models, we evaluate the need for additional bias correction prior to application of the hydrological model for precipitation, temperature and temperature lapse rates.

The novelty of our study is now described in the revised version of the article.

**RC3.2** A concise summary of the method would help (take a hydro model, calibrate on observed meteo and streamflow data, run with meteo scenarios generated based on reanalysis data; contrary to many other works, no simple spatial downscaling of reanalysis data but use of a weather generator to produce mean areal precipitation scenarios).

> **AC3.2** Thank you for this suggestion. In the first version of the article, each step of the model chain was described, but a general description of the method was clearly missing.
>
> We have included in the revised version of the article, at the beginning of the Methods section, a synthetic summary of the model chains.

Details:

**RC3.3** Method: it should be made very clear at what scale MAP (mean areal precip) and MAT (temperature) are defined. I see from the sentence "bucket-type model that uses time series of Mean Areal Precipitation (MAP) and Temperature (MAT) as inputs for each hydrological unit." that MAP and MAT is defined at the elevation band scale (?)

> **AC3.3** Yes, mean areal precipitation and mean areal temperature are defined at the scale of elevation bands. The mean areal precipitation and temperature time series are different from one elevation band to another, due to the different elevations and Thiessen weights for the neighboring stations.
>
> This point has been clarified in the Methods section.
>
> Note that MAP and MAT are also estimated for other spatial scales. In our evaluations, we compare simulated and observed MAP and MAT at the scale of the 5 major sub-basins shown in Fig. 1.
>
> We have used different terminologies: "mean areal precipitation" and "mean areal temperature" for hydrological units, MAP and MAT for regional or sub-regional scales.

**RC3.4** "a daily potential evapotranspiration time series is derived from temperature", how? Linear?

> **AC3.4** To develop our PET-T model, we use monthly CRU values of PET and temperature T at a 0.5° spatial resolution for the 1900-2010 period. PET is assumed to be a linear function of temperature T: $PET(t) = a \times T(t) + b$. The coefficient b depends on the calendar month. This model, estimated for the entire URR catchment, is then used at a daily time step to produce daily PET for all its hydrological units.
>
> This point has been clarified in the article.

**RC3.5** "For each RHHU, daily MAP and MAT are estimated from neighboring weather stations using the Thiessen's weighting", why Thiessen? Probably very inappropriate for mountain environments? And how is the Thiessen obtained between points (weather station locations) and elevation bands?

> **AC3.5** As shown on the two maps in Fig. A, the Thiessen polygons are derived from the network of available stations in the region (within the catchment and in its neighborhood).

[Figure]

Figure A. Thiessen polygons for (left) the precipitation stations and (right) the temperature stations considered for the study.

> The Thiessen weights of the different stations for any given surface area are estimated as follows. Let's call Ai, the surface area of the considered unit covered by the ith station. For most stations, Ai will be zero. The weight associated to station i is simply Ai / A where A is the surface area of the considered spatial unit. Whatever the spatial unit (elevation band, sub-basin, URR basin, …), the sum of the weights is always equal to one for this unit. The weights are always different from one spatial unit to another, from one elevation band to another, depending on the proximity of the stations to that unit.
>
> Note that the URR basin is divided into 18 sub-basins. For each of them, we always have 5 to 10 stations for which the weights are non-zero for precipitation. If we disregard the issue of elevation dependency, which is widely discussed in the article, we considered that this number provides a reasonable estimate of MAP for these areas. Note that this issue of elevation dependency would not be better addressed by other spatialisation approaches (inverse distance, kriging). This point is now mentioned in the discussion.
>
> For temperature, the number of stations with non-zero weights is lower, but this variable, when considered at a given reference elevation, varies much less spatially. The way temperature is estimated for each unit is as follows: apply a weighted Thiessen sum of station temperatures, corrected in a previous step to correspond to the reference elevation for that unit (thanks to the temperature lapse rate discussed in the article). For most time steps, the linear temperature-elevation relationship is very robust, allowing a very reasonable temperature estimate for altitudes different from those of the stations.

**RC3.6** What is the storage equation in this model? Is it the storage-volume relationship? Where do you get if from?

> **AC3.6** This is the water level-storage equation mentioned in the next sentence.

**RC3.7** How do you construct the linear outflow relationship? As far as I see from one year of observed data (extract from the pdfs https://www.hydrodaten.admin.ch/en/seen-und-fluesse/stations/2028 and https://www.hydrodaten.admin.ch/en/seen-und-fluesse/stations/2606) any outflow is possible for low water levels (see figures below) and a linear relationship only holds for high water levels and flow beyond 500 m3/s

[Figure]

[Figure]

AC3.7 We could not find these figures on the above-mentioned websites. However, the data used for these figures are likely those of the last few years, decades when a regulation applies. In this recent period, the water level-outflow relationship has no reason to be univocal, since the operations for a given day (hour) must be defined according to the state of electricity demand and electricity prices for that day (hour). An exception is made for high water levels, when operations aim to limit the water level in the lake to avoid riparian flooding. In this case, the only decision variable for operation is the lake water level, which explains the univocal relationship found in the data.

> If the lake were unregulated, we would probably have a univocal relationship in first approximation, due to its very large surface area and the topographical configuration of its outlet. The discharge would be a function of the hydraulic head. As the water level does not vary a lot in the lake (much less than 2 m), we considered the relationship was to be linear.

**RC3.8** Can you say something about the evaporation equation, what does it include? I have not seen the Rohwer's equation (Rohwer, 1931) before.

> **AC3.8** We used Rohwer's equation (see equation below) as it uses an additional variable, namely atmospheric pressure at 10 m, and this was found of potential interest for further use of the model in other climate contexts. Note that Rohwer's equation is very close to the better-known Penman (1948) equation (see below), and that the difference of the mean annual evaporated water over the 1961-2010 period between Rohwer's and Penman's estimates is only 3 %. This choice is thus expected to have little impact on the main conclusions of our study.
>
> Rohwer (1931) equation:
> $E(t) = 0.77 \times (1.465 - 0.0186 \times p\_10m(t)) \times (0.44 + 0.118 \times U\_10m(t)) \times (es\_surf(t) - ea\_10m(t))$
> where :
> - $E(t)$ (in j-1) is the evaporation from Lake Geneva at time step t.
> - $p\_10m(t)$ (inHg) is the atmospheric pressure at 10 m at time step t.
> - $U\_10m(t)$ (mph) is the wind speed at 10 m at time step t.
> - $es\_surf(t)$ (inHg) is the vapor pressure of saturated air at lake surface temperature $T\_surf(t)$ at time step t.
> - $ea\_10m(t)$ (inHg) is the current vapor pressure in the air at 10 m at time step t.
>
> Penman (1948) equation:
> $E(t) = 0.35 \times (1 + 0.24 \times U\_10m(t)) \times (es\_surf(t) - ea\_10m(t))$

**RC3.9** "In the present work, SCAMP was used to generate 30 time series of daily spatial weather scenarios for the 1902-2009 period from ERA-20C reanalysis outputs." At what spatial scale? At the scale of the 7 km x 7km of the MAR model? This would be in contradiction with Fig. 3 that says that you generate MAP (mean areal precip) estimates and not first spatial estimates? So, do you first produce spatial estimates, i.e. daily time series per pixel? If yes, how do you combine the pixels into MAP estimates? By taking the pixels within the elevation band? Does this make any sense?

> **AC3.9** Thank you for this comment. There was indeed a mistake in Fig. 6, where disaggregated data are shown for MAR and SCAMP. SCAMP indeed produces precipitation and temperature time series at station level, and the stations at the end of the process are exactly the same as the observation stations. Sorry for this. We understand it was almost impossible to understand the spatial aggregation/disaggregation process. We hope it is now clear.
>
> Fig. 6 has been corrected: the spatial resolution of P/T is "stations'' for obs and SCAMP, and "7x7 km" for MAR.

**RC3.10** Method overall: how do you have confidence that the conceptual model calibrated on observed streamflow data with observed station meteo does a good job if used with weather generator scenarios produced at a different scale? What part of your method ensures this?

**AC3.10** Thank you for this comment. The conceptual hydrological model is mainly used here as a complex and non-linear filter to assess the weather scenarios. Note that the model can be imperfect as we also use it to produce the hydrological reference against which hydrological scenarios derived from weather scenarios will be compared. The hydrological model is used as a filter to assess the relevance of the downscaling models.

As mentioned in our article, downscaling models are forced with time series of large-scale atmospheric circulation. Perfect downscaling models would allow to perfectly reproduce the multi-scale spatio-temporal dynamics of weather over the time period used for the simulation. Downscaling models can be assessed, at several scales, for their ability to reproduce observed spatio-temporal dynamics of weather variables. This assessment is rather difficult, especially with regard to the covariability between weather conditions in different sub-basins of the URR basin, or with regard to the covariability between precipitation and temperature.

The hydrological filter is a powerful alternative filter for making this assessment, as the hydrological behavior of sub-basins is highly non-linear, and depends on all the covariability and dependency structures of meteorological conditions, both spatially and between variables. In short, we need to assess the meteorological variability and covariability features that are important for hydrology and the hydrological model does this job well.

This impact-oriented evaluation of downscaled weather scenarios is not very widespread. It is, however, very informative. We have mentioned this point in the Introduction. We have also added a section to the discussion to address this point.

**RC3.11** Model calibration on signatures: did you check for the catchments with observed concomitant streamflow if the signature calibration gives good results?

**AC3.11** Thank you for this comment. To assess the relevance of the signature-based calibration, we carried out the following experiment. We recalibrated the parameters of the four URR sub-basins with a natural (or at least not significantly altered) hydrological regime, using only the hydrological signatures. The signatures were estimated using the same data (period P0) than those considered for the classical calibration (see Fig. S5 in Supplementary Materials). The simulated time series remain in good agreement with the observed ones. The NSE coefficients are logically lower than those obtained with the classical calibration approach, but the differences are quite small (see Fig. S6 in Supplementary Materials).

We have added this point in the Methods section and Fig. S5 and S6 in Supplementary Materials.

Bias correction

**RC3.12** Do I understand correctly that bias correction is done independently for temperature and precipitation, which are also downscaled independently. How do you ensure a good link between the two for the simulation of snow accumulation and melt?

**AC3.12** Thank you for this comment. Yes, the bias correction is performed independently for temperature and precipitation. The downscaling also appears to be performed independently, but this is not exactly the case. As the downscaling is conditioned each day by the atmospheric

conditions of the day, some dependency between all variables (in space and between temperature and precipitation) is expected to be introduced, as shown by Mezghani and Hingray (2009).

In the work of Mezghani and Hingray (2009), which also applies to the URR catchment, the downscaling model was evaluated in terms of its ability to reproduce - for different spatial and temporal scales - different observed weather statistics: namely, observed MAP, observed MAT and "observed" Mean Areal Liquid Precipitation. The latter depends entirely on the variables MAP and MAT, as MAT defines the nature of precipitation. Mezghani and Hingray (2009) show in Fig. 13 that Mean Areal Liquid Precipitation statistics are well reproduced whatever the catchment. This suggested that a relevant link between precipitation and temperature is indeed derived thanks to the conditioning of the weather generator to large-scale information.

Due to lack of space, we have decided not to comment this point in the manuscript.

Results

**RC3.13** In the MAT series, we seem to see nicely the 1980 global regime shift (Reid et al., 2015), with a sharp increase of temperatures, perhaps worth commenting?

> **AC3.13** The statement describing this shift has been adapted as follows:
>
> "The positive trend in temperature starting in 1980 is also adequately reproduced." replaced by "The positive trend in temperature starting in 1980 is also adequately reproduced, resulting from the global combination of the warming related to anthropogenic greenhouse gases and the reduced anthropogenic aerosol cooling (Reid et al., 2016), especially pronounced over Europe (Nabat et al., 2014)."

**RC3.14** Figure 11c would certainly contain some very interesting information but I cannot see anything. I would split into two figures, and rescale to 200 and 500 m3/s; furthermore, you could perhaps some coefficient of variation and autocorrelation at lag 1 instead of simply the figure?

> **AC3.14** We acknowledge that the figure was difficult to read. We have split it into two figures. Fig. 12c concerns simulations with bias-corrected scenarios and Fig. 12d concerns simulations with raw scenarios. The better agreement with observations when bias-corrected scenarios are used is now clear. On the other hand, we will retain the scale of 0 to 600 m3 s-1, as simulated flows extend beyond 200 and 500 m3 s-1.
>
> Thanks for the suggestion to add the coefficient of variation and the lag-1 autocorrelation. However, this would not allow us to assess the ability of the chains to reproduce the reference temporal variations from one year to another. A more informative numerical criterion for this would be a NSE criterion for deterministic simulations (from MAR) and a probabilistic efficiency criterion (e.g. CRPSS) for ensemble simulations (from SCAMP). For our purpose, however, a graphical comparison of reference and simulated time series is, in our opinion, much more effective in demonstrating the ability of the chains to reproduce reference interannual variations.

**RC3.15** Figure 11b: it is written "The results for low flow activity are less satisfactory, especially for the 1920-1949 sub-period (Fig. 11b). The number of low flow sequences below the threshold in both simulations is indeed twice that of the reference. This suggests a limitation of both downscaling models to simulate long persistent dry sequences." This interpretation is perhaps a bit too limited; most readers might not know that there was a e.g. a heat wave in 1946 and a period of strong melting in the 1940 due to enhanced solar radiation (Huss et al., 2009); there were also some very cold winters in this period and some very warm years (see here https://www.meteoswiss.admin.ch/climate/climate-change.html). I would give some more details here and perhaps use other subperiods to better understand what is going on with the modelling chain? This low flow underestimation is really too striking to not discuss it in far more depth. This is really important because many modelling chains do a rather poor job for snow-influenced low flow but this is rarely discussed. And: it is important to be precise here: is the reference for period 1920-1960 simulated or observed? To understand if the lake management can or cannot explain the low flow differences.

> **AC3.15** Thank you for this comment. Despite some more investigation of this issue, we do not currently have a clear explanation for these results.
>
> We have clarified the text as follows:
>
> The results for low flow activity are less satisfactory (Fig. 12b). On the one hand, the number of mean monthly discharges below the threshold is, whatever the sub-period, significantly higher than that of the reference. It is more than three times higher for the 1920-1949 sub-period. This overestimation is mainly due to longer winter low flow durations (not shown). On the other hand, the variations in low flow activity from one sub-period to another are only partially reproduced. While the small decrease observed between the last two sub-periods is rather well captured, the large increase between the first two sub-periods is fully missed. The reasons for this are unclear. The large differences obtained with the raw and bias-corrected downscaling simulations suggest that some limitations remain in the bias-corrected weather scenarios (e.g. too low simulated winter temperatures). The large differences in the reference low flow activity obtained between the sub-periods also suggest an issue with the stationarity assumption (e.g. hydrological signatures) and/or with the low temporal homogeneity of the data set considered in this work (e.g. the reference discharge time series made from observed/simulated discharges for the beginning/end of the period). These points will be worth further investigations in future works.

**RC3.16** In fact, do I understand correctly that the reference is not always the same for all result plots (which would not be ideal and should be mentioned in all plots, in the caption)? At line 382 following it is written "Simulated year-to-year variations of mean annual discharges are next compared to the "reference" ones over the 1920-2009 period (Fig. 11c). The "references" are observed discharges for the 1920-1960 period and simulated discharges from observed weather for the 1961-2009 period."

> **AC3.16** We acknowledge that this is not an ideal configuration, but unfortunately we did not find a better way to carry out the evaluation. Ideally, the reference should be the observations. However, for the 1961-2009 period, observed discharges are significantly perturbed by dams (most of which were built in the 1950s). See the figure below from Hingray et al. (2010). Comparing simulated discharges from meteorological scenarios with observed discharges is therefore meaningless. The observation was thus replaced by a proxy: the discharges simulated from observed weather. The "references" are observed discharges for the 1920-

1960 period and simulated discharges from observed weather for the 1961-2009 period for Fig. 12 only.

This point has been clarified in the text and specified in the caption to Fig. 12.

[Figure]

Fig. 1 Effects of water works on the URR basin hydrology: (a) dam storage capacity evolution over the 1905–2005 period; (b) and (c) 10, 50 and 90% percentiles of the mean monthly discharge at Porte du Scex for 1905–1955 and 1955–2005, respectively; and (d) annual maximum peak discharge. For 1987, 1993 and 2000 floods: ◆: observed discharge; □: reconstructed discharge (from Hingray et al., 2009).

Discussion

**RC3.17** What could explain a warm bias in the lower atmospheric layers of the model? Besides: would be interesting to see winter and summer lapse rates separately to gain more insight ?

**AC3.17** The warm bias found in the MAR-ERA20C simulation was pointed out in Beaumet et al. (2021). This bias is found mainly during the winter, associated with a too strong lapse rate. It might be linked to both surface and atmospheric processes. Looking for the exact causes of this bias is a complex discussion, and we estimate that it is out of the scope of our article.

**RC3.18** Line 493 following: "Note that in our simulations, the dry bias in the precipitation input is probably corrected via an adjusted parameterization of the hydrological model, (..)". I fully agree with this explanation but I would make it clearer for non experts; now it reads like if someone adjusted it, perhaps say something like: "the dry bias is probably compensated during hydrological parameter optimization; parameter optimization generally results in forcing the model to close the water balance; conceptual reservoir-based models as the one used here can thereby compensate missing rainfall input by lowering the evapotranspiration losses. This problem is well known but very rarely discussed. An example is the work of Minville et al. (2014).

**AC3.18** Thank you for this comment. We have adapted the text to clarify this point.

**References**

Beaumet, J., Ménégoz, M., Morin, S., Gallée, H., Fettweis, X., Six, D., Vincent, C., Wilhelm, B., and Anquetin, S.: Twentieth century temperature and snow cover changes in the French Alps, Reg. Environ. Change, 21, 114, https://doi.org/10.1007/s10113-021-01830-x, 2021.

Hingray, B., Schaefli, B., Mezghani, A., and Hamdi, Y.: Signature-based model calibration for hydrological prediction in mesoscale Alpine catchments, Hydrolog. Sci. J., 55, 1002–1016, https://doi.org/10.1080/02626667.2010.505572, 2010.

Huss, M., Funk, M., and Ohmura, A.: Strong Alpine glacier melt in the 1940s due to enhanced solar radiation, Geophys. Res. Lett., 36, L23501, https://doi.org/10.1029/2009GL040789, 2009.

Mezghani, A. and Hingray, B.: A combined downscaling-disaggregation weather generator for stochastic generation of multisite hourly weather variables over complex terrain: Development and multi-scale validation for the Upper Rhone River basin, J. Hydrol., 377, 245-260, https://doi.org/10.1016/j.jhydrol.2009.08.033, 2009.

Minville, M., Cartier, D., Guay, C., Leclaire, L., Audet, C., Le Digabel, S., and Merleau, J.: Improving process representation in conceptual hydrological model calibration using climate simulations, Water Resour. Res., 50, 5044–5073, https://doi.org/10.1002/2013WR013857, 2014.

Nabat, P., Somot, S., Mallet, M., Sanchez-Lorenzo, A., and Wild, M.: Contribution of anthropogenic sulfate aerosols to the changing Euro-Mediterranean climate since 1980, Geophys. Res. Lett., 41, 5605–5611, https://doi.org/10.1002/2014GL060798, 2014.

Penman H. L.: Natural evaporation from open water, bare soil and grass, Proc. R. Soc. Lond. A, 193, 120-145, https://doi.org/10.1098/rspa.1948.0037, 1948.

Quintana Seguí, P., Ribes, A., Martin, E., Habets, F., and Boé, J.: Comparison of three downscaling methods in simulating the impact of climate change on the hydrology of Mediterranean basins, J. Hydrol., 383, 111–124, https://doi.org/10.1016/j.jhydrol.2009.09.050, 2010.

Reid, P. C., Hari, R. E., Beaugrand, G., Livingstone, D. M., Marty, C., Straile, D., Barichivich, J., Goberville, E., Adrian, R., Aono, Y., Brown, R., Foster, J., Groisman, P., Hélaouët, P., Hsu, H., Kirby, R., Knight, J., Kraberg, A., Li, J., Lo, T., Myneni, R. B., North, R. P., Pounds, J. A., Sparks, T., Stübi, R., Tian, Y., Wiltshire, K. H., Xiao, D., and Zhu, Z.: Global impacts of the 1980s regime shift, Global Change Biol., 22, 682–703, https://doi.org/10.1111/gcb.13106, 2016.

Rohwer, C.: Evaporation from free water surfaces, Technical Bulletin, 271, United States, Department of Agriculture, https://doi.org/DOI:10.22004/ag.econ.163103, 1931.

Wood, A. W., Leung, L. R., Sridhar, V., and Lettenmaier, D. P.: Hydrologic Implications of Dynamical and Statistical Approaches to Downscaling Climate Model Outputs, Clim. Change, 62, 189–216, https://doi.org/10.1023/B:CLIM.0000013685.99609.9e, 2004.

**Response to Anonymous Referee #4**

This paper by Legrand et al. presents and discusses the results of a modelling chain that uses large-scale atmospheric information (ERA-20C reanalysis) to produce continuous daily river discharge at several stations of the Upper Rhône river catchment (~ 11000 km²). This modelling chain includes a semi-distributed hydrological model (GSM-SOCONT) and a downscaling model. Two downscaling models (statistical or dynamical) are actually used and tested. The results are analysed over several periods (1902-2009, 1961-2009, 1981-1983, 1920-1949, 1950-1979, 1980-2009) and according to various indicators (daily discharges, mean monthly and annual discharges, low and high flows).

This paper clearly corresponds to a huge amount of work. A lot of material and results are presented, the Figures are very rich.

We thank Referee #4 for his/her encouraging general comments.

The drawback of this very rich content is that it is not always easy for the reader to navigate and appreciate the results in the present form of the manuscript. My main remarks and recommendations to improve the paper are listed below :

Title and focus.

**RC4.1** The paper is entitled "Simulating one century (1902-2009) of river discharges, low flow sequences and flood events of an alpine river from large-scale atmospheric information" which is a very general title and does not really correspond to the real focus of the paper. In my opinion, as stated by the authors themselves, the objective of the work presented here is to assess the ability of downscaling chains to simulate hydrologically relevant weather scenarios (p12, l. 282-284). This is consistent with the detailed description of the different steps of the downscaling models, and the presentation of the results as comparison to a reference that is not the observed discharge (but rather the discharge simulated by the hydrological model forced by reference observations). I therefore suggest to make this objective more clear in the Introduction, to change the title accordingly and to trim the parts of the text that do not contribute directly to this objective (a lot of details about the hydrological model description and set up could be placed in supplementary material for example).

**AC4.1** Thank you for this comment.

In the introduction, we now explain more clearly our aim to better understand the crucial steps required to simulate hydrologically relevant weather scenarios.

Note that a description of the novelty of our study has been included in the introduction, as this point was also highlighted by Referee #1 and Referee #3:

We combine three innovative features. i) We evaluate the modeling chains in contrasting and demanding hydro-meteorological configurations where the interplay between weather variables, both in space and time, is determinant. The Upper Rhône River (URR) catchment, a mesoscale alpine catchment straddling France and Switzerland, indeed presents a number of different hydrological regimes, ranging from highly glaciated ones in its upper part to mixed ones dominated by snow and rain downstream. ii) We evaluate the modeling chains over the entire 20th century, a period long enough to assess the ability of the modeling chains to

reproduce daily variations in observed discharges, low frequency events (low flow sequences and annual discharge maxima), and variations in low flow and flood activities. iii) For both downscaling models, we evaluate the need for additional bias correction prior to application of the hydrological model for precipitation, temperature and temperature lapse rates.

As suggested, we propose to replace the title of the study with the following words: "Assessing downscaling methods to simulate hydrologically relevant weather scenarios from a global atmospheric reanalysis: case study of the Upper Rhône River (1902-2009)"

We have decided to keep the part devoted to the hydrological model in the main article, as it contains crucial and innovative steps, in particular those related to the calibration process in the presence of dams along the river, as well as the discussion related to lapse rates.

In response to the different comments made by the 4 reviewers, we have also added a section in the discussion to address the limitations of the hydrological model.

Figures.

The Figures contain a lot of information, maybe too much compared to what the reader is able to see and to what is necessary to illustrate the author's point. They would gain a lot from a bit of trimming. As examples :

**RC4.2** Fig 1 : the names of the gauging stations on the map are not necessary (they are also in Fig 2) and prevent the reading.

**AC4.2** Thank you for this comment. We prefer to keep the names of the gauging stations in Fig. 1, as this helps to identify ungauged sub-basins and to locate the gauging stations where hydrological assessments will be made later.

For reasons of legibility, however, we have used shorter names.

**RC4.3** Fig 7 : is it really useful to present both MAT and DMAT ?

**AC4.3** In Fig. 8b for MAT, it is rather easy to see the monthly biases obtained between observations and SCAMP simulations. However, the bias for the MAR simulation is difficult to identify due to the high seasonality of the raw temperature variable. Fig. 8c is given to highlight this "small" bias, which nevertheless has strong implications on simulated hydrology. We therefore think it is important to keep both figures.

**RC4.4** Figs 8 and 9 : these Figs are more illustrative, maybe present more synthetic results before (Fig 10) + it could be worth presenting a Table with a few synthetic values (bias, KGE) so that we can have a general idea of how the simulation chains perform compared to reference. Why presenting simulations with BC before without BC ? + is it really interesting to present the time series of lake level ? + add the names of the stations directly on the Figures

**AC4.4** Presenting a Table with a few synthetic values**.**

Thank you for this suggestion. A Table with the NSE coefficients for the dynamical downscaling model (MAR/MAR-BC) and the statistical index CRPSS (Continuous Ranked Probability Skill Score) for the statistical downscaling model (SCAMP/SCAMP-BC) has been added to show the performance gain associated with bias correction. This is Table 2.

Why presenting simulations with BC before without BC?

In the section devoted to the evaluation of SCAMP and MAR weather scenarios, we highlight the significant biases for both MAP and MAT variables. It was therefore logical for us to consider that hydrological simulations have to be performed with bias-corrected weather scenarios. This is why we present these results first. However, the presentation of the simulations with the raw scenarios follows. This allows us to show that the discrepancies obtained on discharges do not correspond to what we could have expected from the MAP biases, and this because of the MAT biases. In our opinion, this makes the demonstration of the importance of MAT biases more effective. This is the main reason why simulations without bias correction follow simulations with bias correction.

Is it really interesting to present the time series of lake level?

As the lake has a significant buffering effect on downstream hydrology, due to storage cycles, it was also important to check that the model is able to simulate the main features of storage variations.

Add the names of the stations directly on the Figures.

As suggested, the name of the gauging stations has been added on the figures.

**RC4.5** Fig 11 : too many lines on Fig c) + Figs a) and b) don't work. It is not clear that there is a "column" for each period + the only thing we see is the comparison between SCAMP and SCAMP-BC for each period

> **AC4.5** Thank you for this comment. The columns for the 3 periods have been separated more clearly in Fig. 12a and b, and for greater clarity, Fig. 12c has been split into two figures.

**RC4.6** Fig 12 : Fig a) not legible. What am I supposed to see on Fig b) ? Why not removing MAR simulation (since lapse correction is only done to the MAR-BC simulation) ?

> **AC4.6** The size of the dots in Fig. 5a has been reduced for greater clarity. In Fig. 5a and b, note that the blue and red lines overlap. For greater clarity, this point has been added to the caption.
>
> In Fig. 13, we show that the simulation with a classical bias correction of the Mean Areal Temperature (MAT) (bias correction of the mean for a reference altitude - black curve) leads to worse simulated discharges than those obtained without any bias correction (cyan curve). It is therefore useful to show the MAR simulation. The problem is clearly the bias in the temperature lapse rate. Simulations with double bias correction, i.e. i) bias correction of MAT at a reference altitude, and ii) bias correction of temperature lapse rates, perform well (dark blue curve). Fig.5 a, b and c are different ways to highlight the importance of the bias for the temperature lapse rate. To the best of our knowledge, the discrepancy between the observed and simulated statistical distributions of the temperature lapse rate has never been recognized or illustrated before. In our opinion, it is therefore useful to highlight this issue here.

**RC4.7** Fig 13 : Fig b) : why lines and crosses on this Fig for the different simulations ?

> **AC4.7** Thank you for this comment. We have also used lines for the MAP estimates with the two different precipitation lapse rates.

Acronyms and notations.

**RC4.8** The paper uses many acronyms that are sometimes confusing and could be simplified. I particularly struggled with MAP / MAR / MAT (the name of the model can't be changed, but the names of the variables probably can), with basin-scale, RHHU, grid points, station (Fig 3 and accompanying text), see next remark. BC (Bias correction) is also not clear as it is used for different corrections (not the same for MAR and SCAMP, which I can understand, but also for temperature lapse-rate for MAR. About this last correction, if it is included in the results presented in section 5 (which is the case from what I can understand), why is it not presented along with the other corrections in section 4 ?

> **AC4.8** Thank you for this suggestion. For the notations, we have specified that we are referring to the dynamical downscaling model when we use the notation MAR.
>
> We have used different terminologies: "mean areal precipitation" and "mean areal temperature" for hydrological units, MAP and MAT for regional or sub-regional scales.
>
> Sect. 4.4 "Bias Correction" has been clarified and the equation for the lapse rate correction has been moved there, where we discuss the bias correction for MAP and for MAT.
>
> Note that the principle of bias correction is the same for both models (MAR and SCAMP) and for all variables, i.e. MAP, MAT and temperature lapse rates. It is a quantile mapping bias correction. We simply estimate the correction function on different time series, depending on the model and variable considered.

Spatial resolution

**RC4.9** I really struggled (and did not succeed) at understanding exactly what was done in terms of spatial aggregation / disaggregation for the weather scenarios in SCAMP. In step (1) how do to change from station (= point) observations to basin-scale resampled values (basin-scale = whole catchment or subcatchments?). Are the stations at the end of the process the same as the observation stations ? (I understand rather RHHU = subcatchment but later in the text it is still referred as neighbouring stations ad inputs to the hydrological model, see p 22 l 410-411). Similarly Fig 5 presents the numerical experimentation plan with the hydrological model forced by P and T at 7*7 km resolution. Please, make all this clearer !!

> **AC4.9** Thank you for this comment. There was indeed a mistake in Fig. 6, where disaggregated data are shown for MAR and SCAMP. SCAMP indeed produces precipitation and temperature time series at station level, and the stations at the end of the process are exactly the same as the observation stations. Sorry for this. We understand it was almost impossible to understand the spatial aggregation/disaggregation process. We hope it is now clear.
>
> Fig. 6 has been corrected: the spatial resolution of P/T is "stations" for obs and SCAMP, and "7x7 km" for MAR.

Time periods and indicators for the results.

**RC4.10** As said before, there are a lot of results, presented for various times periods and various indicators that change from Figure to Figure. This does not make the reader's task easy. I would be nice, along with the experimental setup, to define in a Table the various indicators and time periods used and explain why they were selected.

> **AC4.10** Thank you for the suggestion. We have added a table to the Experimental setup section, with the assessment objectives, variables, time periods and areas considered for the evaluations. This is Table 1.

**RC4.11** If I am not mistaken, the results of the calibration of the hydrological model for the reference simulation are not presented. It would be nice to add a few elements about this, just to make sure that the reference model is not completely off the track.

> **AC4.11** The calibration efficiency is already mentioned in Supplementary Materials (Table S2):
> - For all sub-basins for which a classical calibration was carried out, we give the NSE coefficient calculated from observed and simulated time series of discharge.
> - For all sub-basins for which a signature-based calibration was carried out, we give the NSE coefficient calculated from observed and simulated regimes and the Kolmogorov-Smirnov coefficient calculated from observed and simulated distributions of annual discharge maxima.

Reference MAP and MAT.

**RC4.12** I am not a specialist of estimation of weather variables in montainous areas but I am surprised by the methodology used here, which consists of Thiessen polygons with a density of stations that is not so high (62 raingauges for ~ 11000 km², even less temperature stations). There are several other methods for the estimation of areal P and T, from very simple such as inverse-distance weighting to more complicated (kriging). What is the reason of choosing such a simplistic approach ? How confident can we be with these "reference" values ? This should at least be discussed thoroughly.

> **AC4.12** As mentioned previously, spatial variations in weather in a mountainous environment can be obviously significant, often due to the effect of topography. A satisfactory estimate of MAT and MAP is however impossible to achieve from stations only, even in this URR configuration where the network of precipitation and temperature stations is denser than in many basins worldwide. In our experience, it is even rather inextricable.
>
> The inverse distance is of course interesting, but has other limitations: the distance to be considered in a mountainous environment is not trivial at all. Two stations distant by 5 km in two different neighboring valleys are probably much more distant than 2 stations distant by 10 km in the same valley. A topographical distance has been proposed in some papers, but the choice of weight linked to topography has to be estimated and is always based on different assumptions and choices that are rarely justified (at least from a physical point of view).
>
> Using an inverse distance approach to estimate MAP is not necessarily straightforward either. It would be necessary to estimate the weights of all stations for all (grid) points in the spatial unit considered. In most cases, the weights are estimated from the distances to the centroid

of the spatial unit, which is also an important and not really satisfactory assumption. Let's consider 2 basins A and B of the same shape, very elongated (this is often the case in the URR catchment), with the same surface area and centroid. The only difference between the two would be their orientation: basin A is oriented in a West-East direction, basin B in a North-South direction. With an inverse distance approach, all weather stations would have exactly the same weights for both basins, which would obviously not be the case in reality.

Kriging would also be interesting. We could try Kriging with external drift, where the drift is linked to the topography. A number of works have been presented on this subject. However, they generally apply to interannual mean variables, i.e. mean annual precipitation, mean annual temperature. At these aggregated scales, there is indeed a clear relationship between meteorological conditions and altitude. However, this relationship is no longer valid for each individual time step of the period to be considered. As mentioned in the article, the temperature-elevation relationship (if any) varies from day to day. Kriging is much more computationally demanding than Thiessen polygons. It also requires the calibration of a functional relationship (the variogram model). The quality of this model must be verified. It must be verified for each time step, which would be really difficult to do.

For precipitation, as mentioned in the article, the nonexistent precipitation-elevation relationship at high resolution and at the scale of precipitation events is another important limitation.

For these reasons, and for the sake of simplicity, we have chosen to work with the Thiessen method. Note that estimates of MAP and MAT are obtained from a rather high number of stations (5 to 10 for each of the 18 sub-basins for example for precipitation). We therefore expect not to produce very poor estimates of MAP and MAT.

Note also that this choice should not influence the main results of our work, notably the critical issue of bias in downscaling models, especially that of the temperature lapse rate for dynamical downscaling, and the critical issue of the precipitation lapse rate at the event-scale.

We agree that this point should ideally be discussed in depth. However, the article is rather long and covers many other issues.

We have added a short comment on this point in the discussion.

**RC4.13** p 24 the authors write "no precipitation-elevation relationship was considered" although p 6: "and a regional and time-varying elevation-temperature relationship".

**AC4.13** The issue of elevation dependency is different for precipitation and temperature.

For temperatures, the dependency is very strong and robust for most time steps, and is accounted for in (likely) all hydrological models (for mountainous areas at least, where snow is important) with a linear temperature-elevation relationship. The slope (the lapse rate) of this relationship can vary from one time to another. We have accounted for this temporal variability, estimated from station observations.

For precipitation, as discussed in Sect. 6.3, there is no clear precipitation-elevation relationship at the event-scale, and the inclusion of the mean relationship observed for a long time period is problematic for flood simulation. This is why our simulations do not take into account a precipitation lapse rate. To the best of our knowledge, the precipitation-elevation issue was not described in previous works using hydrological models. However, it is potentially critical. We hope that our work will lead scientists to pay more attention to this issue in future studies.

Floods.

**RC4.14** I have a few comments on the "flood" part of the paper. First, I do not really agree with the use of the term "flood" given what is presented here. In 5.3 (Fig 10) what is done is picking the maximum daily flow for each year, which does not necessarily correspond to a flood. Similarly, flood events can be expected to last several days given the size of the catchment so I don't think that the max daily discharge can be called a flood event.

> **AC4.14** Thank you for this comment. We agree with the reviewer's comment: we analyze the annual maxima of daily flows and they do not always correspond to floods.
>
> We have specified in the revised version of the article that, for this part, we use the annual discharge maxima as flood proxy indicators and replaced the term "Flood events" with "Annual discharge maxima" in Fig. 11.

**RC4.15** In 5.4 (Fig 11), I suppose that the "flood activity" is defined as the number of days over the threshold, which again does not necessarily correspond to single flood events. The way of calculating the thresholds should also be precised (p 21, l 373-374 : are they defined through a flood frequency analysis as the value of the flood return period 1 year, or from the flow duration curve ? Therefore I would be more cautious with the term "flood" and would preferrably use terms like high flow indicators or maybe flood proxy indicators.

> **AC4.15** For flood activity, the threshold was calculated by considering the 90 largest daily discharges over the 90-year period. If the threshold was exceeded several times during five consecutive days, only the date of the maximum discharge was retained. In this analysis, several flood events could be identified in the same year. Here we come close to the definition of "flood" referred to by the reviewer.
>
> The low flow activity is similarly defined as the average number of months per 30-year period during which the mean monthly discharge is below a given discharge threshold. The threshold retained was again the reference discharge values exceeded on average once a year over the entire 90-year simulation period (1920-2009). It corresponds to the 10th percentile, i.e. the 90 lowest values of the mean monthly reference discharges over the 90-year period. Here again, several low flow sequences could be identified in the same year.
>
> These points have been clarified in the text.

**RC4.16** p 26, l 483-488 : the unrealistic simulated discharges obtained with altitude-elevation correction can also be due to the calibration of the hydrological model, considering in particular that the model was "high flow calibrated" (4.1.2). A rigorous way to test that would be to re-calibrate the

hydrological model with the new corrections added to the observations before testing them on the downscaling models.

AC4.16 Thank you for this comment. A recalibration would obviously be interesting, but based on previous analyses, we argue that it would not allow to fix this issue in a relevant way.

As mentioned in the article, if a precipitation-elevation relationship can be identified from aggregated (e.g. annual) precipitation data, this is not the case at the scale of individual events. This is illustrated by the daily rainfall amounts observed at the different stations in the catchment area for the 4 largest floods recorded during the period. As shown in Fig. A below, there is no dependency on elevation. For event 2, precipitations at the highest elevation are even lower than in the lowlands.

As shown in Fig. B below, a precipitation lapse rate of 10 %/100 m significantly increases the precipitation amounts, even for the heaviest precipitation events. On average, the annual MAP maxima at the URR basin scale are increased by 20 %. Due to the high non-linearity of the production processes, this 20 % increase leads to a much higher increase in peak flows. A recalibration of the model would reduce the "productivity" of the catchments, i.e. it would increase the storage capacity of the soil reservoir. The filling rate of this reservoir determines the runoff coefficient of the catchment for that period, and the lower the capacity, the easier it is to fill the reservoir. To avoid huge and unrealistic floods in a +10 %/100 m lapse rate configuration, a much higher soil capacity is therefore required. Previous analyses show, however, that this can be detrimental for smaller (and more numerous) rainfall-runoff events, leading to a number of minor floods being underestimated.

[Figure]

Figure A.

[Figure]

Figure B.

---

## Author Response (AR2)

We are pleased to send you the final version of our manuscript. We have taken into account the last comments and suggestions of Referee #4 and Referee #1. Our responses are detailed below.

**Response to Anonymous Referee #4**

I really appreciated the careful revision of the paper made by the authors. The responses to the reviewers' comments are very complete. Substantial modifications were made to the paper that, in my opinion, improve it a lot and highlight the very interesting methodology and results about producing relevant weather data in a large mountain catchment.

We thank Referee #4 for his/her very positive feedback.

I still have some minor comments.

L 76: « low flow and flood activities »: I know that this is defined later in the text, but it would be really nice to reformulate of provide a brief definition here so the reader knows what it is about (maybe « low flow and flood occurrence » ?)

At the end of the sentence, we added the following text in brackets: « (rate of occurrence of flood/low flow discharges above/below a given threshold) »

Fig 1: Something is still not clear on the bottom map. The locations of the gauging stations / hydro network do not match with the boundaries of the subcatchments (Viège, Taninges); it makes understanding the text difficult at several places (e.g. L 205; L 219-220; L 360-366; ...)

We agree that the locations of Viège and Taninges suggest some discrepancy with the sub-basin boundaries, but both locations are well located on the hydrographic network and match with the boundaries of sub-basins 2 and 14. For reasons of clarity, we have only shown the main hydrographic network of the Rhône River and its main tributaries on the map at the bottom of Fig. 1. The complete hydrographic network is, however, visible on the map at the top of Fig. 1.

L 96-98: Do you really need to refer to French meteorological jargon « retour d'Est »; « Binn-Simplon » ? This is not used in the rest of the text, and the description of the typical precipitation patterns you provided is enough and much more understandable for an international audience.

We agree that this is likely not of capital importance to an international audience, but both situations are well known to French and Swiss scientists and operational partners. We have kept these references in brackets in the text.

L 120: Maybe you could provide a short list of the variables that can be found in the ERA-20C reanalysis (for those who are not familiar with this type of data).

We have extended the sentence to: « This reanalysis provides 6-hourly data over the 1900-2010 period at a 1.25° spatial resolution for a number of atmospheric variables (e.g. geopotential height, wind speed, temperature and humidity of air masses). »

L 145: Indicate here that the choice of Thiessen method is discussed further (section 6.4)

Thank you for this comment. We will add this sentence.

« The choice of the Thiessen's weighting method is discussed in Sect. 6.4. »

Section 4.2.2: calibration method: The authors argued that the calibration methodology presented here is original (to deal with influence obs data) and therefore should be kept in the main text of the paper. I agree, but in that case the originality of this methodology should be highlighted. Maybe not in the main objectives in the paper, but in the conclusion and abstract, as a secondary (yet very interesting) result.

Thank you for this suggestion. We slightly modified the conclusion to mention this point. It now reads as follows:

« The originality of this study is fourfold. i) We evaluated the modeling chains in contrasting and demanding hydro-meteorological configurations where the interplay between weather variables, both in space and time, is determinant. ii) The spatio-temporal relevance of the weather scenarios is assessed by their hydrological responses, simulated using an ad hoc hydrological model at several gauging stations. iii) The simulations cover the entire 20th century, a period long enough to assess the ability of the modeling chains to reproduce daily variations in observed discharges, low frequency events, and variations in low flow and flood activities. iv) For both downscaling models, we evaluated the need for additional bias correction of the weather scenarios, including that of temperature lapse rates.

This framework allowed to highlight important criteria to be met for the simulation of relevant hydrological scenarios for the Upper Rhône River catchment. The alpine configuration of the Upper Rhône River catchment (unknown effects of the large upstream dams and of the regulation of Lake Geneva, scarcity of concomitant weather/discharges observations) made the calibration of the hydrological model rather difficult. This required the development of an original multiple calibration strategy, based on both observed discharge time series and hydrological signatures.

For both modeling chains, given this difficult modeling context and the fact that the weather scenarios are only produced from large-scale atmospheric information, the simulated discharges are globally in good agreement with the reference ones. »

L 219-220 : not clear. Rephrase « the four URR sub-basins that present a natural regime »

Ok, rephrased.

« To assess this, we recalibrated the parameters of the four URR sub-basins that present a natural (or at least not significantly altered) hydrological regime, using only the hydrological signatures. »

5.3: reformulate the title to match the changes that were made to the text. e.g. « low frequency hydrological events »? + following sentence « with simulations of low frequency hydrological events, i.e. low flow sequences and annual discharge maxima used as flood proxy indicators »

Title: « Flood events and low flow sequences » was rephrased to: « Floods and low flows »

The sentence « The hydrological relevance of simulated weather scenarios is further evaluated with simulations of low flow sequences and flood events. Note that the annual discharge maxima are used as flood proxy indicators. » was changed for:

« The hydrological relevance of simulated weather scenarios is further evaluated with simulations of floods and low flows. Note that the annual daily discharge maxima are used as flood proxy indicators and that the annual monthly discharge minima are used as low flows proxy indicators ».

Figure 11. « Scatter plots of mean monthly discharges, low flows and annual discharge maxima at … » was changed for: Figure 11. « Scatter plots of mean monthly discharges, annual monthly discharge minima and annual daily discharge maxima at … »

L 455: Define of reformulate « flood and low flow activities »

The terms « flood and low flow activities » are explained in more detail a few lines later in the text (from line 465).

6.3: The discussion about Thiessen's method would match better in this section + reformulate a more general and understandable title such as « Estimation of precipitation from observations ».

Thank you for this comment. In a former version, we first presented the discussion of the Thiessen's method in this section, but this added noise to the main message we wanted to highlight (the issue of the precipitation lapse rate and its effect on simulations).

The lapse rate issue would be the same whatever the spatialisation method. On the other hand, the issue of the spatialisation method refers to the modeling problems that can be encountered in hydrological modeling. This why we have finally chosen to discuss the choice of the Thiessen's method in Sect. 6.4.

6.4: I don't understand the title. Hydrological model limitations ?

The title refers to the key message of this section. Whatever the limitations of the hydrological model, it is important to make an impact-oriented evaluation of weather scenarios. Here, the evaluation of the hydrological scenarios obtained from the generated weather scenarios allowed to highlight some important issues that we could have missed with an evaluation of weather scenarios only.

We have changed the title for:

« The hydrological model: a powerful assessment tool despite its limitations »

**Response to Anonymous Referee #1**

I would like to thank the authors for considering my initial comments. All my major concerns (including formulation of the novelty and process-based interpretation of results) have been adequately addressed. Finally, I have only few minor comments, which might be considered before final publication. The manuscript presents a complex and robust analysis, congratulations to it!

We thank Referee #1 for his/her very positive feedback.

Minor comments

1) Abstract: « The low frequency hydrological situations, such as low flow sequences and annual discharge maxima (used as flood proxy indicators) are reasonably well reproduced. … The results for low flow activity are less satisfactory. » This is not clear. What is the meaning of reasonably well and less satisfactory? Some quantification here will be useful. Perhaps replace also low frequency situations with extreme hydrological situations (to reduce the word low used in different contexts).

Thanks for these suggestions. We will change the text for:

« The low frequency hydrological situations, such as annual monthly discharge minima (used as low flows proxy indicators) and annual daily discharge maxima (used as flood proxy indicators) are reasonably well reproduced. The observed increase in flood activity over the last century is also rather well reproduced. The observed low flow activity is conversely overestimated, and its variations from one sub-period to another are only partially reproduced. »

2) Flood and low flow activities. The definitions of these « activities » are rather unusual, so difficult to assess their interpretations or compare it with other modeling evaluations. The word « activity » is a bit confusing in that context. What about to compare the distributions of annual Q95 (as a low flow index) and Q5 (as a high flow index)?

We agree that different definitions are possible for « flood and low flow activities ».

As mentioned in our conclusion, the follow-up to our work (see Legrand, 2024) is to force the SCAMP/GSM-SOCONT chain with a CMIP6-PMIP4 paleosimulation ensemble (Jungclaus et al., 2017; Kageyama et al., 2018) to assess the variations in hydro-meteorological regimes of the Upper Rhône River catchment over the last millenium and to confront the simulated variations in flood activity with those obtained in previous works from the sediments archives of Lake Bourget (Jenny et al., 2014; Evin et al., 2019; Wilhelm et al., 2022).

We defined « flood activity » similarly to what is often done in paleohydrology (cf. Vasskog et al., 2011; Ballesteros-Cánovas et al., 2015; Wilhelm et al., 2022), which refers to the frequency of occurrence of flood events. In most paleohydrological studies, flood events are identified from paleoenvironmental archives from tree rings, lacustrine or channel sediments. Here, floods are identified from discharges (observed or simulated) that exceed a given threshold.

For the sake of consistency, the low flow activity was defined in a symmetrical way.

3) Discussion. Please add some discussion about the impact/uncertainty of the objective function used for model calibration. Estimation of low flows and their changes (and scenarios) might be significantly impacted by the way hydrologic model is calibrated. Perhaps you learned some experience during preparation of presented results.

We added the following sentences:

The signature-based calibration of the model, used for sub-basins with altered discharge data, is also not optimal. « The objective function considered for the calibration, for instance, gives considerable weight to the statistical distribution of annual daily discharge maxima. Other results may be obtained adding criteria for low flows (e.g. distribution of annual monthly discharge minima). This will be worth further investigations. On the other hand », parameters were calibrated so that simulated signatures reproduce at best observed ones, but observed and simulated signatures come from different periods.

**References**

Legrand, C.: Simulation des variations de débits et de l'activité de crue du Rhône amont à partir de l'information atmosphérique de grande échelle sur le dernier siècle et le dernier millénaire, Thèse de doctorat, Université Grenoble Alpes, France, 2024.

Vasskog, K., Nesje, A., Støren, E. N., Waldmann, N., Chapron, E., and Ariztegui, D.: A Holocene record of snow-avalanche and flood activity reconstructed from a lacustrine sedimentary sequence in Oldevatnet, western Norway, The Holocene, 21, 597–614, https://doi.org/10.1177/0959683610391316, 2011.

Ballesteros-Cánovas, J. A., Rodríguez-Morata, C., Garófano-Gómez, V., Rubiales, J. M., Sánchez-Salguero, R., and Stoffel, M.: Unravelling past flash flood activity in a forested mountain catchment of the Spanish Central System, J. Hydrol., 529, 468–479, https://doi.org/10.1016/j.jhydrol.2014.11.027, 2015.